# Evaluation of Transport Processes over North China Plain and Yangtze River Delta using MAX-DOAS Observations

Yuhang Song[1],#, Chengzhi Xing[2],#, Cheng Liu[1,2,3,4],*, Jinan Lin[2], Hongyu Wu[5], Ting Liu[7], Hua Lin[5], Chengxin Zhang[1], Wei Tan[2], Xiangguang Ji[5], Haoran Liu[6], and Qihua Li[6]

[1]Department of Precision Machinery and Precision Instrumentation, University of Science and Technology of China, Hefei, 230026, China

[2]Key Lab of Environmental Optics & Technology, Anhui Institute of Optics and Fine Mechanics, Hefei Institutes of Physical Science, Chinese Academy of Sciences, Hefei, 230031, China

[3]Center for Excellence in Regional Atmospheric Environment, Institute of Urban Environment, Chinese Academy of Sciences, Xiamen, 361021, China

[4]Key Laboratory of Precision Scientific Instrumentation of Anhui Higher Education Institutes, University of Science and Technology of China, Hefei, 230026, China

[5]School of Environmental Science and Optoelectronic Technology, University of Science and Technology of China, Hefei, 230026, China

[6]Institute of Physical Science and Information Technology, Anhui University, Hefei, 230601, China

[7]School of Earth and Space Sciences, University of Science and Technology of China, Hefei, 230026, China

# These two authors contributed equally to this work.

*Correspondence to*: Cheng Liu (chliu81@ustc.edu.cn)

**Abstract.** Pollutant transport has a substantial impact on the atmospheric environment in megacity clusters. However, owing to the lack of knowledge of vertical pollutant structure, quantification of transport processes and understanding of their impacts on the environment remain inadequate. In this study, we retrieved the vertical profiles of aerosols, nitrogen dioxide ($NO_2$), and formaldehyde (HCHO) using multi-axis differential optical absorption spectroscopy (MAX-DOAS) and analyzed three typical transport phenomena over the North China Plain (NCP) and Yangtze River Delta (YRD). We found the following: (1) The main transport layer (MTL) of aerosols, $NO_2$ and HCHO along the southwest–northeast transport pathway in the Jing-Jin-Ji region were approximately 400–800 m, 0–400 m and 400–1200 m, respectively. The maximum transport flux of HCHO appeared in Wangdu (WD), and aerosol and $NO_2$ transport fluxes were assumed to be high in Shijiazhuang (SJZ), both urban areas being significant sources feeding regional pollutant transport pathways. (2) The NCP was affected by severe dust transport on March 15, 2021. The airborne dust suppressed dissipation and boosted pollutant accumulation, decreasing the height of high-altitude pollutant peaks. Furthermore, the dust enhanced aerosol production and accumulation, weakening light intensity. For the $NO_2$ levels, dust and aerosols had different effects. At the SJZ and Dongying (DY) stations, the decreased light intensity prevented $NO_2$ photolysis and favored $NO_2$ concentration increase. In contrast, dust and aerosols provided surfaces for heterogeneous reactions, resulting in reduced $NO_2$ levels at the Nancheng (NC) and Xianghe (XH) stations. The reduced solar radiation favored local HCHO accumulation in SJZ owing to the dominant contribution of the primary HCHO.

(3) Back-and-forth transboundary transport between the NCP and YRD was found. The YRD-to-NCP and NCP-to-YRD transport processes mainly occurred in the 500–1500 m and 0–1000 m layers, respectively. This transport, accompanied by the dome effect of aerosols, produced a large-scale increase in $PM_{2.5}$, further validating the haze-amplifying mechanism.

## 1 Introduction

With rapid economic development, urbanization in China has increased. Many cities of different scales have recently emerged, forming megacity clusters such as Jing-Jin-Ji (JJJ) and the Yangtze River Delta (YRD). With this rapid urbanization, air pollution has become one of the most serious environmental threats that China must address. Heavy air pollution adversely affects every aspect of human life, including climate, air visibility, and human health (Pokharel et al., 2019; Gao et al., 2017; Su et al., 2020a; Li et al., 2017a).

Currently, air pollution sources can be broadly classified as direct emissions, secondary production, and transport. Transport contributes significantly to pollution in some megacities. Firstly, transport carries large amounts of pollutants, directly deteriorating air quality. Regional transport plays a predominant role in pollution formation in many major cities in China, such as Beijing, Shanghai, Guangzhou, Hong Kong, Hangzhou, and Chengdu, contributing more than 50% of the particulate matter, $PM_{2.5}$, during polluted periods (Sun et al., 2017). In the JJJ region, regional transport from southwest to northeast, driven by southwesterly winds, is the dominant influence on the daytime increase in $PM_{2.5}$ and ozone ($O_3$) concentrations (Ge et al., 2018). In addition to regional transport, cross-regional transport has a significant impact. For example, from 2014 to 2017, intra- and inter-regional transport accounted for 25% and 28% of the total $PM_{2.5}$ in the JJJ region, respectively, while the local contribution was 47% (Dong et al., 2020). During the 2019 National Day parade, cross-regional dust contributed more than 74% of Beijing's particulate matter (PM) concentrations below 4 km (Wang et al., 2021). Furthermore, under certain conditions, some transported pollutants can interact with the planetary boundary layer (PBL) and create an environment favorable for direct emission accumulation and secondary formation enhancement, thereby indirectly amplifying the impacts of pollution (Li et al., 2017b; Wilcox et al., 2016; Petaja et al., 2016). A typical example is aerosols from the YRD being transported to the upper PBL over the North China Plain (NCP), which decreases the PBL heights and increases pollutant accumulation (Huang et al., 2020). The movement of warm and humid air masses likely increases secondary aerosol formation by aggravating aqueous and heterogeneous reactions (Huang et al., 2014). Hence, we must understand the air pollutant transport that occurs in megacity clusters by using an appropriate measurement method.

Current technological means of monitoring and analyzing air pollution mainly include in situ measurements, satellite observations, model simulations, and ground-based remote sensing monitoring. By 2021, the number of China National Environmental Monitoring Centers (CNEMCs) that provide in situ measurements had extended to 2734, forming a comprehensive and mature air quality monitoring network. CNEMCs monitor many pollutants, including sulfur dioxide ($SO_2$), carbon monoxide (CO), nitrogen dioxide ($NO_2$), $PM_{10}$, $PM_{2.5}$, and $O_3$. However, the pollutant concentrations monitored by CNEMCs are limited to the surface. Characterizing pollutants in the upper-level air column using surface observations is difficult (Huang et al., 2018b) because various factors, including local emissions, regional transport, geographical factors, and meteorological conditions, must be considered (Tao et al., 2020; Che et al., 2019). Therefore, the vertical distribution of

pollutants cannot be diagnosed using the CNEMC dataset alone. Satellite observations can be used to investigate the horizontal distribution of vertical column densities (VCDs) of $NO_2$, formaldehyde (HCHO), $O_3$, and aerosols on a global scale, providing support for horizontal pollutant transport analysis. However, because of their limited temporal and spatial resolutions, satellite data cannot be used for the continuous monitoring of a specific area (Bessho et al., 2016; Veefkind et al., 2012). It is difficult to characterize the vertical distribution of atmospheric composition using only satellite remote sensing or CNEMC data. Chemical transport models can be used to simulate pollutant distribution and they are also important tools for monitoring, forecasting, and analyzing atmospheric quality (Huang et al., 2018a). However, considerable uncertainties remain in estimating pollutant distribution using model simulations, primarily owing to the effects of emission inventories, meteorological fields, and of assumptions made (Grell et al., 2005; Huang et al., 2016; Xu et al., 2016; Zhang et al., 2017). Moreover, model simulations cannot completely characterize air composition profiles because of inadequate modeling of atmospheric pollutants in the vertical direction. To meet the need to understand the vertical distribution of air pollutants, some monitoring methods have been developed, such as light detection and ranging (LiDAR) (Collis, 1966; Barrett and Ben-Dov, 1967) and in situ monitoring instruments carried by aircraft, balloons, or unmanned aerial systems (UASs) (Corrigan et al., 2008; Tripathi et al., 2005; Ferrero et al., 2011). Nevertheless, the number of detectable pollutants is limited for a single LiDAR device and a single set is expensive. Alternatively, monitoring based on moving platforms requires substantial labor and material resources, which prevents continuous observation.

The differential optical absorption spectroscopy (DOAS) technique (Platt and Stutz, 2008) is a well-established and reliable method for the quantitative analysis of many crucial atmospheric gases. The DOAS method uses high-frequency molecular absorption structures in the ultraviolet (UV) and visible regions of the spectrum. Multiaxis differential optical absorption spectroscopy (MAX-DOAS), which employs the DOAS technique at multiple elevation angles, is used for long-term atmospheric quality monitoring (Hönninger et al., 2004). Combined with radiative transfer modeling, MAX-DOAS can be used to retrieve the vertical profiles of aerosols and trace gases based on scattered sunlight signals from multiple elevation angles (Frieß et al., 2006). This method has been widely used to retrieve aerosols, HCHO, $NO_2$, $O_3$, and glyoxal (CHOCHO) concentrations (Hönninger and Platt, 2002; Meena, 2004; Wagner et al., 2004; Frieß et al., 2006; Hönninger et al., 2004; Irie et al., 2008; Xing et al., 2020; Hong et al., 2022b). Compared with the above techniques, MAX-DOAS has many advantages such as simple design, low power demand, possible automation, low cost, and minimal maintenance. Moreover, the MAX-DOAS is capable of operating regularly in harsh environments, such as those over the Tibetan Plateau (Xing et al., 2021a). On this basis, we established approximately 30 MAX-DOAS stations covering seven regions in China (north, east, south, northwest, southwest, northeast, and central China) to form a mature ground-based remote sensing network (Xing et al., 2017; Liu et al., 2021; Hong et al., 2022a). This monitoring network successfully meets the actual demands for vertical observations, providing powerful support for analyzing pollution sources and transport (Liu et al., 2022).

In this study, we aim to understand the vertical distribution characteristics of air pollutants during transport and analyze their possible impacts on regions. The remainder of this paper is structured as follows. Section 2 describes the stations, instruments, algorithms, and ancillary data used in the study. In Section 3, we discuss three typical transport processes (regional, dust, and transboundary long-range transport). Finally, we present the summary and conclusions in Section 4.

## 2 Methods

### 2.1 Geographical locations and selected stations

Our study focused mainly on the transport phenomena in the NCP and YRD, two of the main plains within China. The NCP is partially enclosed by Mt. Taihang, Mt. Yan, and the Bohai Sea, whereas the YRD is close to the Yellow Sea and East China Sea. Many megacities are located in these two regions (i.e., Beijing and Tianjin in the NCP, and Shanghai and Nanjing in the YRD). Beijing, Tianjin, and Hebei Province form large megacity clusters within the NCP, named the JJJ region. Owing to numerous industries, combined with traffic emissions, the JJJ region is one of the most polluted areas in China. In addition, the JJJ has a typical continental monsoon climate, indicating that wind plays an important role in the local climate and environment. The semi-basin geographical features and continental monsoon climate indicate that regional transport is a significant factor affecting local air quality in the JJJ region. Similarly, the YRD is affected by several local pollution sources and by pollutant transport. Therefore, we selected eight MAX-DOAS stations in the NCP and YRD to explore the corresponding transport phenomena. Figure 1 depicts the topography of the NCP and YRD and the distribution of MAX-DOAS stations, and Table 1 lists the exact latitudes and longitudes of each station.

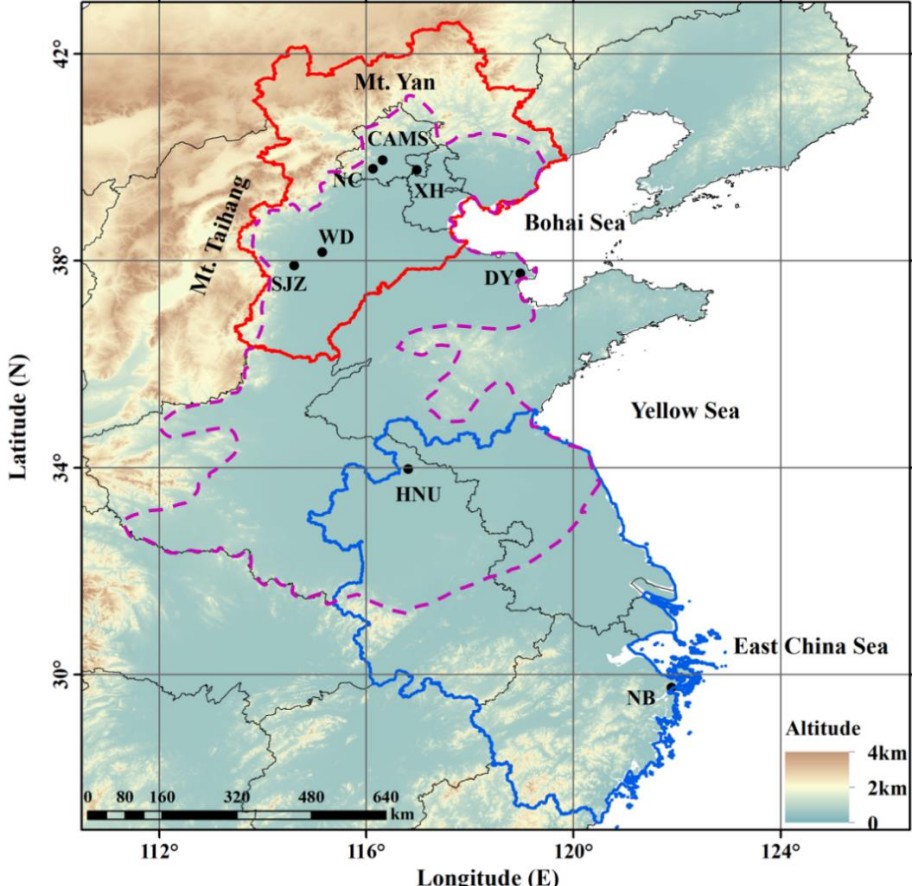

**Figure 1. Study area location, topography, and distribution of MAX-DOAS stations: black points represent the stations; solid red contour line indicates the JJJ region; solid blue contour line shows the YRD region. dashed purple contour line indicates the NCP region.**

**Table 1.** The names (codes), latitudes and longitudes of stations and their corresponding regions.

| Region | | Station (Code) | Longitude (°E) | Latitude (°N) |
|---|---|---|---|---|
| North China Plain (NCP) | Jing-Jin-Ji (JJJ) | Shijiazhuang (SJZ) | 114.61 | 37.91 |
| | | Wangdu (WD) | 115.15 | 38.17 |
| | | Nancheng (NC) | 116.13 | 39.78 |
| | | Chinese Academy of Meteorological Sciences (CAMS) | 116.32 | 39.95 |
| | | Xianghe (XH) | 116.98 | 39.76 |
| | | Dongying (DY) | 118.98 | 37.76 |
| Overlapping Zone | | Huaibei Normal University (HNU) | 116.81 | 33.98 |
| Yangtze River Delta (YRD) | | Ningbo (NB) | 121.90 | 29.75 |

125

## 2.2 Instrument setup

We operated eight commercial MAX-DOAS instruments (Airyx SkySpec-1D, Heidelberg, Germany) from January 1 to March 31, 2021, consisting of three major components: two spectrometers inside a thermoregulated box, a telescope unit, and a

computer for instrument control and data storage. One spectrometer covered the UV wavelength range (296–408 nm), and the

other worked in the visible region (420–565 nm), with a spectral resolution of 0.45 nm. Scattered sunlight was collected by a

telescope and then directed to the spectrometer through a prism reflector and quartz fiber. The instrument automatically

recorded the spectra of scattered sunlight at sequences that we set to 11 elevation angles: $1°$, $2°$, $3°$, $4°$, $5°$, $6°$, $8°$, $10°$, $15°$,

$30°$, and $90°$. The duration of each measurement sequence was approximately 5–15 min depending on the received radiance.

Additionally, the setup only collected scattered sunlight during daytime, whereas the dark current and offset spectra were

automatically measured at night and removed from all measured spectra before data analysis. To avoid the strong influence of

stratospheric absorbers, we filtered the measured spectra with a solar zenith angle (SZA) of $> 75°$ (Supplementary Sect. S1).

**Table 2.** Setting for the $O_4$, $NO_2$, HCHO, and nitrous acid (HONO) DOAS spectral analyses.

| Parameter | Data source | Fitting interval | | | |
|---|---|---|---|---|---|
| | | $O_4$ | $NO_2$ | HCHO | HONO |
| Wavelength range | | 338-370nm | 338-370nm | 322.5-358nm | 335-373nm |
| $NO_2$ | 220 K, $I_0$*correction (SCD of $10^{17}$ molecules cm$^{-2}$); (Vandaele et al., 1998) | × | × | √ | × |
| $NO_2$ | 294 K, $I_0$ correction (SCD of $10^{17}$ molecules cm$^{-2}$); (Vandaele et al., 1998) | √ | √ | √ | √ |
| $O_3$ | 223 K, $I_0$ correction (SCD of $10^{18}$ molecules cm$^{-2}$); (Serdyuchenko et al., 2014) | √ | √ | √ | √ |
| $O_3$ | 246 K, $I_0$ correction (SCD of $10^{18}$ molecules cm$^{-2}$); (Serdyuchenko et al., 2014) | × | × | √ | × |
| $O_4$ | 293 K, $I_0$ correction (SCD of $3×10^{43}$ molecules$^2$ cm$^{-5}$); (Thalman and Volkamer, 2013) | √ | √ | √ | √ |
| HCHO | 293 K, $I_0$ correction (SCD of $5×10^{15}$ molecules cm$^{-2}$); (Orphal and Chance, 2003) | √ | √ | √ | √ |
| BrO | 273 K, $I_0$ correction (SCD of $10^{13}$ molecules cm$^{-2}$); (Fleischmann et al., 2004) | √ | √ | √ | √ |
| Ring | Ring spectra calculated with DOASIS | √ | √ | √ | √ |
| HONO | $I_0$ correction (SCD of $10^{15}$ molecules cm$^{-2}$); (Stutz et al., 2000) | × | × | × | √ |

| | | | | |
|---|---|---|---|---|
| Polynomial degree | 4 | 4 | 4 | 4 |
| Intensity offset | Order 1 | Order 1 | Constant | Order 1 |

\* Solar $I_0$ correction (Aliwell, 2002);

### 2.3 Data processing

We analyzed the spectra using DOAS Intelligent System (DOASIS) spectral fitting software, which is based on the least squares algorithm (Kraus, 2006). Slant column density (SCD) is defined as the integrated concentration along the light path. Firstly, we calculated the differential slant column densities (DSCDs), which are defined as the difference between the off-

zenith and zenith SCDs. Subsequently, we analyzed the DSCDs of the oxygen dimer ($O_4$) and $NO_2$ in the interval between 338 and 370 nm, and we used the 322.5–358 nm and 335–373 nm wavelength intervals for HCHO and nitrous acid (HONO) absorption analysis, respectively (Xing et al., 2020; Xing et al., 2021b). Table 2 lists the DOAS fitting settings for $O_4$, $NO_2$, HCHO, and HONO in detail. The Ring spectrum was added to the fitting settings to remove the influence of inelastic rotational Raman scattering on solar Fraunhofer lines (Chance and Spurr, 1997; Grainger and Ring, 1962). In the fitting process, we

included a small shift or squeeze of the wavelengths to compensate for the possible instability caused by small thermal variations in the spectrograph. Figure S1 shows a typical DOAS fit for the four species. To ensure the validity of the retrieved data, we removed the DOAS fit results with a root mean square (RMS) larger than $1.0 \times 10^{-3}$. After applying the RMS threshold, the results for $O_4$, $NO_2$, HCHO, HONO remained at 69.8%, 71.6%, 64.8%, and 73.1% respectively. The DSCD detection limits were roughly estimated using two times of the mean RMS divided by the absorption cross-section (Nasse et al., 2019; Wang

et al., 2017; Lampel et al., 2015), which were $2.4 \times 10^{42}$ (molec$^2$ cm$^{-5}$), $1.7 \times 10^{15}$, $8.9 \times 10^{15}$, and $2.5 \times 10^{15}$ molec cm$^{-2}$ for $O_4$, $NO_2$, HCHO, and HONO, respectively. Moreover, to remove the effects of clouds, we used only data with slowly varying $O_4$ DSCDs and intensities for vertical profile retrieval (Supplementary Sect. S2). The inversion algorithm we used for aerosols and trace gases was based on the optimal estimation method (OEM), and we selected the radiative transfer model (RTM) library for radiative transfer (libRadtran) as the forward model (Mayer and Kylling, 2005). By minimizing the cost function

$\chi^2$, we determined the a posteriori state vector $\boldsymbol{x}$:

$$\chi^2 = (\mathbf{y} - F(\mathbf{x},\mathbf{b}))^T \mathbf{S}_\varepsilon^{-1} (\mathbf{y} - F(\mathbf{x},\mathbf{b})) + (\mathbf{x} - \mathbf{x}_a)^T \mathbf{S}_a^{-1} (\mathbf{x} - \mathbf{x}_a) \tag{1}$$

where $F(\boldsymbol{x}, \boldsymbol{b})$ is the forward model; $\boldsymbol{b}$ denotes the meteorological parameters (e.g., atmospheric pressure and temperature profiles); $\boldsymbol{y}$ is the measured DSCDs; $\boldsymbol{x}_a$ is the a priori vector that serves as an additional constraint; $\boldsymbol{S}_\varepsilon$ and $\boldsymbol{S}_a$ are the covariance matrices of $\boldsymbol{y}$ and $\boldsymbol{x}_a$, respectively. We classified the retrieval of vertical profiles of aerosols and trace gases in

two steps. As $O_4$ absorption is closely linked to the optical properties of aerosols, our first step was to retrieve vertical aerosol profiles based on the retrieved $O_4$ DSCDs at different elevation angles (Wittrock et al., 2003; Frieß et al., 2006; Wagner et al.,

2004). In the second step, using the retrieved aerosol extinction profiles as the input parameter to the RTM, we obtained the NO$_2$, HCHO, and HONO vertical profiles. In this study, we separated the atmosphere into 20 layers from 0 to 3 km with a vertical resolution of 0.1 km under 1 km, and of 0.2 km from 1 to 3 km. Exponentially decreasing functions with a scale height of 0.5, 0.5, 1.0, and 0.2 km were utilized as a priori profiles for aerosols, NO$_2$, HCHO, and HONO, respectively. For the aerosol profile retrieval, we set its aerosol optical density (AOD) to 0.4. For the a priori trace gas profile, we set the VCD to $1.5 \times 10^{16}$, $1.5 \times 10^{16}$, and $5 \times 10^{14}$ molec·cm$^{-2}$ for NO$_2$, HCHO, and HONO, respectively. The vertical distribution of trace gas above the retrieval height (3 km) was fixed to follow the U.S. Standard Atmosphere (Anderson et al., 1986). We set the a priori uncertainties of the aerosol, NO$_2$, and HCHO to 50%, and HONO to 100%, with a correlation height of 0.5 km. During the retrieval, we employed a fixed set of aerosol optical properties with a single-scattering albedo of 0.95, asymmetry parameter of 0.70, and surface albedo of 0.04. Figure S2 shows the averaging kernels, which indicate that the retrieval profile was sensitive to the layers within 0–1.2 km. The sum of the diagonal elements in the averaging kernel matrix is the degrees of freedom (DOF), which denotes the number of independent pieces of information that can be measured. The profiles of aerosols and trace gases were filtered out when DOF was less than 1.0 and the retrieved relative error was larger than 50% (Tan et al., 2018). About 0.5 %, 10.7 %, and 11.6 % of all measurements were discarded for aerosol, NO$_2$ and HCHO profile retrievals, respectively. A more detailed description of the retrieval process can be found in previous studies (Chan et al., 2019; Chan et al., 2018).

## 2.4 Error analysis

For profile retrieved results, we conducted an error analysis on the trace gas VCDs and AOD, and near-surface (0–100 m) trace gas concentrations and aerosol extinction coefficients (AECs). The error sources considered are listed below and the final results are summarized in Table 3. Detailed demonstrations and calculation methods can be found in Supplementary Sect. S3.

a.  Smoothing and noise errors refer to the errors caused by the limited vertical resolution of profile retrieval, and the fitting error of DOAS fits, respectively. By calculating the averaged error of retrieved profiles, we obtained the sum of smoothing and noise errors on near-surface concentrations and column densities, which were 24 and 5 % for aerosols, 11 and 19 % for NO$_2$, and 42 and 25 % for HCHO.

b.  Algorithm error (i.e., the difference between the measured and modeled DSCDs) mainly arises from an imperfect representation of the real radiation field in the RTM - spatial inhomogeneities of absorbers and aerosols, clouds, real aerosol phase functions etc. This error is a function of the viewing angle. However, it is difficult to assign discrepancies between the measured and modeled DSCDs at each profile altitude. Therefore, the algorithm error on the near-surface values and column densities cannot be realistically estimated. Given that measurements at 1 ° and 30 ° elevation angles are sensitive to the lower and upper air layers, respectively, the average relative differences between the measured and

modeled DSCDs for a 1 and 30° elevation angles can be used to estimate the algorithm errors on the near-surface values and column densities, respectively (Wagner et al., 2004). Considering its trivial role in the total error budget, we estimated these errors on the near-surface values and the column densities at 4 and 8 % for aerosols, 3 and 11 % for $NO_2$, and 4 and 11 % for HCHO, respectively, according to Wang et al. (2017).

c.  Cross section errors were 4, 3, and 5 % for $O_4$ (aerosols), $NO_2$, and HCHO, respectively (Thalman and Volkamer, 2013; Vandaele et al., 1998; Orphal and Chance, 2003).

d.  The errors related to the temperature dependence of the cross sections were then estimated. We multiplied the amplitude changes of the cross sections per Kelvin by the maximum temperature difference to quantify this systematic error. Given that the measurement period (from January 1 to March 31, 2021) is in the winter-spring season, we roughly estimated the maximum temperature difference to be 45 K. The corresponding errors for $O_4$ (aerosols), $NO_2$, and HCHO were approximately 10, 2, and 6 %, respectively.

e.  The trace gas retrieval errors, arising from the uncertainty in aerosol retrieval, were estimated as the total error budgets of the aerosols. Based on a linear propagation of the aerosol errors, the errors of trace gases were roughly estimated at 27% for VCDs and 14% for near-surface concentrations for the two trace gases. The perturbations of trace gas concentrations at each altitude caused by aerosol profile retrieval uncertainty resulted in a slight change in the profile shape. According to Friedrich et al. (2019), trace gas concentrations at 1.5-3.5 km respond most sharply to perturbations in the AEC profile, especially oscillations in the AEC below 0.5 km. The trace gas profile below 1.5 km shows a low sensitivity to AEC variation. Therefore, in this study, we focus mainly on the concentration variation below 1.5 km.

We calculated the total error by combining all the error terms in the Gaussian error propagation, which are listed in the bottom row of Table 3. Smoothing and noise errors played a dominant role in total error estimation.

**Table 3.** Averaged error estimation (in %) of the retrieved near-surface (0-100 m) trace gas concentrations and AECs, and trace gas VCDs and AOD.

|  | Near-surface | | | VCD or AOD | | |
|---|---|---|---|---|---|---|
|  | aerosol | $NO_2$ | HCHO | AOD | $NO_2$ | HCHO |
| Smoothing and noise error | 24 | 11 | 42 | 5 | 19 | 25 |
| Algorithm error | 4 | 3 | 4 | 8 | 11 | 11 |
| Cross section error | 4 | 3 | 5 | 4 | 3 | 5 |
| Related to temperature dependence of cross section | 10 | 2 | 6 | 10 | 2 | 6 |
| Related to the aerosol retrieval (only for trace gases) | - | 27 | 27 | - | 14 | 14 |
| Total | 27 | 30 | 51 | 14 | 26 | 32 |

## 2.5 Transport flux calculation and main transport layer definition

Owing to the semi-basin topography, southwesterly or southerly winds play a dominant role in pollutant transport in the JJJ region. In this study, we mainly focused on pollutant transport in the southwest-northeast direction, and thus selected four different stations along this pathway, namely, Shijiazhuang (SJZ), Wangdu (WD), Nancheng (NC), and Chinese Academy of Meteorological Sciences (CAMS) (Fig. 1). We calculated the hourly transport fluxes of each layer ($F_i$) and column transport fluxes ($F_c$) at each station to illustrate the dynamic transport process of pollutants along the southwest-northeast pathway. The detailed calculation methods are described below.

First, the wind speed projection (WS) in the southwest-northeast direction was calculated as follows:

$$WS = v \times \cos\frac{\pi}{4} + u \times \sin\frac{\pi}{4} \tag{2}$$

where $v$ and $u$ represent the meridional and zonal wind components, respectively; $WS$ above zero means that the wind came from the southwest and blew northeast, whereas $WS$ below zero has the opposite meaning.

Then, with $WS$, the $F_i$ can be obtained:

$$F_i = C_i \times W_i \tag{3}$$

Here, $C_i$ denotes the AEC ($km^{-1}$) or trace gas concentration (molec·$m^{-3}$) at the altitude of the corresponding wind speed, and mixing ratio (ppb) of trace gases need to be converted into molecular density (molec·$m^{-3}$) beforehand (Supplementary Sect. S4); $W_i$ represents the wind speed in layer i from southwest to northeast. A flux above zero indicates that the air pollutant is transported from southwest to northeast, whereas a flux less than zero means that the transmission direction is from northeast to southwest.

Finally, we calculated the $F_c$ per unit width by summing the $F_i$ multiplied by the height of each layer:

$$F_c = \sum (F_i \times H_i) \tag{4}$$

where $H_i$ is the height of each layer i.

For convenience, the layer with the highest transport flux was defined as the main transport layer (MTL) for the corresponding pollutants. According to the definition and Eq. 3, we know that the MTL is determined by the concentration and wind speed in the corresponding layer. Owing to the large differences in the vertical distributions of various pollutants, their MTLs were bound to have varying characteristics. Some calculation details and error analysis methods are provided in Supplementary Sect. S4.

## 2.6 Ancillary data

We obtained the surface $NO_2$, $PM_{2.5}$, CO, and $O_3$ concentrations from the CNEMCs with a sampling resolution of 1 h (https://quotsoft.net/air/). We validated the MAX-DOAS measurements by comparing the lowest layer results from the MAX-

DOAS observations with the CNEMC data. Using the CO and $O_3$ concentrations, we performed source apportionment of ambient HCHO to identify the contribution ratios of primary and secondary HCHO. Moreover, we depicted a spatiotemporal distribution of $PM_{2.5}$, reflecting surface $PM_{2.5}$, and concentration variations during the transboundary transport process.

The Aerosol Robotic Network (AERONET) is a ground-based aerosol remote sensing observation network jointly established by NASA and LOA-PHOTONS (B.N.Holben et al., 1998). During the measurement period, there were two AERONET sites, Beijing-CAMS (39.933 °N,116.317 °E) and XiangHe (39.754 °N,116.962 °E), adjacent to our MAX-DOAS stations, namely, CAMS (39.95 °N, 116.32 °E) and XH (39.76 °N, 116.98 °E). In this study, we used the Level 1.5 AOD results from these two AERONET sites to validate AODs measured by MAX-DOAS.

We obtained the spatial distributions of $NO_2$ and HCHO from TROPOMI at a spatial resolution of $3.5 \times 7.0$ km (Veefkind et al., 2012), and the spatial distributions of AOD and dust from Himawari-8 with a $0.5 \times 2.0$ km spatial resolution and a 10 min temporal resolution (Bessho et al., 2016). Satellite observations helped identify the pollutant transport phenomena because transport tends to cause large-scale continuous distribution of pollutants that can be detected by satellite measurements.

We simulated the wind speed and direction using the Weather Research and Forecasting Model, version 4.0 (WRF 4.0). See
Supplementary Sect. S5 which details the model and parameter settings. In terms of wind speeds and pollutant mixing ratios in different layers, we calculated transport fluxes at different heights to reflect the dynamic transport processes of various pollutants. In addition, we used wind-field information to reveal the transport direction at different altitudes.

We calculated the 24-h backward trajectories of the air masses using the Hybrid Single-Particle Lagrangian Integrated Trajectory (HYSPLIT) model (Supplementary Sect. S6). In our study, the 24-h backward trajectories were calculated to
investigate the dust origins and pathways that reached the NCP on March 15, 2021.

## 3 Results and discussion

For validation, we compared the surface $NO_2$ concentrations and AECs from MAX-DOAS measurements from January to March 2021 with in situ $NO_2$ and $PM_{2.5}$ collected by the CNEMCs. We calculated the $O_4$ effective optical path as the distance threshold (~ 5 km) to exclude some MAX-DOAS stations from the correlation analysis (Supplementary Sect. S7). Table S1
lists the selection conditions. Furthermore, we filtered the "abnormal values" of MAX-DOAS and in situ measurements before comparison, which was favorable to lessen the effects of occasional extreme conditions and improve the correlation (Supplementary Sect. 8). As displayed in Fig. 2, we found good agreement between MAX-DOAS and in situ data, with Pearson correlation coefficients (R) of 0.752 and 0.74 for aerosol and $NO_2$, respectively. The fine correlation between AOD from MAX-DOAS and AERONET also confirmed the reliability of this measurement, with R reaching 0.941 and 0.816 for CAMS
and XH, respectively (Fig. S3). The aerosol information obtained directly from MAX-DOAS is the AECs. Under most

conditions, aerosol mass concentration is approximately proportional to the extinction coefficient (Charlson, 1969; Robert et al., 1968). However, they are not completely equivalent; relative humidity (RH) influences their correlation (Lv et al., 2017).

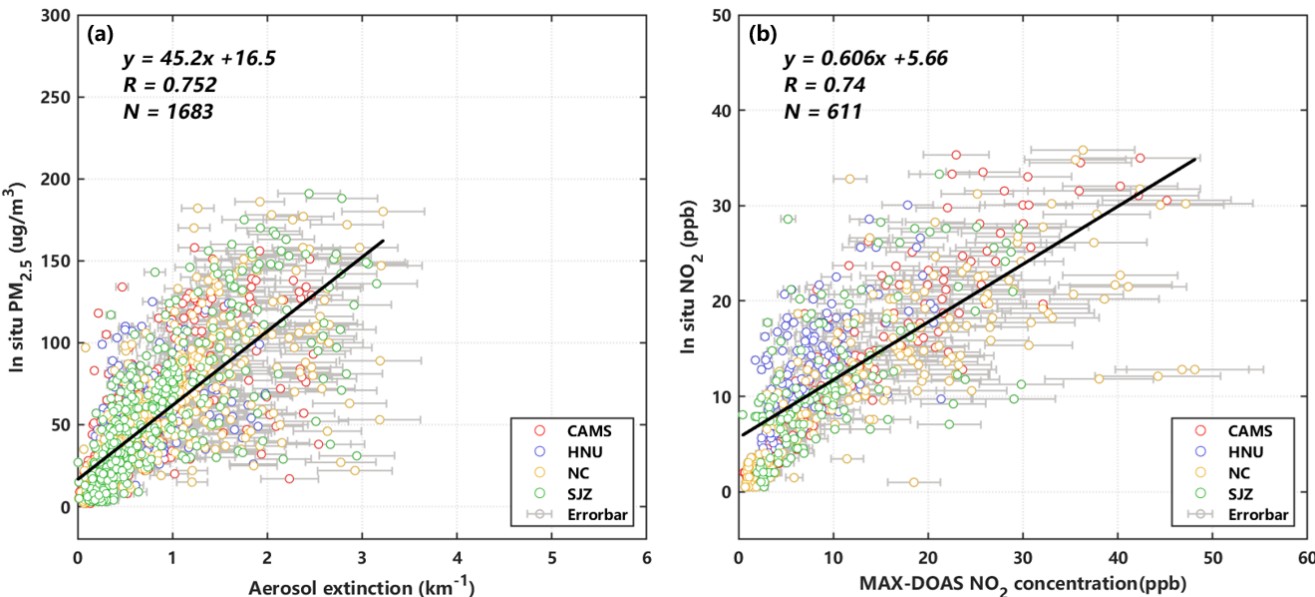

**Figure 2. (a) Correlation analysis of in situ measured PM$_{2.5}$ and surface AECs (0–100 m) retrieved from CAMS, HNU, NC, and SJZ**

**MAX-DOAS stations from January to March, 2021 and (b) their corresponding NO$_2$ comparative results. The black line denotes the linear least-squares fit to the data; R denotes Pearson correlation coefficient; N denotes the number of valid data.**

### 3.1 Dynamic transport processes of NO$_2$, HCHO, and aerosol

Impacted by the semi-basin topography and continental monsoon climate, intra-regional transport in the JJJ region is frequent

and is a dominant factor influencing the environmental air quality of many cities. Based on in situ measurements in the JJJ region, southwest-to-northeast regional transport was found to play a significant role in increasing PM$_{2.5}$ and O$_3$ levels (Ge et al., 2018). In addition, a south-to-north transport belt exists in this region (Ge et al., 2012). Using WRF-Chem simulations, Wu et al. (2017b) successfully evaluated the contributions of regional transport to elevated PM$_{2.5}$ and O$_3$ concentrations in Beijing during summer. Based on vertical LiDAR observations, Xiang et al. (2021) revealed that PM$_{2.5}$ was transported to Beijing via

the southwest pathway. However, it is difficult to fully understand regional transport using model simulations or in situ measurements. Therefore, we combined MAX-DOAS measurements with WRF simulations to describe the regional transport processes of aerosols, NO$_2$, and HCHO more accurately. The TROPOMI results indicated that NO$_2$ was homogeneously distributed between SJZ and WD and that, on February 5, 2021, the HCHO distribution belt connected NC with CAMS (Fig. S4). Figure S5 shows the regional wind information in different layers (0–20, 200–400, 400–600, and 600–800 m), with the

wind direction being mainly southwest–northeast among the four stations (i.e., SJZ, WD, NC, and CAMS). These findings revealed a typical regional transport process along the southwest-northeast pathway.

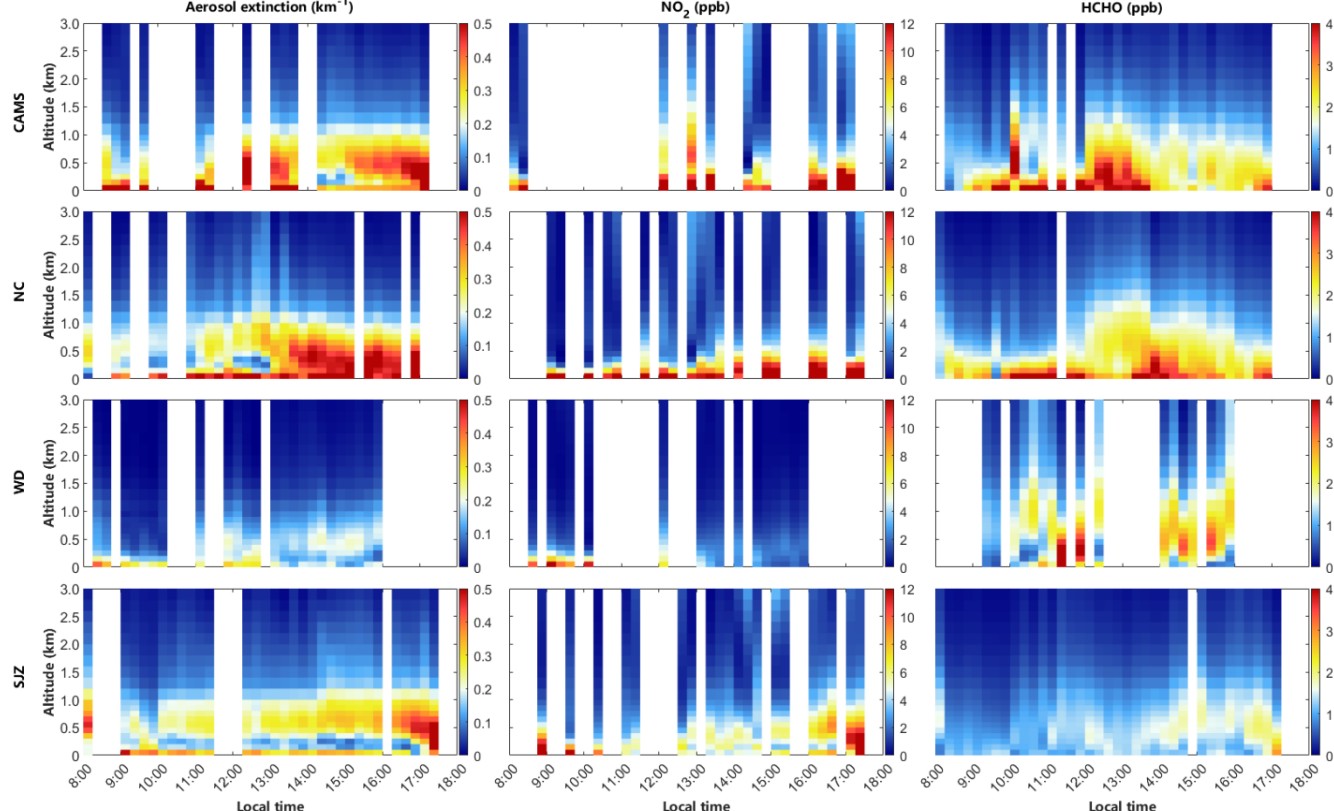

**Figure 3. Vertical profiles of aerosol, NO$_2$, and HCHO at CAMS, NC, WD, and SJZ stations on February 5, 2021. ppb: part per billion.**

Figure 3 presents the temporal variations in the vertical distributions of aerosols, NO$_2$, and HCHO during this regional transport. At the CAMS and NC stations, the aerosol, NO$_2$, and HCHO concentrations were consistently high near the surface, primarily because of the heavy traffic flow and dense factory emissions in Beijing (Zhang et al., 2016; Li et al., 2017). Previous studies have suggested that urban air pollution in Beijing is dominated by a combination of coal burning and vehicle emissions, which results in severe particulate pollution (Wang and Hao, 2012; Wu et al., 2011). At SJZ, the NO$_2$ concentration was high (~ 12 ppb) in the morning and late afternoon, whereas the concentration was lowest (~ 6 ppb) near noon, which is explained by the morning and evening rush hour. Comparatively, the overall AEC and NO$_2$ levels were relatively low at the WD station, whereas a continuous high-value HCHO distribution (> 2 ppb) occurred at 0-1500 m between 11:00 and 16:00. This occurred because the WD station is located in a farm field with less traffic flow and high vegetation coverage; therefore, large amounts of HCHO are directly emitted by biogenic sources and secondarily produced by natural and anthropogenic volatile organic compound (VOC) photolysis (Wang et al., 2016; Wu et al., 2017a).

Nevertheless, Fig. 3 cannot reveal the exact layers in which the main transport phenomena occur. For instance, at the CAMS station, the AEC at the surface and upper layers both reached ~ 0.5 km$^{-1}$ around noon, making it difficult to determine the layer

in which transport was more obvious. To further demonstrate the dynamic transport process of different pollutants, we

calculated the hourly $F_i$ and $F_c$, and defined the MTL. As shown in Fig. 4, a positive $F_i$ indicates that all transport flux

projections in the southwest-northeast pathway are from southwest to northeast at the four stations. The MTLs of aerosols,

HCHO, and NO$_2$ exhibited different spatiotemporal characteristics. Although surface and high-altitude (400–800 m) AECs

both remained at a relatively high level (> 0.3 km$^{-1}$) at CAMS during 12:00-17:00 (Fig. 3), there was a large discrepancy

between their corresponding $F_i$ values (Fig. 4). The aerosol near-surface $F_i$ was ~ 1 km$^{-1}$ m s$^{-1}$ after 12:00, while $F_i$ in layers

of 400–800 m all exceeded 1.2 km$^{-1}$ m s$^{-1}$, and even reached ~ 2 km$^{-1}$ m s$^{-1}$ around 12:00. At the SJZ station, the AECs at

surface and 300–1000 m layer mostly ranged from 0.3 to 0.4 km$^{-1}$, especially after 10:00 (Fig. 3). However, the MTLs of

aerosols were mostly at 400–800 m throughout the day, with many transport fluxes in those layers even reaching ~ 2 km$^{-1}$ m s$^{-1}$

$^{-1}$ (Fig. 4). At the WD station, the highest $F_i$ also tended to occur at high layers (400–800 m), with maximum $F_i$ exceeding

1.7 km$^{-1}$ m s$^{-1}$ at 400–500 m at 15:00. This suggested that aerosol transport occurred mainly in the upper layers. In the late

afternoon, aerosols gradually accumulated towards the surface, and triggered a variation in the distribution of $F_i$. After 16:00,

the shift in the high-AEC air mass caused the transport fluxes in the lower layers (100–200 m) to increase to > 1.1 and ~ 2 km$^{-1}$

$^{-1}$ m s$^{-1}$ for the CAMS and SJZ stations, respectively. Surface aerosol accumulation is closely linked to the collapse of the

mixing layer and formation of a stable nocturnal boundary layer (Ding et al., 2008; Ran et al., 2016). Remarkably, high-altitude

aerosol air masses began to mix with near-surface aerosols after 14:00 at the NC station (Fig. 3), triggering a variation in the

MTL (Fig. 4). This might be explained by enhanced vertical mixing due to the heating of the surface during the course of the

day (Castellanos et al., 2011; Wang et al., 2019). Generally, the MTL of aerosols was situated at 400–800 m during the daytime,

where variations in the boundary layer and increased vertical mixing can influence the MTL. In contrast to aerosols, we found

that a high-value NO$_2$ $F_i$ frequently occurred in the 0–400 m layer. Except that $F_i$ reached the highest level of ~ 1.8 $\times 10^{18}$

molec·m$^{-2}$·s$^{-1}$ in the 400–600 m layer at 16:00 at the SJZ station, the other highest $F_i$ all occurred below 400 m at any station

and at any time. This indicated that the MTL of NO$_2$ was 0–400 m. Near-surface NO$_2$ emission sources (e.g., vehicle and

factory emissions) might be the main reason for this phenomenon. Compared with aerosols and NO$_2$, we found that high-value

HCHO $F_i$ extended to higher altitudes. Taking CAMS as an example, we found the strongest HCHO $F_i$ constantly emerging

at 1000–1200 m from 8:00 to 13:00, and averaging 2.51 $\times 10^{17}$ molec m$^{-2}$ s$^{-1}$. During the same period, surface HCHO $F_i$ only

averaged 1.72 $\times 10^{17}$ molec m$^{-2}$ s$^{-1}$. However, at the CAMS station, the surface HCHO concentration was much higher than

that of the 1000–1200 m layer between 8:00 and 13:00 (Fig. 3), proving that high-altitude transport contributed more to overall

HCHO transport. After 10:00, we found that the highest HCHO $F_i$ gradually increased from ~ 3.5 $\times 10^{17}$ to ~ 4.5 $\times 10^{17}$

molec m$^{-2}$ s$^{-1}$ at WD, with the MTL of HCHO ranging from 400 to 1000 m. At station SJZ, the strongest HCHO $F_i$ increased

from ~ 2.6 $\times 10^{17}$ to ~ 4.5 $\times 10^{17}$ molec m$^{-2}$ s$^{-1}$ during 11:00–17:00, with the highest transport fluxes occurring mostly at 400–

800 m. These findings indicated that the MTL of HCHO was mainly 400–1200 m. The sharp variation in the MTL at the NC

station might be caused by atmospheric vertical mixing (Castellanos et al., 2011; Wang et al., 2019). As shown in Fig. 3, high HCHO concentrations tend to appear at higher altitudes than those of aerosols and $NO_2$. A possible explanation is that the precursor compounds of HCHO are transported to higher layers and converted into HCHO through photochemical reactions, resulting in elevated HCHO concentrations at higher altitudes (Kumar et al., 2020). Furthermore, strong high-altitude winds were more conducive to HCHO transport (Fig. S5), which further increased the corresponding transport flux. Notably, HCHO $F_i$ was enhanced around noon because the increased solar radiation promotes the secondary generation of HCHO. Long-term observations have revealed that secondary HCHO formation through VOCs photolysis plays a significant role in Beijing (Liu et al., 2020; Zhu et al., 2018).

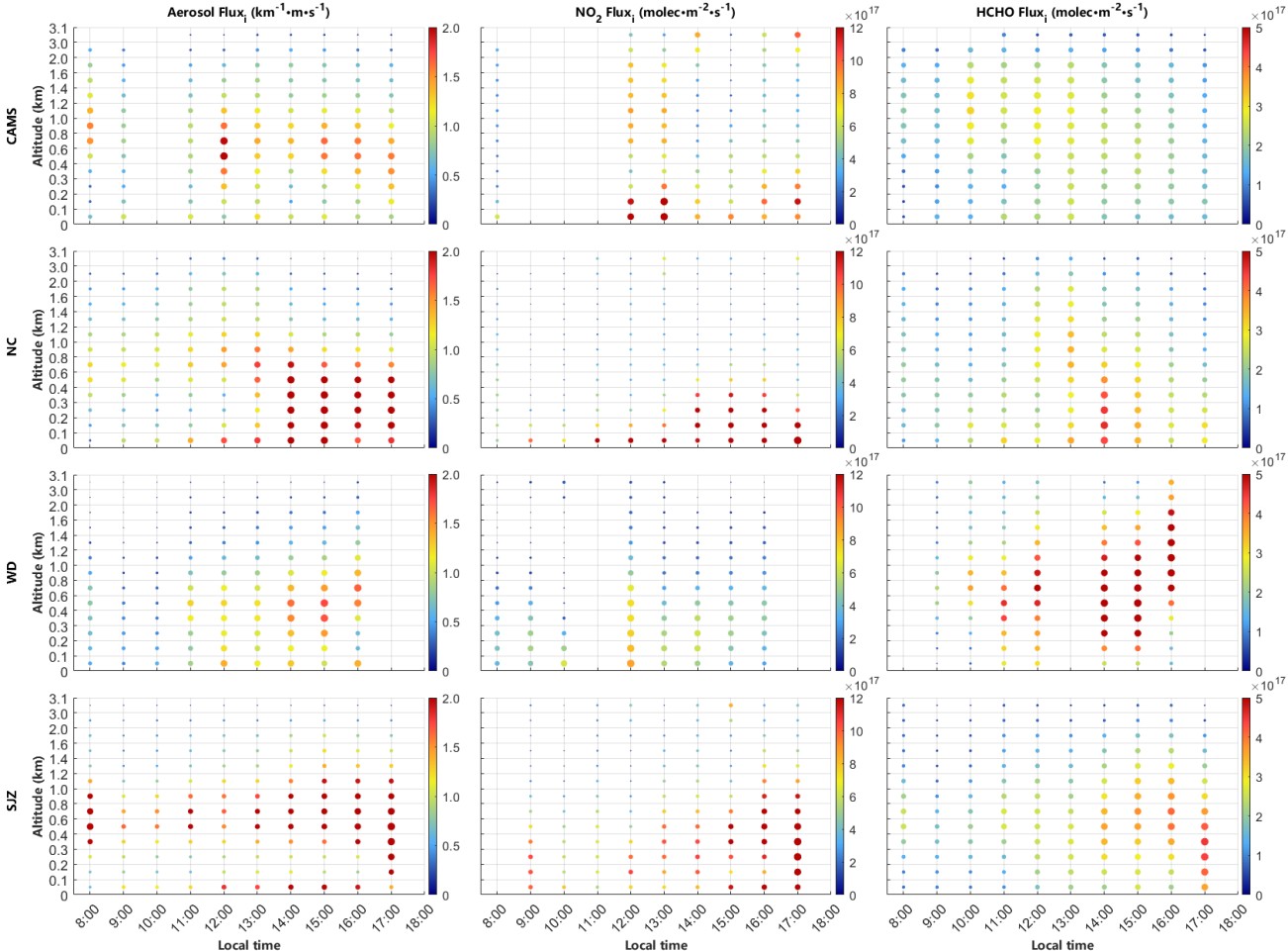

**Figure 4. Transport flux at different altitudes ($Flux_i$) at CAMS, NC, WD, and SJZ stations on February 5, 2021.**

In addition, we discovered a wide discrepancy in the $F_c$ among stations for various pollutants (Fig. 5). The average aerosol $F_c$ decreased in the following order: SJZ ($3.21 \times 10^3$ km$^{-1}$ m$^3$ s$^{-1}$) > NC ($2.69 \times 10^3$ km$^{-1}$ m$^3$ s$^{-1}$) > CAMS ($2.43 \times 10^3$ km$^{-1}$ m$^3$ s$^{-1}$) > WD ($1.42 \times 10^3$ km$^{-1}$ m$^3$ s$^{-1}$). For $NO_2$ transport, the average $F_c$ values at SJZ ($1.56 \times 10^{21}$ molec·s$^{-1}$), NC ($1.10 \times 10^{21}$ molec·s$^{-1}$), and CAMS ($1.58 \times 10^{21}$ molec s$^{-1}$) were substantially higher than those at WD ($5.57 \times 10^{20}$ molec s$^{-1}$). Conversely, the average $F_c$ of HCHO was the highest in WD ($8.82 \times 10^{20}$ molec·s$^{-1}$), whereas the $F_c$ values in SJZ, NC, and

CAMS were $4.81 \times 10^{20}$, $5.16 \times 10^{20}$, and $5.12 \times 10^{20}$ molec·s$^{-1}$, respectively. In terms of the relative locations of stations (Fig. 1) and the $F_c$ results, we considered that SJZ was an important source of transported aerosol and NO$_2$, and WD was one of the main HCHO sources during this regional transport, which largely affected the air quality of cities along the southwest-northeast transport pathway. The corresponding error distributions of $F_i$ and $F_c$ were provided in Figs. S6 and S7.

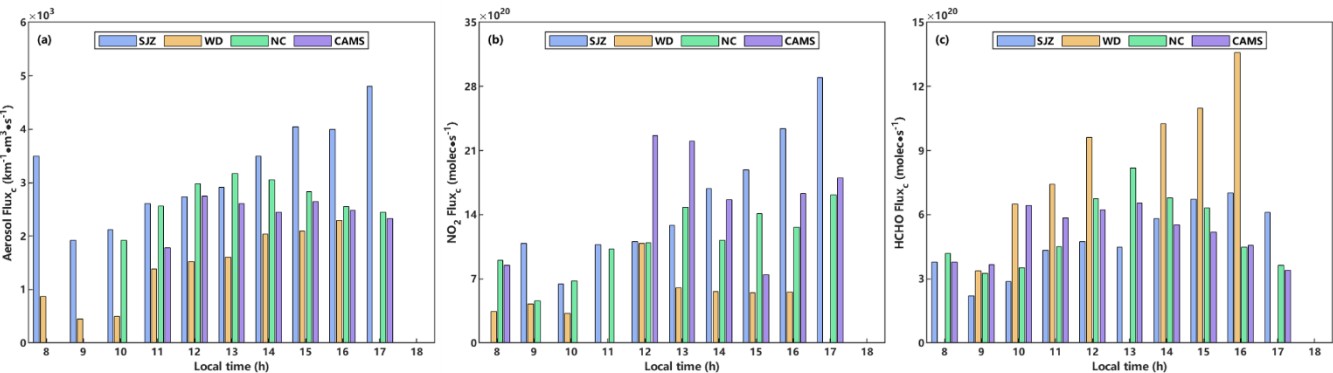

**Figure 5.** Diurnal variation of $F_c$ at SJZ, WD, NC, and CAMS stations for (a) AECs, (b) NO$_2$, and (c) HCHO.

## 3.2 Effects of dust transport on regional air pollution

The Himawari-8 satellite observations revealed that a dust storm occurred in northern China on March 15, 2021 (Fig. S8), with the NCP being one of the most severely affected. Combined with the 24-h backward trajectories (Fig. S9), we found that the dust storm originated in Mongolia and its major transport pathway was Mongolia–Inner Mongolia–NCP. According to the selection standards described in Supplementary Sect. S9, we confirmed that March 15 was a dusty day; we chose March 6 and 22 as comparison benchmarks because they were the nearest clean days before and after the dust storm, respectively.

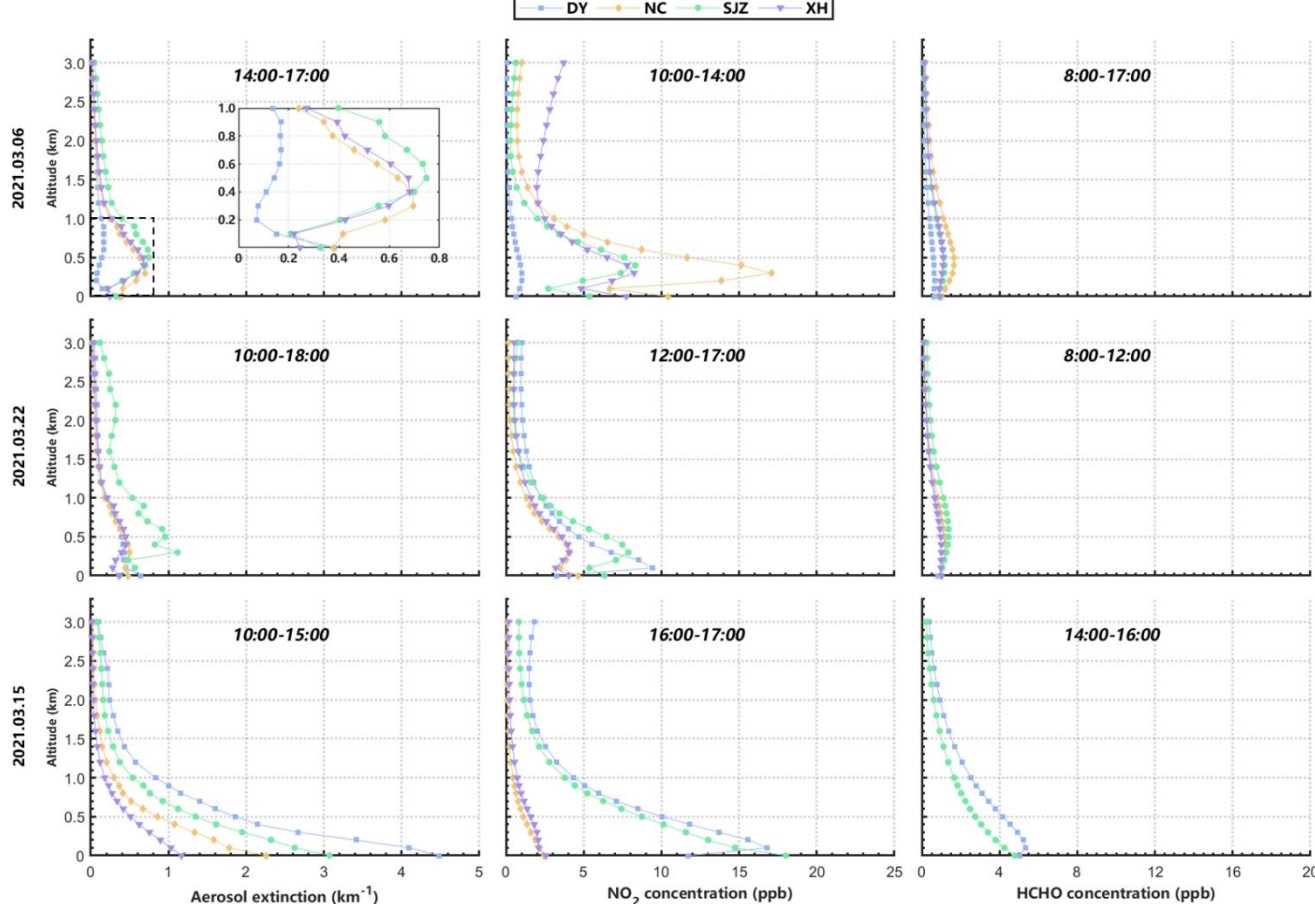

**Figure 6. The daily averaged vertical profiles of AECs (left), NO₂ (middle), and HCHO (right) at DY, NC, SJZ, and XH stations during two clean days (March 6 and 22, 2021) and one dusty day (March 15, 2021). The upper annotation in each subplot represents their corresponding time periods, during which each station generated the greatest amount of data as possible.**

Dust can suppress dissipation and aggravate the local pollution accumulation. On the dust storm day, the AEC and HCHO concentrations substantially increased, especially near the surface (Fig. S10 and S12). Meanwhile, NO₂ concentrations also increased significantly in SJZ and Dongying (DY) (Fig. S11). As shown in Fig. 6, the vertical profiles at all four stations displayed peaks in the high layers on clean days. According to Liu et al. (2021), vertical profile shapes can be used to perform

an overall evaluation of pollutant sources and meteorological conditions in a certain area. In the JJJ region, we selected three stations (i.e., NC, SJZ, and XH) and found that high-altitude peaks occurred at 300–500 m for aerosols, NO₂, and HCHO (Table S3-S5). For example, the AEC vertical distribution at SJZ displayed a high peak of 0.75 km$^{-1}$ at 500 m on March 6 (Table S3). At the NC station, the AEC vertical distribution exhibited the only peak (0.70 km$^{-1}$) at 300 m. This may be explained by the prevalent regional transport, which strongly influences the air quality in the JJJ region (Ge et al., 2018; Wu et al., 2017;

Xiang et al., 2021). As discussed in Section 3.1, high-altitude transport phenomena trigger high-values of pollutant distribution in the high layers. The surface peaks on clean days were possibly caused by dense traffic and factory emissions in the JJJ region (Qi et al., 2017; Zhu et al., 2018; Yang et al., 2018; Han et al., 2020). In contrast to the JJJ region, the DY station is

situated in a rural area surrounded by open oil fields and is adjacent to the Bohai Sea (Guo et al., 2010). The transport of sea salt is a significant source of local aerosols (Kong et al., 2014), which might be the main reason for the high-altitude AEC peaks occurring at 300–900 m (Table S3). In addition, the occurrence of high-altitude $NO_2$ and HCHO peaks may be attributed to the surrounding high-elevation point emission sources (e.g., chemical plants) (Kong et al., 2012). During dusty periods, aerosol, $NO_2$, and HCHO concentrations notably increased, particularly near the surface, at most stations (Fig. 6). The high-layer peaks dropped to lower altitudes and even disappeared (Tables S3-S5). For example, on the dusty day, we found that high-altitude peaks disappeared and the only peak emerged at the surface for aerosol, $NO_2$, and HCHO vertical profiles at the NC, XH, and SJZ stations (Tables S3-S5). Meanwhile, the $NO_2$ and HCHO concentration peaks both dropped to the 100 m layer at the DY station (Tables S4, S5). These changes might trigger variations in the vertical profile shapes and convert many vertical profile shapes (e.g., AEC vertical profiles at all stations) into an exponential shape (Fig. 6). This is because elevated dust concentrations weaken turbulence and decrease PBL heights on the dusty day, mostly through surface cooling and upper PBL heating (Mccormick and Ludwig, 1967; Li et al., 2017b; Mitchell and Jr., 1971). Unfavorable meteorological conditions not only impede pollutant dissipation and transport, but also favor the accumulation of locally produced pollutants (including direct emissions and secondary production). Moreover, some accumulating components (e.g., $NO_2$, $SO_2$, and VOC) are important precursors of aerosols, providing favorable conditions for secondary aerosol formation (Behera and Sharma, 2011; Volkamer et al., 2006; Huang et al., 2014).

In addition to aggravating pollutant accumulation, transported dust can affect the environment and pollutant concentrations in other ways. To quantitatively demonstrate the impacts of dust on various pollutants, we introduced growth rate in the comparative analysis (Supplementary Sect. S9). For convenience, we defined the comparison of the results of March 6 and 15, 2021, as precomparison (PRE), and we defined the comparison between March 15 and 22, 2021, as postcomparison (POST). As shown in Fig. 7A, the AEC noticeably increases at all stations on the dusty day, especially below 0.5 km. To quantitatively evaluate the impacts of increased dust and aerosols on light intensity, we averaged the optical signal intensities received at each channel of the spectrometer as the light intensity of each station. By subtracting the light intensity on clean days from that on the dusty day, we found that light intensity was substantially reduced at each station on the dusty day (Fig. S13). This is because enhanced aerosol concentrations, along with large amounts of dust, aggravate light attenuation and weaken light intensity. At certain wavelengths (e.g., 360 nm), aerosol extinction contributes more to light attenuation than dust (Wang et al. 2020). As described above, the advent of dust results in unfavorable meteorological conditions (e.g., decreased PBL height and more stable PBL) and enhances local pollutant accumulation, which boosts aerosol increase in the lower layer. Moreover, such meteorological conditions are always accompanied by higher levels of RH in the lower PBL (Huang et al., 2020), creating good conditions for enhanced secondary production of aerosols through aqueous-phase and heterogeneous chemical reactions

(Ravishankara, 1997; Mcmurry and Wilson, 1983). These two factors could be the main reasons for the enhanced aerosol concentrations on the dusty day.

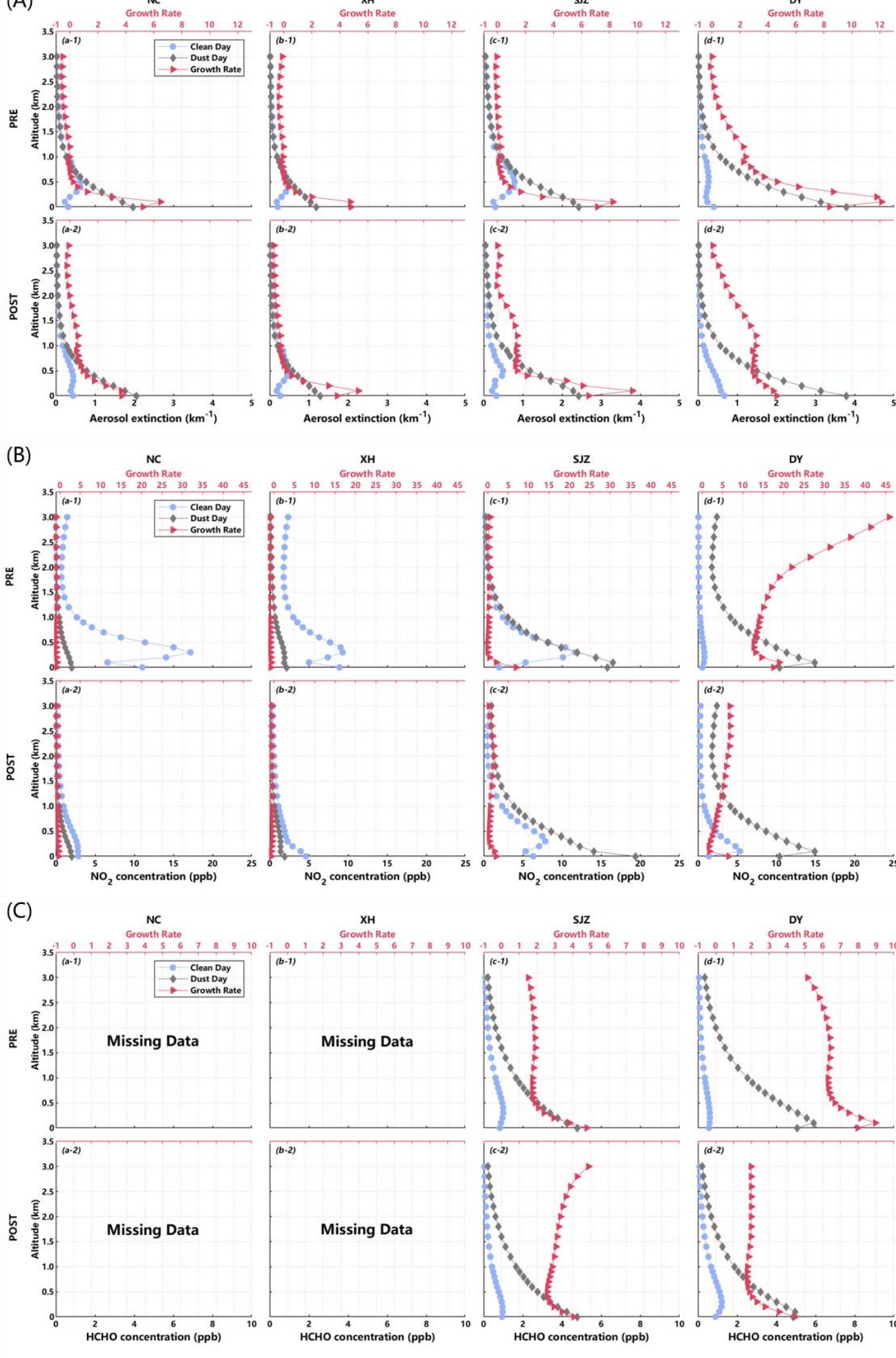

**Figure 7. The growth rates of (A) AEC, (B) NO$_2$ and (C) HCHO at different altitudes at (a) NC, (b) XH, (c) SJZ, and (d) DY stations. PRE indicates precomparison results between March 6 and 15, 2021; POST indicates postcomparison results between March 22 and 15, 2021.**

In contrast to aerosols, we observed large differences in NO$_2$ growth rates (Fig. 7B). At SJZ and DY, NO$_2$ concentrations exhibited a substantially increasing trend. The surface growth rates at the SJZ and DY stations were 6.97 and 17.50 in PRE, and 2.06 and 6.50 in POST, respectively (Fig. 7B (c-1, c-2, d-1, d-2)). In contrast, we observed decreased NO$_2$ concentrations at almost every height at stations NC and XH. The near-surface growth rates at NC and XH were -0.81 and -0.76 in the PRE, and -0.30 and -0.59 in the POST, respectively (Fig. 7B (a-1, a-2, b-1, b-2)). This indicated that dust and aerosols have different

effects on the trace gas concentration. On the one hand, they limited the received radiation (Fig. S13), thereby preventing NO$_2$ photolysis, which prolongs its lifetime and favors its accumulation (Chang and Allen, 2006). Moreover, they also had the effect of reducing turbulence and inhibiting dissipation, eventually intensifying surface NO$_2$ accumulation. On the other hand, aerosols and dust particles can act as surfaces for heterogeneous NO$_2$ destruction processes, leading to a decrease in NO$_2$. On the dusty day, large amounts of dust and aerosols provide surface areas for heterogeneous reactions and deposition of different

trace gases. Heterogeneous reactions on dust and aerosol surfaces can result in a general decrease in the atmospheric concentrations of trace gases, such as O$_3$, nitrogen oxides, and hydrogen oxides (Kumar et al., 2014; Bauer, 2004; Dentener et al., 1996). Among these reactions, the conversion of NO$_x$ (NO + NO$_2$) to HONO plays an important role in NO$_2$ removal (Stemmler et al., 2006; Ndour et al., 2008; George et al., 2005). Under high AEC and RH conditions, the conversion of NO$_2$ to HONO is further promoted (Xing et al., 2021b). With March 6 and 22 as the comparison benchmarks, we found that the

surface growth rates in HONO concentration were 2.70 and 3.52 at the NC station, respectively, which validates our hypothesis (Fig. S14).

Owing to amplified pollutant accumulation, increased HCHO concentrations were recorded at both stations SJZ and DY (Fig. 7C). In the PRE, the surface growth rate was 4.80 at SJZ (Fig. 7C (c-1)), whereas a maximum relative increase of 8.97 occurred at 100 m at DY (Fig. 7C (d-1)). In the POST, the surface concentration increasing ratio was 4.06 at SJZ, and reached the

highest level (4.46) at the surface at the DY station (Fig. 7C (c-2, d-2)). In addition, we believed that reduced solar radiation also influenced the HCHO concentration. Based on the results of the linear model, we analyzed the source apportionment of ambient HCHO on measurements from the SJZ station (Supplementary Sect. S10). Primary HCHO levels played a dominant role in ambient HCHO levels, with the total contribution ratio of primary and background HCHO exceeding 75% in March (Fig. S15b-c). The reduced solar radiation weakened the photolysis of the primary and background HCHO, favoring the overall

increase in HCHO levels in SJZ. Owing to the long distance between the MAX-DOAS station and the CNEMC at DY (Table S1), the CO and O$_3$ concentrations recorded by the CNEMC did not represent their corresponding concentrations around the

MAX-DOAS station. Therefore, we could not separate the source contributions of the measured HCHO concentrations using the method described above. Previous studies have suggested that VOCs play a dominant role in the secondary formation of HCHO at the DY station (Chen et al., 2022; Chen et al., 2020). Thus, we assumed that the considerable increase in HCHO levels at station DY was closely related to VOC accumulation.

The comparison result between the dusty day and two clean days makes it possible to better understand the impacts of dust storm on local environment. However, there remain some uncertainties in this discussion. Although we selected the closest clean days to lessen the effects of some factors (e.g., climate and temperature) on comparison, the uncertainties caused by other meteorological parameters (e.g., wind speed and directions) were unknown to us, since we did not make sure these parameters were nearly the same on these three days. Therefore, this comparison analysis is based on the assumption that there is little difference between meteorological parameters on various days or the effects caused by different meteorological parameters are negligible. Besides, a dust storm would trigger changes at the vertical sensitivity of MAX-DOAS measurements, which might influence profile shape. These impact factors are difficult to control in observations, and modelling correction may be a good solution.

### 3.3 Spatiotemporal characteristics of aerosol during transboundary transport

Back-and-forth transboundary long-range transport between the NCP and YRD is common, especially during winter (Huang et al., 2020; Petaja et al., 2016). During the transport process, the aerosol–PBL interaction can amplify the overall haze pollution and deteriorate the air quality of these two regions (Petaja et al., 2016; Ding et al., 2016; Huang et al., 2014; Huang et al., 2018b). Based on model simulations, Huang et al. (2020) elaborated on this transport process and the haze-amplifying mechanism in three stages. First, air pollutants from the YRD are transported to the upper PBL over the NCP and substantially affect PBL dynamics. Subsequently, under the influence of aerosol-PBL interaction, local pollutant accumulation and secondary production of aerosols are enhanced, causing severe pollution in the NCP. Finally, strong weather patterns (e.g., cold fronts), can dissipate low-PBL pollutants in the NCP and transport them over long distances back to the YRD. Many model simulations have suggested that the mechanism of aerosol–PBL interaction amplifies the overall haze pollution during the transport process (Petaja et al., 2016; Ding et al., 2016; Huang et al., 2014; Huang et al., 2018b). Using MAX-DOAS measurements, we investigated the spatiotemporal variation in aerosols along the transport pathway, and validated the haze-amplifying mechanism of this transboundary transport.

The Himawari-8 satellite observations revealed a substantial increase in aerosol concentrations within the NCP from January 18 to 20, 2021, with an overall AOD overpassing 0.9 (Fig. S16). Subsequently, high-concentration aerosol air masses assumed a southward movement tendency, gradually leaving the NCP and covering the YRD on January 21–22, 2021. We attributed this phenomenon to the back-and-forth transport of aerosols between these two regions, which we validated using wind-field

simulations. The wind field results indicated that the wind blew towards the East China Sea at every altitude on January 18, 2021 (Fig. 8). A south-to-north transport belt firstly formed in the upper layers (500–1500 m) on January 19 and lasted for nearly two days. Around 12:00 on January 21, the wind direction began to change, and the north-to-south transport trend strengthened in the 0–1000 m layer on January 22. The diurnal variation of wind fields in different layers on January 18–22, 2021, were provided in Figs. S17-S21. In terms of overall transport direction, we classified the MAX-DOAS monitoring results into four periods: West-to-East, YRD-to-NCP, Transformation, and NCP-to-YRD, to further explore their vertical characteristics during the transport process (Fig. 9).

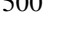

**Figure 8. Wind fields in layers of surface, 500, 800, 1000 and 1500 m at 12:00 from January 18 to 22, 2021.**

On January 18, 2021, no direct long-range transport occurred between the NCP and YRD (Fig. 8 and Fig. S17). These two regions both had acceptable air quality, with the maximum AEC at the four stations being less than 0.88 km$^{-1}$ (Fig. 8) and the overall spatiotemporal average PM$_{2.5}$ concentration approximately 43.71 μg/m$^3$ (Fig. S22a). The wind simulation indicated that south-to-north transport initially took shape in layers of 500–1500 m on January 19 (Fig. 8 and Fig. S18), which we defined as the start of the YRD-to-NCP transport period. During this period, the overall AECs at all stations began to increase in varying degrees. We noted that a continuous high-AEC distribution occurred at XH in the 100–1400 m layer, at DY in the 200–1000 m layer, and at NB in the 100–700 m layer (Fig. 9). The maximum AECs reached 6.10, 1.41, and 0.96 km$^{-1}$ for XH, DY, and NB, respectively. According to the wind simulation for January 19–21, the YRD-to-NCP transport lasted until 12:00 on January 21 (Fig. 8 and Fig. S18-S20). During this period, large amounts of aerosol from the YRD were transported to the upper layers (500–1500 m) of the NCP. In addition, secondary particle formation intensified because the transport of warm and humid air masses favors aqueous and heterogeneous reactions (Huang et al., 2014). These factors jointly led to a sharp increase in AECs in the high layers at stations XH, DY, and NB (Fig. 9). In contrast, the increase in the near-surface AEC was slower than that in higher layers. On January 19, for instance, the surface AECs were mostly less than 0.6 km$^{-1}$ from 10:00 to 16:00 in XH, while surface peak AECs in the morning and late afternoon could be explained by the diurnal variation in PBL height (Ding et al., 2008; Ran et al., 2016). At the DY station, the average surface AEC only increased from 0.61 km$^{-1}$ on January 18 to 0.62 km$^{-1}$ on January 19. The reason is that surface transport was driven mainly by the east wind on January 19 (Fig. 8), resulting in PM$_{2.5}$ concentrations at many western CNEMCs exceeding 80 μg/m$^3$ (Fig. S22b). From January 20 to 21, 2021, the surface wind converted into a south wind, but became so weak that near-surface transport contributed little to the NCP (Fig. 8 and Fig. S19-S20). However, we continued to observe a substantial increase in AEC at ground level on January 20–21, 2021 (Fig. 9). At station DY, for example, the average surface AEC increased from 0.61 km$^{-1}$ on January 18 to 1.03 km$^{-1}$ on January 20, which was a 68.9% growth rate. A possible reason for this was the strong dome effect caused by high-layer aerosols. As a result of the aerosol–PBL interaction, PBL height decreases while temperature and humidity increase in the lower PBL, which favors pollution accumulation and secondary aerosol production (Bharali et al., 2019; Huang et al., 2020; Petaja et al., 2016). Generally, this YRD-to-NCP transport intensifies local pollution in the NCP region, causing a substantial increase in aerosol concentrations on January 18–20 (Fig. S16). Around 12:00 on January 21, the wind direction went through a half-day transformation period, changing from southerly to westerly (Fig. S20). The continuous high-AEC distribution belt in NB was interrupted at 12:30–13:30 and 14:15–16:00 (Fig. 9), possibly owing to western clean-air injection. The northerly wind finally formed on January 22, creating the NCP-to-YRD transport belt (Fig. 8). In contrast to YRD-to-NCP transport, NCP-to-YRD transport mainly occurred at low altitudes (0–1000 m), with surface wind speeds rising to > 4 m s$^{-1}$ (Fig. 8 and Fig. S21). Influenced by strong cold fronts, large amounts of aerosols in the NCP began to disperse and gradually covered the

YRD (Fig. S16), increasing the average surface AEC in NB by 183.33% from 1.56 to 4.42 km$^{-1}$ (Fig. 9). Furthermore, with

11:00 as a dividing line, high-AEC air masses (average AEC of 0–1.6 km layer: 2.60 km$^{-1}$) abruptly vanished at the DY station,

highlighting the effect of quick dispersion driven by cold fronts. The weak dispersion of aerosols in XH may have been affected

by the anticyclones in the JJJ region (Fig. S21). For cities in the area where the NCP and YRD overlap, their location determines

that they suffer a longer dome effect period because the two transport processes (YRD-to-NCP and NCP-to-YRD) pass through

these areas, producing a constant increase in pollution levels in the shallow PBL. For example, from January 18 to 22, the

average AECs in the 0–1 km layer in HNU were 0.35, 0.48, 0.75, 0.76, and 0.85 km$^{-1}$, respectively, showing a continuously

increasing tendency (Fig. 9). Furthermore, on January 22, extreme PM$_{2.5}$ values (> 200 μg/m$^3$) were mostly concentrated in

the overlapping zone (Fig. S22e). Generally, we found that transboundary long-range transport amplified haze pollution within

the NCP and YRD (Fig. 9 and Fig. S22a-e), which agrees well with previous WRF-Chem simulation results (Huang et al.,

2020). This MAX-DOAS measurements accurately demonstrated the spatiotemporal characteristics of aerosols during

transboundary transport. Furthermore, from a practical perspective, we verified the haze-amplifying mechanism (Huang et al.,

2020).

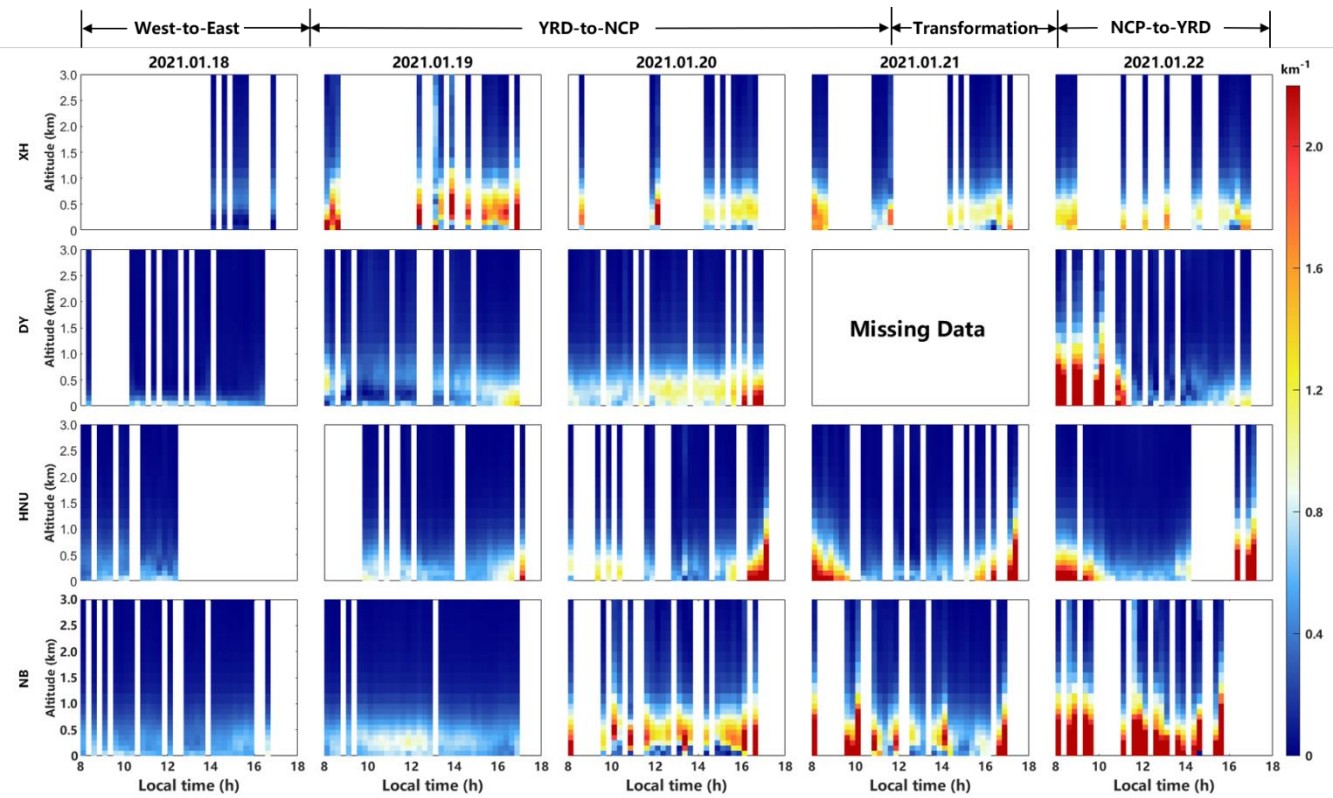

**Figure 9. Temporal and vertical variations in aerosol distribution at XH, DY, HNU, and NB stations from January 18 to 22, 2021.**

**4 Conclusions**

On February 5, 2021, southwest–northeast regional transport of pollutants occurred within the JJJ region. By calculating the $F_i$ and $F_c$ at each station, we demonstrated the dynamic evolution of the MTL for aerosols, $NO_2$, and HCHO. The MTL of aerosols was situated at 400–800 m during the daytime. The variation in the PBL and enhanced vertical mixing could trigger a decreasing trend in the MTL. Nevertheless, the MTL of $NO_2$ was below 400 m, possibly owing to near-surface traffic or factory emissions. The HCHO MTL was 400–1200 m, and extended to higher altitudes than aerosols and $NO_2$. Higher HCHO concentrations and wind speeds at elevated altitudes jointly resulted in much stronger transport fluxes at high altitudes. With respect to the $F_c$ comparison result, we found that the outflows of aerosols and $NO_2$ assumed high levels at the SJZ station. In contrast, the HCHO outflow in WD far exceeded that at other stations. Based on relative station locations and the $F_c$ comparison results, we believed that SJZ and WD were two predominant pollutant sources during this regional transport, exerting a significant impact on cities along the southwest-northeast transport pathway.

A severe dust storm occurred in the NCP along the Mongolia–Inner Mongolia–NCP pathway on March 15, 2021. By comparing the results of the dusty day (i.e., March 15, 2021) and clean days (i.e., March 6 and 22, 2021), we found that high-altitude concentration peaks dropped to a lower layer and even disappeared on the dusty day. We attributed this result to dust being able to suppress dissipation, weaken pollutant transport, and intensify local pollution accumulation. In addition to aggravating pollutant accumulation, transported dust could also affect the environment and pollutant concentrations in other ways. High AEC growth rates were observed at all stations. Large amounts of dust and aerosols intensified light attenuation and weakened light intensity. Notably, dust and aerosols had different effects on $NO_2$ concentration. At stations SJZ and DY, the $NO_2$ concentrations assumed a high growth rate because dust and aerosols limited the received solar radiation and inhibited $NO_2$ photolysis, which favors its accumulation. In addition, dust reduces turbulence and inhibits dissipation, eventually aggravating the surface $NO_2$ accumulation. In contrast, we observed a remarkable decrease in $NO_2$ levels at stations XH and NC. The increase in HONO levels confirmed that heterogeneous reactions on dust and aerosol surfaces played a critical role in the decreases in $NO_2$ levels, with $NO_x$-to-HONO conversion being one of the main removal mechanisms. The source apportionment analysis of ambient HCHO levels revealed that the total contributions of primary and background HCHO exceeded 75% in SJZ, with primary sources playing a dominant role. The reduced solar radiation weakens the photolysis of the primary and background HCHO levels, favoring HCHO level increases in SJZ. The substantial increase in HCHO levels at the DY station may be associated with VOC accumulation.

According to the WRF simulation, we found that the YRD-to-NCP transport mainly occurred in the upper layers (500–1500 m). High-altitude transport triggered a substantially enhanced AEC above the NCP, which might be attributed to direct aerosol injection and secondary particle formation. The increase in near-surface AEC was probably due to the dome effect caused by the aerosol-PBL interaction. Subsequently, NCP-to-YRD transport, situated at 0–1000 m, dispersed the haze over the NCP

and transferred low-level aerosols to the YRD, causing a considerable increase in the surface AEC at station NB. In the overlapping zone between the NCP and YRD, the AEC assumed a continuously increasing tendency in the shallow PBL during the entire transport process, possibly owing to the longer exposure to the dome effect. Generally, this transboundary long-range transport amplified the air pollution in these two regions. Based on practical observations, we investigated the spatiotemporal variation in aerosols and validated the haze-amplifying mechanism of transboundary transport.

In summary, accurate quantification of the vertical distribution of pollutants in the air is a key requirement for understanding the pollutant transport process. Using the MAX-DOAS network, we successfully analyzed three typical transport types (regional, dust, and transboundary long-range transport), emphasizing the unique advantages of the network in monitoring pollutant transport. We believe that our findings can provide the public with a thorough understanding of pollutant transport phenomena and can serve as a reference for designing collaborative air pollution control strategies.

**Acknowledgements**

This research was supported by the National Natural Science Foundation of China (U21A2027); the Anhui Provincial Natural Science Foundation (2108085QD180); the National Natural Science Foundation of China (41977184); the Presidential Foundation of Hefei Institutes of Physical Science, Chinese Academy of Sciences (YZJJ2021QN06); the CAE strategic research and consulting project (No.2021-JZ-05); the Strategic Priority Research Program of the Chinese Academy of Sciences (No. XDA23020301); the Key Research and Development Project of Anhui Province (202104i07020002), the Major Projects of High Resolution Earth Observation Systems of National Science and Technology (05-Y30B01-9001- 19/20-3), the Youth Innovation Promotion Association of CAS (2021443), and the Young Talent Project of the Center for Excellence in Regional Atmospheric Environment, CAS (CERAE202004). We would like to express our gratitude to Fusheng Mou and Wensu Li of Huaibei Normal University for their assistance. We thank Pucai Wang, Xiangao Xia and Brent Holben for their efforts in establishing and maintaining the AERONET Xianghe and Beijing-CAMS sites. We also acknowledge Andreas Richter (Editor of this paper) and three anonymous referees for their helpful comments.

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
