# Peer review of "Evaluation of Transport Processes over North China Plain and Yangtze River Delta using MAX-DOAS Observations"

_EGUsphere, 2022_

## Author Response (AR1)

Referee#1

*Thank you for your careful review and constructive suggestions. These suggestions are quite valuable to us, and help improve our manuscript a lot.*

Point-to-point responses

*We appreciate the reviewers for their valuable and constructive comments, which are very helpful for the improvement of the manuscript. We have revised the manuscript carefully according to the reviewers' comments. We have addressed the reviewers' comments on a point-to-point basis as below for consideration, where the reviewers' comments are cited in **black**, and the responses are in **blue**.*

(1) Section 2.2, Line 131-132: the spectra measured with a solar zenith angle (SZA) of >75° to avoid the strong impact of stratospheric absorbers. Please elaborate the impact clearly.

Re: Thanks for this comment. We describe the impacts in Supplementary materials. Supplementary Section S1: "When SZA is over 75°, the scattering mainly occurs in the lower stratosphere and upper troposphere. At that time, DOAS measurements are very sensitive to stratospheric absorbers, while the sensitivity to near-surface absorbers is relatively lower. In other words, absorbers in stratosphere contribute considerably to the measurements, especially for lower elevation angles during early morning and late evening. In this study, we mainly focused on the tropospheric absorbers close to the ground surface, and thus needed to filter out the measurements with SZA > 75°."

(2) Section 2.3, Line 146-147: the clouds have large impacts on the data quality. Please describe this procedure and put it into the Supplementary materials.

Re: Thanks for this comment. We have added the following paragraph into the Supplementary materials.

Supplementary Section S2: "In the radiative transfer calculations of the aerosol and trace gas profile retrieval, the layers were assumed to be horizontally homogeneous and cloud impacts were not considered in this calculation process. Notably, the presence of cloud would result in inhomogeneous or/and rapidly fluctuating radiation transport conditions, which might bring uncertainties into the retrieval results. Therefore, we needed to filter the retrieved differential slant column densities (DSCDs) by screening out cloudy scenes before further processing for the profile retrieval (Chan et al., 2019). Since the vertical distribution of the oxygen collision complex $O_4$ is nearly constant, the retrieved $O_4$ DSCDs and (relative) intensities ought to vary smoothly with time, or with the solar and viewing geometry. Any rapid change in $O_4$ DSCDs and intensities suggests a sudden variation of the radiative transport condition, which is possibly linked to the presence of clouds.

Thus, to filter data influenced by inhomogeneous and/or rapidly varying radiation transport conditions, we applied a locally weighted regression smoothing filter (LOWESS) (Cleveland, 1981) with a regression window of 3 h to the $O_4$ DSCDs and intensity time series at each elevation angle. Data with sharp changes in $O_4$ DSCDs and intensities were filtered out. Only data with slowly varying $O_4$ DSCDs and intensities were adopted for the subsequent profile retrieval. The limitation of this cloudy scenes removing algorithm is that the algorithm is not able to distinguish between continuous and homogeneous cloud conditions. Nevertheless, it is rare that the cloud does not alter for a long time (within an hour) and the cloud layer keeps homogeneous for all viewing directions.
"

(3) Section 3.1, Line 273-275: 'After 16:00, the high-extinction air mass shifted MTL from to 300–1000 m toward the surface at SJZ, with the AEC gradually exceeding 0.5 km$^{-1}$ (Fig. 3)'. Which shift do you want to emphasize, the shift of MTL caused by the high-extinction air mass or the shift of air mass? Please reorganize the sentence.

Re: Thank you for this correction. As shown in Fig. 3, the fact that the near-surface AEC gradually exceeded 0.5 km$^{-1}$ after 16:00 indicated that high-extinction air mass had a tendency of moving towards the ground. According to the definition of main transport layer (MTL) in Section 2.5, MTL was determined by the concentration and wind speed in the corresponding layer. We want to emphasize that the shift of high-extinction air mass intrigued an increase in AEC in the corresponding layers, eventually causing the MTL to drop from 300-1000 m towards the surface. We have followed the suggestion and reorganized the sentence.

"In the late afternoon, aerosols gradually accumulated towards the surface, and triggered a variation in the distribution of $F_i$. After 16:00, the shift in the high-AEC air mass caused the transport fluxes in the lower layers (100–200 m) to increase to > 1.1 and ~ 2 km$^{-1}$·m·s$^{-1}$ for the CAMS and SJZ stations, respectively."

(4) Fig. 7: Why is the data missing at NC and XH stations during March 6-22, 2021?

Re: Thank you for this comment. This Figure was comparison result between dusty day and clean days. According to Fig. S9, we knew the HCHO results of NC and XH were both sound on clean days (March 6 and 22). In contrast, the poor quality of HCHO results on dusty day (March 15) made it impossible for us to calculate growth rates here, as G=([P]dust-[P]clean)/[P]clean.

We have checked the data of these two stations. For NC station, the number of retrieved HCHO profiles was very few (only 10), and only 1 profile met the filtering standard (DOF>1, relative error<0.5). For XH station, we only obtained 7 HCHO profiles and all of them were filtered according to our standard.

Technical comments:

Line 177, 'the wind speed in the southwest-northeast direction (WS)' → 'the wind speed (WS) in the southwest-northeast direction'

Line 191-192, 'Due to the large discrepancy in their vertical distribution, the MTLs of various pollutants were bound to have different varying characteristics' → 'Due to the large discrepancy in the vertical distribution of various pollutants, their MTLs were bound to have different varying characteristics'

Line 220, 'semibasin' → 'semi-basin', 'intraregional' → 'intra-regional'

Line 264, add space between 'MTL' and 'from', the logic of this sentence needs to be reconsidered.

Line 302: 'According to the selection standards described in Supplement Sect. S3, we confirmed that March 15 was a dusty day' and 'dusty day' is used in the following paragraphs. However, in Supplement Sect. S3, the date when the dust storm happened is defined as 'dust day'. Please use the unified definition between the manuscript and the supplementary materials.

Line 360, 'two stations assigned to the dark group (DG) located on the right' → 'two stations assigned to the dark group (DG) are located on the right.'

Line 411, 'four periods: west-to-east, YRD to NCP, transformation, and NCP to YRD' → 'four periods: West-to-East, YRD-to-NCP, Transformation, and NCP-to-YRD'. To keep in accordance with captions in Fig 12.

Fig. 11: the date format of 'yyyy-mm-dd' is different from that of other figures. Please take the unified date format.

Fig. S11-15: 'surface, 500 m, 800 m, 1000 m and 1500 m' → 'surface, 500, 800, 1000 and 1500 m'

Re: Thank you for these comments. We have followed these suggestions and corrected these mistakes accordingly.

**References**

Cleveland, W. S.: LOWESS: A Program for Smoothing Scatterplots by Robust Locally Weighted Regression The American Statistician, 35, 54, 10.2307/2683591 1981.

Referee#2

*Thank you for your careful review and constructive suggestions. These suggestions are quite valuable to us, and help improve our manuscript a lot.*

Point-to-point responses

*We appreciate the reviewers for their valuable and constructive comments, which are very helpful for the improvement of the manuscript. We have revised the manuscript carefully according to the reviewers' comments. We have addressed the reviewers' comments on a point-to-point basis as below for consideration, where the reviewers' comments are cited in **black**, and the responses are in **blue**.*

**General Comments:**

I will address the authors directly in this comments and hence use the personal pronoun "you" (and, correspondingly the possessive pronoun "your") instead of writing "the authors".

I am still not convinced, as already mentioned in the pre-review, that the statements you make, are really supported by the data. In the "details" section below, I go through the different occasions where I either disagree or simply cannot see your statement confirmed by the data. You should have many more data to choose from and I wonder if you made the best choice of data to show in order to demonstrate your "transport phenomena". I suggest to screen your data again and to see if you find occasions with more continuous (less missing data) and possibly more consistent data. For me, the data seems to very fit to discriminate between secondary formation and transport. Only the data used to show your point (2) is convincing. The other two examples do not convince me.

Re: Thank you for this comment. According to your suggestions, we have found some key problems to cause misunderstanding and confusion.

For Section 3.1, we find there are many speculative arguments making Section 3.1 confusing and less convincing. We have separated the discussions of Fig.3 and Fig. 4 into two parts, reducing the frequency of jumping descriptions between two Figures to some extent. To make Section 3.1 more understandable and emphasize the main points, we have deleted many speculative conclusions (e.g., secondary aerosol generation, aerosol transport among stations). The main point of Section 3.1 is to discuss different varying characteristics of MTL for aerosol, $NO_2$, and HCHO. We have changed the structure of Section 3.1 and added some demonstrations to make the descriptions about the MTL more prominent in this section.

For Section 3.2, we feel that we paid much attention to the differences between the vertical profile shapes among different stations before. For example, we thought the vertical profile shapes could be classified into multiple-peak and Gaussian shapes on clean days. However, as you said, some little differences, like the difference between Gaussian-shaped vertical profile with high surface concentration and two-peak shape,

could be caused by other factors (e.g., an imperfect aerosol retrieval) instead of pollutant vertical distribution itself. Instead, we find a better method, which can summarize the varying features of vertical profiles on the dusty day better—the movement of high-altitude peaks. Previous study have indicated that the AECs in layers of 0-500 m contribute most to trace gas profile variation, and trace gas concentration at 1.5-3.5 km responded most sharply to perturbations in AEC below 500 m (Friedrich et al., 2019). The trace gas profile below 1 km shows low sensitivity to AEC variation. Moreover, the increase at AEC tends to intrigue a decrease at trace gas concentration in 0-500 m layers. However, we find that the considerable growth at low-altitude trace gas concentration always accompany a sharp increase at surface AEC on the dusty day. Therefore, we think the peaks of trace gas below 0.5 km are mainly attributed to concentration increase itself instead of imperfect aerosol retrieval.

For Section 3.3, we have modified the wind field Figures to make them clearer, including enhancing the arrow thickness and changing the arrows' colors according into wind speeds.

After these revisions, we believe that the discussions about three examples will become more convincing.

I positively note that you improved on some occasions your references. As detailed in the the section below, I would like to motivate you to keep working on improving your references.

Re: Thank you for this comment. We improve our references according to the detailed comments.

I find it hard to remember all the stations and which stations belong to which region. This, combined with the fact that the station abbreviations in Figure 1 are not all very well visible, I suggest that you make a table of all stations, color-coded by "region" in the main text (essentially, move Table 1 to main text and improve the visual appeal a bit). Regarding the station abbreviations: You use both NC and NB, if the choice of abbreviations is yours, I recommend to change this. However if those are fixed, nothing you can do. Similarly, if you could avoid the abbreviation CAMS for one of your stations, that would be something to consider. In general, it might be good practice to use a two-letter abbreviation for stations and a three letter acronym for regions. That way, the text immediately becomes a bit easier to read.

Re: Thank you for this comment. We have tried color-coding before, but we were told by editors that colored text is not allowed in the manuscript. In addition, many stations are assigned to the overlap of two regions, and such stations are difficult to be color-coded by "region". For example, all the stations belonging to the JJJ region (i.e., CAMS, NC, WD, SJZ, and XH), are also located in the NCP region. The HNU station is situated at the overlap of the NCP and YRD.

For abbreviations, they are fixed, and it is hard for us to change them. To make the text easier to read, we make a new Table. 1 in the manuscript, which lists all the full names, acronyms, and exact locations of selected stations. In addition, for some demonstrations containing a certain region, we add some station information behind it. For example, in Section 3.2, "In the JJJ region (including NC, XH, and SJZ stations), we found that high-altitude peaks…".

**Table 1.** The names (codes), latitudes and longitudes of stations and their corresponding regions.

| Region | | Station (Code) | Longitude (°E) | Latitude (°N) |
|---|---|---|---|---|
| North China Plain (NCP) | Jing-Jin-Ji (JJJ) | Shijiazhuang (SJZ) | 114.61 | 37.91 |
| | | Wangdu (WD) | 115.15 | 38.17 |
| | | Nancheng (NC) | 116.13 | 39.78 |
| | | Chinese Academy of Meteorological Sciences (CAMS) | 116.32 | 39.95 |
| | | Xianghe (XH) | 116.98 | 39.76 |
| | | Dongying (DY) | 118.98 | 37.76 |
| Overlapping Zone | | Huaibei Normal University (HNU) | 116.81 | 33.98 |
| Yangtze River Delta (YRD) | | Ningbo (NB) | 121.90 | 29.75 |

The division of figures presented in the main paper and in the supplements seems very arbitrary. In many of the descriptions of the data, you refer very frequently to plots at different locations in both documents. This makes it very cumbersome and time consuming to find the relevant information. I encourage you to overthink the distribution of information in the main article and the supplement, as well as the general choice of figures.

Re: Thank you for this comment. After reviewing the manuscript, we think the diurnal variation of column transport fluxes ($F_c$) ought to be presented in the main text. This is because the $F_c$ comparison result helps us find that SJZ is an important source of the transported aerosol and NO2, and WD is one of main HCHO sources during this regional transport, which largely affect air quality of cities along southwest-northeast transport pathway. We moved this Figure to the main text as Figure. 5.

In addition, we incorporated growth rates of AECs, $NO_2$, and HCHO into one Figure (Fig. 7). The second main discussion point of Section 3.2 is to demonstrate the impacts of dust on aerosol, $NO_2$, and HCHO. Thus, the AEC growth rate should be put in the main text instead of in the Supplementary materials.

In order to reduce the frequency of jumping descriptions between Fig. 3 and Fig. 4, we have changed the structure of Section 3.1 largely. We firstly followed Fig. 3, and introduced the spatiotemporal distribution of dominant pollutants at each station. Then, we mainly followed Fig. 4, and discussed the MTL of aerosol, $NO_2$, and HCHO.

You frequently jump in the description of figures, most notably for Figures 3 and 4 which makes it difficult or almost impossible to follow. More generally, in the description of the data, you neither strictly follow a certain molecule, nor a certain station/ region nor an order in the figures. All this makes it hard to impossible to follow and very much obscure the points you want to make. I suggest to make a bullet point list (or maybe better a numbered list) of points you want to make and then describe one by one how this statement is supported by the data.

Re: Thank you for this comment. There are many speculative arguments making Section 3.1 confusing and less convincing. In addition, our descriptions jump between

Fig. 3 and Fig. 4, which easily cause misunderstanding. Therefore, we have done a major revision of Section 3.1. We firstly followed Fig.3 and introduced the spatiotemporal distribution of dominant pollutants at each station. Then, we mainly followed Fig.4 and discussed the MTL of aerosol, $NO_2$, and HCHO. Meanwhile, we deleted many speculative conclusions (e.g., secondary aerosol generation, aerosol transport among stations) to make paragraphs more understandable and emphasize the main points. Finally, combining with Fig. S4, we thought that "the aerosol and $NO_2$ from SJZ and HCHO from WD would largely affect air quality of cities along southwest-northeast transport pathway during this regional transport". The main paragraphs are listed as follows.

[revised manuscript text omitted]

Additional confusion is introduced by random use of past and present tense. It is never clear whether a statement is made and refers to something that was shown in a previous section or paragraph, or whether the following paragraph will contain the affirmation of the statement, based on the data. Since I am not a native English speaker, I refrain from giving advice here and instead suggest to consult a native speaker about the best use of different tenses.

Re: We let a language revision institution help us polish our language and correct mistakes.

[Figure]

**Certificate of Elsevier Language Editing Services**

The following article was edited by Elsevier Language Editing Services:
"Evaluation of Transport Processes over North China Plain and Yangtze River Delta using MAX-DOAS Observations"

Authored by:
Yuhang Song

Except for paragraphs around line 291 and the paragraph following line 386, I do not see a lot of support for the statements made. Maybe because I simply cannot follow the argumentation, or maybe because the data in fact does not support the claims. In any case, this is not good and both the suggestions above [regarding the organization and order of arguments] as well as the comments following [more on the presentation of data and some lacking analysis] will help to improve this.

The quality and presenting choices of the figures should be improved. Especially the choice how to display wind fields is not well made. It is absolutely impossible to see the actual orientation, the arrow ends are not visible at all. Due to the size, the actual speed is also unclear. For the latter, I suggest to use an underlying semi-transparent color map layer. For the former, I suggest to use larger (and thicker) but more sparsely placed arrows. However, also other figures need improvement, e.g. choice of color scale or combination of colors and ordering of line plots, details see below.

Re: We have improved all the figures according to the detailed comments below. Regarding the error analysis: You do now include a section on integrated column and surface error. I note this positively, thank you for taking up my critics of the pre-review. However, neither the quality of that, nor the extend are very satisfying. I cannot follow how you get to the percentage values you quote. Please include some equations you used to calculate those values.

Re: Thank you for this comment. We elaborate each error definition, calculation formulae, and estimation methods in the Supplementary Sect. 3 as follows.

"**Section S3. Error calculation and estimation**

The smoothing error ($S_s$) is a quantification of the error arising from the limited vertical resolution of profile retrieval, which can be calculated by Eq. s1. The noise error ($S_n$) represents the fitting error of the DOAS fits, primarily owing to the uncertainty in the measurements. The error of the retrieved state vector ($\widehat{S}$) is considered as the sum of these two independent error sources, $\widehat{S} = S_s + S_n$, and can be quantified by Eq. s2 (Frieß et al., 2006). Thus, in this study, we obtained the sum of smoothing and noise errors by averaging the error of retrieved profiles.

$$\mathbf{S_s} = (\mathbf{AK} - 1)\mathbf{S_a}(\mathbf{AK} - 1)^T \tag{s1}$$

$$\hat{\mathbf{S}} = (\mathbf{K^T S_\varepsilon^{-1} K} + \mathbf{S_a^{-1}})^{-1} \tag{s2}$$

where $\boldsymbol{AK}$ is the averaging kernel, which is the sensitivity of the retrieved state to the true state; $\boldsymbol{S_a}$ and $\boldsymbol{S_\varepsilon}$ are the covariance matrices of a priori and measurement, respectively; $\boldsymbol{K}$ is the weighting function matrix (Jacobi matrix), describing the sensitivity of the measurement to perturbations in the state vector.

The algorithm error is the discrepancy between the measured ($\boldsymbol{y}$) and modelled DSCDs ($F(\boldsymbol{x}, \boldsymbol{b})$), which is mainly caused by an imperfect minimum of the cost function ($\chi^2$) in Eq. s3. This error is a function of the viewing angle. Due to the difficulty of assigning this error to each altitude of profile, the algorithm errors on the near-surface values and column densities are usually estimated by calculating the average relative differences between the measured and modeled DSCDs at the minimum and maximum elevation angle (except 90°), respectively (Wagner et al., 2004).

$$\chi^2 = (\mathbf{y} - F(\mathbf{x}, \mathbf{b}))^T \mathbf{S}_\varepsilon^{-1}(\mathbf{y} - F(\mathbf{x}, \mathbf{b})) + (\mathbf{x} - \mathbf{x}_a)^T \mathbf{S}_a^{-1}(\mathbf{x} - \mathbf{x}_a) \tag{s3}$$

where $F(\boldsymbol{x}, \boldsymbol{b})$ is the forward model; $\boldsymbol{b}$ represents the meteorological parameters; $\boldsymbol{y}$ is the measured DSCDs; $\boldsymbol{x_a}$ is the a priori vector that serves as an additional constraint; $\boldsymbol{x}$ is the state vector.

The absorption cross section uncertainty is also an inevitable error source. Assuming the relative error of the cross section is $\boldsymbol{\delta}$, the uncertainty translated into an error in the retrieval space $\boldsymbol{S_c^x}$ can be calculated in the following operators:

$$\mathbf{S_c^y} = (\delta)^2 \cdot \mathbf{yy^T} \tag{s4}$$

$$\mathbf{gain} := \frac{\partial \mathbf{x}}{\partial \mathbf{y}} = (\mathbf{K^T S_\varepsilon^{-1} K} + \mathbf{S_a^{-1}})^{-1} \mathbf{K^T S_\varepsilon} \tag{s5}$$

$$\mathbf{S_c^x} = \mathbf{gain} \cdot \mathbf{S_c^y} \cdot \mathbf{gain^T}$$

(s6)

where $S_c^y$ represents the error in the measurement space; the $\boldsymbol{gain}$ matrix denotes the sensitivity of the state vector $\boldsymbol{x}$ to measurement $\boldsymbol{y}$. Previous study have indicated that the propagated error to the vertical column and vertical profile is similar to the original uncertainty in the cross section (Friedrich et al., 2019). Therefore, we used original cross section uncertainties ($O_4$: 4 %, $NO_2$: 3 %, and HCHO: 5 %) as our final results.

Owing to a temperature dependence of trace gas absorption, we needed to take into account the error related to the temperature dependence of the cross sections. With two cross sections at two temperatures, we firstly calculate the amplitude changes of the cross sections per Kelvin. Subsequently, we multiply this with the maximum temperature difference (~45K) during the measuring period to estimate this systematic error.

As one of input parameters for trace gas profile retrieval, the aerosol extinction profile plays a crucial role in retrieving the trace gas profile due to its strong impact on the air mass factor (AMF). The errors in the aerosol extinction profile retrieval (e.g., smoothing and noise errors) can be propagate to the trace gas vertical mixing ratio (VMR) and vertical column density (VCD). To quantify this propagated error, the sensitivity study of the trace gas profile to perturbations in the aerosol extinction profile is demanded. The sensitivity mainly includes slightly increasing the partial aerosol extinction of the $i$ th layer by 1% of the total optical density, and recording the difference between the perturbed and original trace gas profile in the matrix $\boldsymbol{D}$. The partial air column information is contained in the diagonal matrix $\boldsymbol{U}$. The uncertainty in aerosol profile retrieval is denoted by the matrix $\boldsymbol{S_{a,aerosol}}$. The errors translated into trace gas VMR profile ($\boldsymbol{S_{TG,VMR}^{aerosol}}$) can be calculated by Eq. s7, and the errors on the VCD ($\sigma_{TG,VCD}^{aerosol}$) is quantified by Eq. s8:

$$\mathbf{S_{TG,VMR}^{aerosol}} = \mathbf{D S_{a,aerosol} D^T}$$

(s7)

$$\sigma_{TG,VCD}^{aerosol} = \sqrt{\mathbf{g^T \cdot U \cdot D \cdot S_{a,aerosol} \cdot D^T U \cdot g}}$$

(s8)

where $\mathbf{g}$ is the total column operator for partial column profiles: $\boldsymbol{g^T} = (1,1,1,1, \dots 1)$. In our study, we just roughly estimated the errors of trace gas based on a linear propagation of the errors according to the total error budgets of aerosol retrievals, using Eq. s9:

$$\sigma_{TG}^{aerosol} = \sqrt{(\sigma_{aerosol}^{smooth\_noise})^2 + (\sigma_{aerosol}^{algorithm})^2 + (\sigma_{aerosol}^{cross\_section})^2 + (\sigma_{aerosol}^{temperature})^2}$$

(s9)

where $\sigma_{TG}^{aerosol}$ is the error of trace gas profile caused by aerosol profile retrieval error; $\sigma_{aerosol}^{smooth\_noise}$, $\sigma_{aerosol}^{algorithm}$, $\sigma_{aerosol}^{cross\_section}$, and $\sigma_{aerosol}^{temperature}$ represent the error budgets of aerosol retrieval related to smoothing and measurement noises, algorithm, cross section, and temperature dependence of cross section, respectively. It is worth

noting that algorithm error is not independent of the other error sources, and thus Eq. s9 can only be considered as a rough general estimation of errors related to aerosol retrieval. If a more realistic error estimate is demanded, additional sensitivity tests should be performed for different observation geometries.

Similarly, a general estimation of the total error is based on the square root of the sum of squares of different error terms, using Eq. s10 (for aerosol) or Eq. s11 (for trace gas).

$$\sigma_{total} = \sqrt{(\sigma_{smooth\_noise})^2 + (\sigma_{algorithm})^2 + (\sigma_{cross\_section})^2 + (\sigma_{temperature})^2} \qquad (s10)$$

$$\sigma_{total} = \sqrt{(\sigma_{smooth\_noise})^2 + (\sigma_{algorithm})^2 + (\sigma_{cross\_section})^2 + (\sigma_{temperature})^2 + (\sigma_{aerosol})^2} \qquad (s11)$$

"

Further, for the error analysis, you concentrate on integrated column and surface errors, however you mainly use profiles in your analysis. Hence, it is of uttermost importance to discuss the reliability of the profile shape. This is absent in your analysis. Since you also often argue that the data shows that the presence of aerosol triggers the formation of certain trace gases, it is important to discuss retrieval artifacts of aerosols leading to possibly incorrect ("too peaked") trace gas profiles, important in this context is also the frequent underestimation of dSCD errors and the effect on the trace gas profile (double peak, oscillations).

Re: We have added some discussions in Section 2.4.

"The trace gas retrieval errors, arising from the uncertainty in aerosol retrieval, were estimated as the total error budgets of the aerosols. Based on a linear propagation of the aerosol errors, the errors of trace gases were roughly estimated at 27% for VCDs and 14% for near-surface concentrations for the two trace gases. The perturbations of trace gas concentrations at each altitude caused by aerosol profile retrieval uncertainty resulted in a slight change in the profile shape. According to Friedrich et al. (2019), trace gas concentrations at 1.5-3.5 km respond most sharply to perturbations in the AEC profile, especially oscillations in the AEC below 0.5 km. The trace gas profile below 1.5 km shows a low sensitivity to AEC variation. Therefore, in this study, we focus mainly on the concentration variation below 1.5 km.

"

In Section 3.2, we mainly discussed the variation in high-altitude peaks instead of focusing too much on profile shapes. This is because some little differences, like the difference between Gaussian-shaped vertical profile with high surface concentration and two-peak shape, can be caused by other factors (e.g., an imperfect aerosol retrieval) instead of pollutant vertical distribution itself. Given that trace gas concentration below 1.5 km exhibits a relatively low sensitivity to AEC variation and high-altitude peaks are all at the 0-1 km with values mostly much higher than that in nearby layers, the peaks are less affected by an imperfect aerosol retrieval. Previous study has indicated that the AECs in layers of 0-500 m contribute most to trace gas profile variation, and trace gas concentration at 1.5-3.5 km responds most sharply to perturbations in AEC below 500 m (Friedrich et al., 2019). The trace gas profile below 1 km shows low sensitivity to AEC variation. Moreover, the increase at low-altitude AEC tends to intrigue a decrease

at trace gas concentration in 0-500 m layers. However, we find that the explosive growth at low-altitude trace gas concentration always accompany a sharp increase at surface AEC on the dusty day. Therefore, we think the peaks of trace gas below 0.5 km are mainly attributed to concentration increase itself instead of imperfect aerosol retrieval.

It would also be good to include more information about typical degrees of freedom (for the valid data), to make a comment on the percentage of data filtered out by the RMS and DOF criteria and to show at least an example of an averaging kernel. Additionally, since this (trace gas and aerosol profiles) is in some occasions not your "final" quantity you use for the interpretation of the data, you should also include further error analysis (the contribution of the model error on the wind and what this means for the flux).

Re: Thank you for this comment. We have added the demonstrations about the percentage of data filtered out in the main text.

Section 2.3: "To ensure the validity of the retrieved data, we removed the DOAS fit results with a root mean square (RMS) larger than $1.0 \times 10^{-3}$. After applying the RMS threshold, the results for $O_4$, $NO_2$, HCHO, HONO remained at 69.8%, 71.6%, 64.8%, and 73.1% respectively."

"The profiles of aerosols and trace gases were filtered out when DOF was less than 1.0 and the retrieved relative error was larger than 50% (Tan et al., 2018). About 0.5 %, 10.7 %, and 11.6 % of all measurements were discarded for aerosol, $NO_2$ and HCHO profile retrievals, respectively."

We add an example of an averaging kernel in the Supplementary materials (Fig. S2).

[Figure]

**Fig. S2.** An example of averaging kernel results from MAX-DOAS measurements at HNU station (March 6, 2021 at 14:33 LT) for (a) aerosol extinction, (b) $NO_2$, and (c) HCHO.

We supplement the transport flux error analysis in Supplement Sect. S4.

Supplementary Sect. S4: "Remarkably, there is an error of wind speed ($\delta_{w_i}$) caused by model uncertainty and an error of pollutant concentration ($\delta_{c_i}$) at each layer. These two kinds of errors propagate into the final transport flux results (i.e., $F_i$ and $F_c$), which can be quantified as follows:

$$\delta_{F_i} = \frac{\partial F_i}{\partial W_i} + \frac{\partial F_i}{\partial C_i} = C_i \times \delta_{W_i} + W_i \times \delta_{C_i} \tag{s12}$$

$$\delta_{F_c} = \sum \left( \delta_{F_i} \times H_i \right) = \sum \left( H_i \times C_i \times \delta_{W_i} + H_i \times W_i \times \delta_{C_i} \right) \tag{s13}$$

However, an accurate evaluation of wind speed simulation error is an enormous project and involves many factors, such as input parameters, topography and resolution (García-Bustamante et al., 2008; Carvalho et al., 2012; Orrell et al., 2001). By comparing simulation and observation results, Shimada et al. (2011) gives a relationship between relative biases and altitudes. Accordingly, we roughly estimated wind speed relative errors at different heights (0-400 m: 50 %, 400-800 m: 40 %, 800-1200 m: 20 %, 1200-1500 m: 10 %, 1500-3000 m: 3 %, > 3000m: 1 %). For pollutant concentration errors, we mainly considered retrieved errors (i.e., the sum of smoothing and noise errors), which play a dominant role in the total error budgets. The errors of $F_i$ and $F_c$ are displayed in Fig. S6 and Fig. S7, respectively.
"

[Figure]

**Fig. S6.** The vertical distribution of $F_i$ errors at SJZ, WD, NC, and CAMS stations for AECs, NO₂, and HCHO."

[Figure]

**Fig. S7.** The $F_c$ errors at SJZ, WD, NC, and CAMS stations for (a) AECs, (b) NO₂, and (c) HCHO.

Somewhat related to this: it is also not clear how exactly you treat the different height grids from the retrieval and the model wind. I think it is best to include a sentence or a small paragraph on this. Further I am not convinced that fluxes should consider the mixing ratios, I think they should be calculated using concentrations (just as you actually state you would do), more on this below in the detailed comments.

Re: Thank you for this comment. We supplement transport flux calculation details in Supplementary materials as follows (Supplementary Sect. S4).

"Due to the different height grids from the retrieval and the model wind, we needed to design a unified height grid. Thus, we divided 3.1 km into 13 layers: 0–100, 100–200, 200–300, 300–400, 400–600, 600–800, 800–1000, 1000–1200, 1200–1400, 1400–1600, 1600–2000, 2000–3000, 3000–3100 m. We averaged the wind speeds and pollutant concentrations at each layer to represent $W_i$ and $C_i$ in layer i, respectively. In addition, the time resolution of wind simulation is 1 hour, whereas that of the vertical profile is 15 minutes. In order to unify the time resolution, we averaged vertical profiles of an hour to calculate transport flux. Given that the results of last 15 minutes in each hour fit the situation of next hour better, we averaged the results from -15 min to +45 min as the hourly vertical profiles. For instance, we calculated the average from 9:45 to 10:45 to represent the vertical profile of 10:00."

Honestly speaking, we had considered using mass concentrations (ug/m3) before. However, converting equations are related to temperature (T) and pressure (P) as follows.

$$X(ppb) = \frac{C(\mu g/m^3) \bullet M_g(g/mol)}{V_m(L/mol)}$$

$$V_m = \frac{R \bullet T}{P}$$

There are large differences at temperatures and pressures of different altitudes. Unfortunately, we don't have any profiles of T or P at these four stations. If we adopt unified standard atmospheric pressure or temperature, it will introduce larger uncertainties, since the differences of T and P at different layers cannot be neglected. But we think that using mixing ratios can represent the density of pollutants at corresponding heights as well. In addition, we just use the highest transport flux to roughly determine the MTL, and using other units has little impacts on the main

conclusions. Therefore, we didn't use mass concentrations here.

Regarding molecule notation: HCHO: check journal guidelines whether you should use HCHO or H2CO and whether or not you have to introduce the chemical formula (i.e. writing "Formaldehyde (H2CO)...." at the first occurrence or not.
Re: Thank you for this correction.
 "we retrieved the vertical profiles of aerosol, $NO_2$, and HCHO using…" -> "we retrieved the vertical profiles of aerosol, nitrogen dioxide ($NO_2$), and formaldehyde (HCHO) using…"
 "…and $O_3$ concentrations" -> "…and ozone ($O_3$) concentrations"
 "…, including $SO_2$, CO, $NO_2$" -> "…, including sulfur dioxide ($SO_2$), carbon monoxide (CO), nitrogen dioxide ($NO_2$)"
"…, and HONO" -> "…, and nitrous acid (HONO)"
Apart from the strange use of time (which I decide not to comment on more than I already did), I do not have many comments regarding the use of language. The few I have are listed together with the detailed comments.
Re: We let a language revision institution help us polish our language and correct mistakes.

**Detailed Comments:**

l.27: "..oppositely...also occurred in this station" I do not follow here, how can something be "oppositely" and "also" at the same time?
Re: We have rewritten the sentence here.
"The maximum transport flux of HCHO appeared in Wangdu (WD), and aerosol and $NO_2$ transport fluxes were assumed to be high in Shijiazhuang (SJZ), both urban areas being significant sources feeding regional pollutant transport pathways."
l.32: "Comparatively" to what?
Re: We have given up classifying the four stations and basing the following discussions on the classification. Therefore, these sentences are reorganized as follows.
"For the $NO_2$ levels, dust and aerosols had different effects. At the SJZ and Dongying (DY) stations, the decreased light intensity prevented $NO_2$ photolysis and favored $NO_2$ concentration increase. In contrast, dust and aerosols provided surfaces for heterogeneous reactions, resulting in reduced $NO_2$ levels at the Nancheng (NC) and Xianghe (XH) stations."
l.37: What are "practical observations"?
Re: The "practical observations" pointed to MAX-DOAS observations. The haze-amplifying mechanism had been confirmed in previous studies, mainly based on the simulation. Some previous studies also provided practical measurements to validate haze-amplifying mechanism, however, those measurements were mostly limited to the surface. This mechanism involved some phenomena occurring at upper layers, and thus surface pollutant distribution information was not enough to validate this mechanism. The MAX-DOAS observations provided more convincing validations for the haze-amplifying mechanism. To avoid misunderstanding, we have deleted "by practical

observations".

l. 46: "remarkably contributes" --> "contributes remarkably"

Re: Thanks for this comment. We have corrected this sentence as follows.

"Transport contributes significantly to pollution in some megacities."

l.46: Maybe add some references?

Re: We explained "Transport remarkably contributes to pollution in some megacities" in the following sentences "Firstly, transport…heterogeneous reactions (Huang et al., 2014)" (Line 46-60). And we introduced many previous studies related to the impacts of transport on some megacities. Thus, we didn't cite any reference in Line 46, since we had talked about the relevant papers behind.

l.46: I do not follow the phrase "transportation directly deteriorates the environment through the production and emission..."

Re: Thank you for this comment. We have changed this sentence to avoid misunderstanding.

"transportation directly deteriorates the environment through the production and emission of a large number of pollutants" -> "transport carries large amounts of pollutants, directly deteriorating air quality."

l.51/52: What is the difference between cross-regional and inter-regional transport? And is intra-regional transport the same as regional transport?

Re: The cross-regional is the same as inter-regional transport, while intra-regional transport has the same meaning as regional transport. The intra-regional transport (regional transport) referred to the transport phenomena happening within a region, with the southwest-northeast transport occurring in the JJJ region (Section 3.1) being a good example. The back-and-forth transboundary transport between NCP and YRD was a good case of cross-regional (inter-regional) transport, which pointed to the transport between two specific regions. We just used another expression to avoid repeat.

l.53: "local contributions was": plural or singular

Re: Thank you for this correction. "local contributions was" - > "local contribution was"

l.55/56: "interact with the planetary boundary layer (PBL) and create an environment favorable for direct emission accumulation" what is meant by this?

Re: For example, the aerosol transport can efficiently influence the PBL dynamics when they are transported to the upper PBL over a certain region, resulting in a suppressed PBL height and weakened turbulence, thereby facilitating local pollutant accumulation. In addition, if some aerosol precursors (nitrate, sulfate) are transported to the upper PBL, it could result in enhanced secondary aerosol production, and produce the same effects as described above.

l.59: "The movement of warm and humid air masses..." How does this fit to the rest of the paragraph?

Re: Line 55-57, "Furthermore, under certain conditions, some transported pollutants can interact with the planetary boundary layer (PBL) and create an environment favorable for direct emission accumulation and secondary formation enhancement, thereby indirectly amplifying the impacts of pollution (Li et al., 2017b; Wilcox et al., 2016; Petaja et al., 2016)." Here, we mainly discussed that transport could indirectly affect air quality by aggravating emitted pollutants accumulation and enhancing

secondary formation. "The movement of warm and humid air masses likely increases secondary aerosol formation by aggravating aqueous and heterogeneous reactions (Huang et al., 2014)." was an example that transport phenomena could boost secondary formation.

l.60: "Hence, ..." what does the "hence" refer to? To the movement of the warm and humid air masses?

Re: The "hence" refer to the direct and indirect impacts that could be caused by transport. Because there are so many possible impacts that could be caused by transport, it is necessary for us to understand the air pollutant transport that occurs in megacity clusters by using an appropriate measurement method.

l.66: Add "To" before "characterize"
Re: "Characterize" -> "Characterizing"
l.66: "monitoring" --> "monitored"
l.66: "ground level" --> "surface"?
l.70: add "to" between "used" and "investigate"
Re: Thank you for these great suggestions. We have followed these suggestions and corrected these mistakes accordingly.

l.71: what is "technological support"?
Re: Here, "technological support" pointed to "satellite observations". Since satellite observations include horizontal distribution information of pollutants, it can be considered an important tool to conduct horizontal pollutant transport analysis. To convey more clearly, we planned to adopt participle as adverbial of result.
"..., to provide technological support for horizontal pollutant transport analysis" - > "Satellite observations can be used to investigate the horizontal distribution of vertical column densities (VCDs) of $NO_2$, formaldehyde (HCHO), $O_3$, and aerosols on a global scale, providing technological support for horizontal pollutant transport analysis."
l.74: Are those references the best fitting references here?
Re: "(Yumimoto et al., 2016; Su et al., 2020b; Bessho et al., 2016)" -> "(Bessho et al., 2016; Veefkind et al., 2012)"
l.75/75: "The chemical transport model" --> "A chemical ..."
Re: Thank you for this correction. We plan to adopt plural here.
"The chemical transport model can be used to simulate pollutant distribution, and is an important tool for monitoring, forecasting, and analyzing atmospheric quality (Huang et al., 2018a)." -> "Chemical transport models can be used to simulate pollutant distribution and they are also important tools for monitoring, forecasting, and analyzing atmospheric quality (Huang et al., 2018a)."
l.77: What are "hypothetical conditions"?
Re: "However, considerable uncertainties remain in estimating pollutant distribution using model simulations, primarily owing to the effects of emission inventories, meteorological fields, and of assumptions made (Grell et al., 2005; Huang et al., 2016; Xu et al., 2016; Zhang et al., 2017)."
   As we all know, there are some basic model assumptions in simulation models (e.g.,

WRF/Chem, MM5/Chem), and the uncertainties caused by these assumptions cannot be ignored.

For example, Grell et al. (2005) indicated that the leaf temperature assignment assumption influenced $O_3$ statistics. In addition, other model components, such as surface layer parameterizations and boundary condition assumptions, contributed to model uncertainty.

l.80: What are "technical methods"?

Re: All methods used for monitoring atmospheric composition were called "technical methods" here. To avoid misunderstanding, we replaced "technical methods" by "monitoring methods".

l.80: Maybe remove Wang from the reference list here or make clear why it is important to add Wang here

Re: "(Collis, 1966; Barrett and Ben-Dov, 1967; Wang et al., 2020)" -> "(Collis, 1966; Barrett and Ben-Dov, 1967)"

l.92: This has been used by many many many groups, please add more representative references.

Re: "(Xing et al., 2020; Hong et al., 2022b)" -> "(Hönninger and Platt, 2002; Meena, 2004; Wagner et al., 2004; Frieß et al., 2006; Hönninger et al., 2004; Irie et al., 2008; Xing et al., 2020; Hong et al., 2022b)"

l.96: What do you mean by "hyperspectral stereoscopic"?

Re: We have deleted this description to make the sentence more understandable.

"a mature ground-based hyperspectral stereoscopic remote sensing network" -> "a mature ground-based remote sensing network"

l.98: "technical support"?

Re: "The data provided by the monitoring network successfully meet the actual demands for vertical observations, and the network provides powerful technical support for analyzing pollution sources and transport" -> "This monitoring network successfully meets the actual demands for vertical observations, providing powerful support for analyzing pollution sources and transport (Liu et al., 2022)."

l.101: "...impacts on and between regions"?

Re: "...impacts on and between regions"->"...impacts on regions"

l.104f: I think this belongs to conclusions.

Re: Thanks for this question. We have deleted this sentence here.

l.107: Is it not more the other way around: Since you mainly analyzed data from NCP and YRD, you concentrate on transport phenomena between those regions?

Re: Thank you for this correction.

"The analyzed transport phenomena mainly occurred in the NCP and YRD, which are two of the main plain areas in China" -> "Our study focused mainly on the transport phenomena in the NCP and YRD, two of the main plains within China."

l.112: This is unclear: Is the BTH region also called JJJ or what?

Re: The JJJ region is also called the BTH region, and abbreviations of BTH are also used in other papers. Given that we used JJJ here, we have deleted some unnecessary descriptions here to avoid some misunderstanding.

l.113: What characterizes the continental monsoon climate?

Re: Here, we wanted to emphasize that the JJJ region was largely affected by the wind. Due to thermal difference between land and sea, the change of wind direction between winter and summer is very obvious. In winter, the cold air comes from the high latitude continental area, blowing northerly wind, cold and dry; in summer, the wind mainly comes from the ocean, mostly southerly wind, humid and warm.

"In addition, the JJJ has a typical continental monsoon climate" -> "In addition, the JJJ has a typical continental monsoon climate, indicating that wind plays an important role in the local climate and environment."

l.113: "The regional transport of pollutants is prevalent within the JJJ region" seems to refer to a specific one, "The". Which?

Re: In order to better conclude the above and introduce the following, we have changed the sentence.

"The regional transport of pollutants is prevalent within the JJJ region, which exerts serious effects on local air quality." -> "The semi-basin geographical features and continental monsoon climate indicate that regional transport is a significant factor affecting local air quality in the JJJ region."

Figure 1: Add a scale, enlarge the color bar, black on dark green/ blue is not well visible. Blue on dark green/blue is not well visible. magenta and read on such a "colored" plot are not very well distinguishable. I suggest to make the underlying map semi-transparent (the orography color scaling only, not the region contours) and to additionally use a different line style to indicate the regions.

Re: Thank you for your advice. We have revised our picture as follows.

[Figure]

l.124: This is a skyspec 1D? Please specify

Re: Thank you for this suggestion. We have added this information.

"MAX-DOAS instrument (Airyx, Heidelberg, Germany)" -> "MAX-DOAS instruments (Airyx SkySpec-1D, Heidelberg, Germany)"

end of page 6: There is a loose Table caption here.

Re: Thank you for this suggestion. We have corrected it.

Table 1: For the fitting interval for HCHO, are the two $NO_2$ and $O_3$ cross sections orthogonalized?

Re: Yes, they are orthogonalized.

l. 139: "measured DSCDs") maybe retrieved since you do not measure them directly?

Re: "…measured $O_4$ DSCDs …" -> "Spectral analysis derives the slant column densities (SCDs), i.e., the integrated concentration along the light path. Subsequently, we calculated the differential slant column densities (DSCDs), which are defined as the difference between the off-zenith and zenith SCDs."

l.140: Please check your statement about the ring spectrum. It seems confusing.

Re: "Furthermore, we calculated the ring spectrum as the measured spectrum, considering the contribution of the stratosphere to the DSCDs"-> "The Ring spectrum

was added to the fitting settings to remove the influence of the stratosphere on the DSCDs."

l.142ff: regarding the choice of retrieval windows, do you base this on some reference? If so, please add.

Re: We cited some references in the following sentence, "We have used similar retrieval settings in our previous studies (Xing et al., 2020; Xing et al., 2021b)". To avoid misunderstanding, we have incorporated these two sentences into one.

"We analyzed the DSCDs of the oxygen dimer ($O_4$) and $NO_2$ in the interval between 338 and 370 nm, and we used the 322.5–358 nm and 335–373 nm wavelength intervals for HCHO and HONO absorption analysis, respectively. We have used similar retrieval settings in our previous studies (Xing et al., 2020; Xing et al., 2021b)." -> "We analyzed the DSCDs of the oxygen dimer ($O_4$) and $NO_2$ in the interval between 338 and 370 nm, and we used the 322.5–358 nm and 335–373 nm wavelength intervals for HCHO and nitrous acid (HONO) absorption analysis, respectively (Xing et al., 2020; Xing et al., 2021b)."

l.147: Which fraction of data (approximately) does not pass your RMS criterion?

Re: We add some descriptions about it.

"To ensure the validity of the retrieved data, we removed the DOAS fit results with a root mean square (RMS) larger than $1.0 \times 10^{-3}$. After applying the RMS threshold, the results for $O_4$, $NO_2$, HCHO, HONO remained at 69.8%, 71.6%, 64.8%, and 73.1% respectively."

l.148: Can you specify "slowly"? How do you implement this statement "we only use data with slowly varying..."

Re: We had thought this filtering procedure was not the main discussion point, and thus we just cited the corresponding reference behind, "The detailed filtering methods is provided by Chan et al. (2019)". However, given that this question was frequently proposed, we added a paragraph of description about filtering procedure and the cloud impacts in the Supplementary materials as follows.

Supplementary Sect. 2: "In the radiative transfer calculations of the aerosol and trace gas profile retrieval, the layers were assumed to be horizontally homogeneous and cloud impacts were not considered in this calculation process. Notably, the presence of cloud would result in inhomogeneous or/and rapidly fluctuating radiation transport conditions, which might bring uncertainties into the retrieval results. Therefore, we needed to filter the retrieved differential slant column densities (DSCDs) by screening out cloudy scenes before further processing for the profile retrieval (Chan et al., 2019). Since the vertical distribution of the oxygen collision complex $O_4$ is nearly constant, the retrieved $O_4$ DSCDs and (relative) intensities ought to vary smoothly with time, or with the solar and viewing geometry. Any rapid change in $O_4$ DSCDs and intensities suggests a sudden variation of the radiative transport condition, which is possibly linked to the presence of clouds. Thus, to filter data influenced by inhomogeneous and/or rapidly varying radiation transport conditions, we applied a locally weighted regression smoothing filter (LOWESS) (Cleveland, 1981) with a regression window of 3 h to the $O_4$ DSCDs and intensity time series at each elevation angle. Data with sharp changes in $O_4$ DSCDs and intensities were filtered out. Only

data with slowly varying O$_4$ DSCDs and intensities were adopted for the subsequent profile retrieval. The limitation of this cloudy scenes removing algorithm is that the algorithm is not able to distinguish between continuous and homogeneous cloud conditions. Nevertheless, it is rare that the cloud does not alter for a long time (within an hour) and the cloud layer keeps homogeneous for all viewing directions."

l.152: I think "maximum" should not directly be used as adjective to "posteriori state vector" but possibly to something like likelihood or so.

Re: "We selected the cost function $\chi^2$ to determine the maximum a posteriori state vector $x$." -> "By minimizing the cost function $\chi^2$, we determined the a posteriori state vector $x$:"

l.154: add surface albedo and aerosol properties here in brackets or change i.e. to e.g.

Re: "(i.e., atmospheric pressure and temperature profiles)" -> "(e.g., atmospheric pressure and temperature profiles)"

l.155: You come back to how you construct Sa later. However, instead of having this (theoretical concept, practical implementation and construction) at two different places, I would put all of this together here. The same for the a priori profile. Additionally, you do not include information about the trace gas a priori profile (or I missed it). For the aerosol profile, it is unclear (l.164) whether the value you state for the surface, is really the surface or whether it is the value used in your lowest layer and hence at 50 m. This is likely not very different, but I think you should be specific. It might also make sense to actually include which AOD this corresponds to since often the integrated value is stated for the choice of a priori instead of the surface value.

Re: Thank you for this comment. About this part, we elaborated the retrieval in this order; we introduced every parameter (e.g., b, S$_a$, x$_a$) and their exact meaning in this equation firstly; then, we described the vertical profile retrieval procedure; at last, we talked about the specific retrieval settings in this study. Given that the setting descriptions are a little long, we have put them together and at the end of paragraph, which is also clear and understandable.

We feel sorry to make a mistake here. Having checked the file of retrieval settings, we find that there are two setting modes about *a priori* aerosol extinction profile; one is to set the bottom layer extinction; the other is to set AOD to 0.4. The retrieval profile in our study adopted the latter one. Therefore, we have rewritten these demonstrations. In addition, we also add some information about the a priori trace gas profiles. Thank you for this comment.

"In this study, we separated the atmosphere into 20 layers from 0 to 3 km with a vertical resolution of 0.1 km under 1 km, and of 0.2 km from 1 to 3 km. For the aerosol profile retrieval, we selected an exponentially decreasing profile with a scale height of 0.5 km as a priori and set its aerosol optical density (AOD) to 0.4. For the a priori trace gas profile, we set the bottom layer concentration to $8 \times 10^{10}$ molec·cm$^{-3}$, and set the VCD to $15 \times 10^{15}$, $15 \times 10^{15}$, and $5 \times 10^{14}$ molec·cm$^{-2}$ for NO$_2$, HCHO, and HONO, respectively. We set the a priori uncertainties of the aerosol, NO$_2$, and HCHO to 50%, and HONO to 100%, with a correlation height of 0.5 km. During the retrieval, we

employed a fixed set of aerosol optical properties with a single-scattering albedo of 0.95, asymmetry parameter of 0.70, and surface albedo of 0.04."

l.157: "our first" what?

Re: "our first" -> "our first step"

l.158: Which previous study? (move reference form l 160 to l-158) Also, if you use this here, maybe highlight the difference to the "usual way".

l.159: what is meant by "semi-quantify"?

Re: Given that this reference is less associated with our main work or retrieval procedure, we planned to delete this description here (line 158-159).

l.163: add "the" between "For" and "aerosol".

l.163: add "an" between "selected" and "exponentially"

l.163: add "profile" after a priori (or reformulate as : decreasing profile .... as a priori). Also everywhere: put a priori italic.

Responses to l.163: Thank you for the suggestion. We check the ACP guidelines, which says "Foreign words, phrases, and abbreviations that cannot be found in any English dictionary (this does not apply to proper nouns) are italicized. Common Latin phrases are not italicized (for example, et al., cf., e.g., a priori, in situ, bremsstrahlung, and eigenvalue).".   Therefore, we didn't italicized a prior here.

- **Italicization**
  - Italic font may be used for emphasis, although this should be used sparingly (e.g. data were *almost* consistent).
  - Foreign words, phrases, and abbreviations that cannot be found in any English dictionary (this does not apply to proper nouns) are italicized. Common Latin phrases are not italicized (for example, et al., cf., e.g., a priori, in situ, bremsstrahlung, and eigenvalue).
  - Ship names are italic, but their prefixes are roman (e.g. RV *Polarstern*).
  - Genus and species names are italic; high-order taxonomic ranks are roman.
  - When mentioned in running text, the names of books, journals, pamphlets, magazines, and newspapers are italicized.

"For aerosol profile retrieval, we selected exponentially decreasing a priori with a scale height of 0.5 km." -> "For the aerosol profile retrieval, we selected an exponentially decreasing profile with a scale height of 0.5 km as a priori and set its aerosol optical density (AOD) to 0.4."

l.166: what do you use to convert asymmetry factor to phase function moments?

Re:

Here, we adopted a Henyey-Greenstein phase function as follows.

$$P_{HG}(\theta / g) = \frac{1}{2} \frac{1 - g^2}{(1 + g^2 - 2g\cos\theta)^{\frac{3}{2}}}$$

Set. 2.4: Please add the used formulae.

Re: Thank you for this comment. We elaborate each error definition, calculation formulae, and estimation methods in the Supplementary Sect. 3 as follows.

Supplementary Sect. 3: "**Section S3. Error calculation and estimation**

The smoothing error ($S_s$) is a quantification of the error arising from the limited vertical resolution of profile retrieval, which can be calculated by Eq. s1. The noise error ($S_n$) represents the fitting error of the DOAS fits, primarily owing to the uncertainty in the measurements. The error of the retrieved state vector ($\widehat{S}$) is considered as the sum of

these two independent error sources, $\hat{S} = S_s + S_n$, and can be quantified by Eq. s2 (Frieß et al., 2006). Thus, in this study, we obtained the sum of smoothing and noise errors by averaging the error of retrieved profiles.

$$\mathbf{S_s} = (\mathbf{AK} - 1)\mathbf{S_a}(\mathbf{AK} - 1)^T \tag{s1}$$

$$\hat{\mathbf{S}} = (\mathbf{K^T S_\varepsilon^{-1} K} + \mathbf{S_a^{-1}})^{-1} \tag{s2}$$

where $AK$ is the averaging kernel, which is the sensitivity of the retrieved state to the true state; $S_a$ and $S_\varepsilon$ are the covariance matrices of a priori and measurement, respectively; $K$ is the weighting function matrix (Jacobi matrix), describing the sensitivity of the measurement to perturbations in the state vector.

The algorithm error is the discrepancy between the measured ($y$) and modelled DSCDs ($F(x, b)$), which is mainly caused by an imperfect minimum of the cost function ($\chi^2$) in Eq. s3. This error is a function of the viewing angle. Due to the difficulty of assigning this error to each altitude of profile, the algorithm errors on the near-surface values and column densities are usually estimated by calculating the average relative differences between the measured and modeled DSCDs at the minimum and maximum elevation

angle (except 90°), respectively (Wagner et al., 2004).

$$\chi^2 = (\mathbf{y} - F(\mathbf{x}, \mathbf{b}))^T \mathbf{S_\varepsilon}^{-1}(\mathbf{y} - F(\mathbf{x}, \mathbf{b})) + (\mathbf{x} - \mathbf{x}_a)^T \mathbf{S_a}^{-1}(\mathbf{x} - \mathbf{x}_a) \tag{s3}$$

where $F(x, b)$ is the forward model; $b$ represents the meteorological parameters; $y$ is the measured DSCDs; $x_a$ is the a priori vector that serves as an additional constraint; $x$ is the state vector.

The absorption cross section uncertainty is also an inevitable error source. Assuming the relative error of the cross section is $\delta$, the uncertainty translated into an error in the retrieval space $S_c^x$ can be calculated in the following operators:

$$\mathbf{S_c^y} = (\delta)^2 \cdot \mathbf{yy^T} \tag{s4}$$

$$\mathbf{gain} := \frac{\partial \mathbf{x}}{\partial \mathbf{y}} = (\mathbf{K^T S_\varepsilon^{-1} K} + \mathbf{S_a^{-1}})^{-1} \mathbf{K^T S_\varepsilon} \tag{s5}$$

$$\mathbf{S_c^x} = \mathbf{gain} \cdot \mathbf{S_c^y} \cdot \mathbf{gain^T} \tag{s6}$$

where $S_c^y$ represents the error in the measurement space; the $gain$ matrix denotes the sensitivity of the state vector $x$ to measurement $y$. Previous study have indicated that the propagated error to the vertical column and vertical profile is similar to the original uncertainty in the cross section (Friedrich et al., 2019). Therefore, we used original cross section uncertainties (O$_4$: 4 %, NO$_2$: 3 %, and HCHO: 5 %) as our final results.

Owing to a temperature dependence of trace gas absorption, we needed to take into account the error related to the temperature dependence of the cross sections. With two cross sections at two temperatures, we firstly calculate the amplitude changes of the cross sections per Kelvin. Subsequently, we multiply this with the maximum

temperature difference (~45K) during the measuring period to estimate this systematic error.

As one of input parameters for trace gas profile retrieval, the aerosol extinction profile plays a crucial role in retrieving the trace gas profile due to its strong impact on the air mass factor (AMF). The errors in the aerosol extinction profile retrieval (e.g., smoothing and noise errors) can be propagate to the trace gas vertical mixing ratio (VMR) and vertical column density (VCD). To quantify this propagated error, the sensitivity study of the trace gas profile to perturbations in the aerosol extinction profile is demanded. The sensitivity mainly includes slightly increasing the partial aerosol extinction of the $i$ th layer by 1% of the total optical density, and recording the difference between the perturbed and original trace gas profile in the matrix $\mathbf{D}$. The partial air column information is contained in the diagonal matrix $\mathbf{U}$. The uncertainty in aerosol profile retrieval is denoted by the matrix $\mathbf{S_{a,aerosol}}$. The errors translated into trace gas VMR profile ($\mathbf{S^{aerosol}_{TG,VMR}}$) can be calculated by Eq. s7, and the errors on the VCD ($\sigma^{aerosol}_{TG,VCD}$) is quantified by Eq. s8:

$$\mathbf{S^{aerosol}_{TG,VMR} = DS_{a,aerosol}D^{T}} \tag{s7}$$

$$\sigma^{aerosol}_{TG,VCD} = \sqrt{\mathbf{g^{T}U \cdot D \cdot S_{a,aerosol} \cdot D^{T}U \cdot g}} \tag{s8}$$

where $\mathbf{g}$ is the total column operator for partial column profiles: $\mathbf{g^{T}} = (1,1,1,1,\dots 1)$. In our study, we just roughly estimated the errors of trace gas based on a linear propagation of the errors according to the total error budgets of aerosol retrievals, using Eq. s9:

$$\sigma^{aerosol}_{TG} = \sqrt{(\sigma^{smooth\_noise}_{aerosol})^2 + (\sigma^{algorithm}_{aerosol})^2 + (\sigma^{cross\_section}_{aerosol})^2 + (\sigma^{temperature}_{aerosol})^2} \tag{s9}$$

where $\sigma^{aerosol}_{TG}$ is the error of trace gas profile caused by aerosol profile retrieval error; $\sigma^{smooth\_noise}_{aerosol}$, $\sigma^{algorithm}_{aerosol}$, $\sigma^{cross\_section}_{aerosol}$, and $\sigma^{temperature}_{aerosol}$ represent the error budgets of aerosol retrieval related to smoothing and measurement noises, algorithm, cross section, and temperature dependence of cross section, respectively. It is worth noting that algorithm error is not independent of the other error sources, and thus Eq. s9 can only be considered as a rough general estimation of errors related to aerosol retrieval. If a more realistic error estimate is demanded, additional sensitivity tests should be performed for different observation geometries.

Similarly, a general estimation of the total error is based on the square root of the sum of squares of different error terms, using Eq. s10 (for aerosol) or Eq. s11 (for trace gas).

$$\sigma_{total} = \sqrt{(\sigma_{smooth\_noise})^2 + (\sigma_{algorithm})^2 + (\sigma_{cross\_section})^2 + (\sigma_{temperature})^2} \tag{s10}$$

$$\sigma_{total} = \sqrt{(\sigma_{smooth\_noise})^2 + (\sigma_{algorithm})^2 + (\sigma_{cross\_section})^2 + (\sigma_{temperature})^2 + (\sigma_{aerosol})^2} \tag{s11}$$

"

l.175: Smoothing error is related to AK and hence on Sa. How does this refer then

directly to the DOAS fit?

Re: Sorry, there is a mistake in our demonstrations. We have revised it as follows. "Smoothing and noise errors refer to the errors caused by the limited vertical resolution of profile retrieval, and the fitting error of DOAS fits, respectively."

l.177ff: You start arguing about local minima ("imperfect minima") but then continue talking about which elevation angles hold more info on which profile height (that should be characterized by your gain matrix, right?), however, I don't see the direct connection here. Please elaborate.

Re: There is a mistake in our demonstrating order, and we have modified it. We elaborate it in the main text and Supplementary materials as follows.

Main text: "Algorithmic error (i.e., the difference between the measured and modeled DSCDs) arises from an imperfect minimum of the cost function. This error is a function of the viewing angle. However, it is difficult to assign discrepancies between the measured and modeled DSCDs at each profile altitude. Therefore, the algorithm error on the near-surface values and column densities cannot be realistically estimated. Given that measurements at 1° and 30° elevation angles are sensitive to the lower and upper air layers, respectively, the average relative differences between the measured and modeled DSCDs for a 1 and 30° elevation angles can be used to estimate the algorithm errors on the near-surface values and column densities, respectively (Wagner et al., 2004). Considering its trivial role in the total error budget, we estimated these errors on the near-surface values and the column densities at 4 and 8 % for aerosols, 3 and 11 % for NO₂, and 4 and 11 % for HCHO, respectively, according to Wang et al. (2017)."

Supplementary Sect. 3: "The algorithm error is the discrepancy between the measured ($y$) and modelled DSCDs ($F(x, b)$), which is mainly caused by an imperfect minimum of the cost function ($\chi^2$) in Eq. s3. This error is a function of the viewing angle. Due to the difficulty of assigning this error to each altitude of profile, the algorithm errors on the near-surface values and column densities are usually estimated by calculating the average relative differences between the measured and modeled DSCDs at the minimum and maximum elevation angle (except 90°), respectively (Wagner et al., 2004).

$$\chi^2 = (\mathbf{y} - F(\mathbf{x}, \mathbf{b}))^T \mathbf{S}_\varepsilon^{-1} (\mathbf{y} - F(\mathbf{x}, \mathbf{b})) + (\mathbf{x} - \mathbf{x}_a)^T \mathbf{S}_a^{-1} (\mathbf{x} - \mathbf{x}_a)$$
(s3)

where $F(x, b)$ is the forward model; $b$ represents the meteorological parameters; $y$ is the measured DSCDs; $x_a$ is the a priori vector that serves as an additional constraint; $x$ is the state vector."

As a more common reference for O4, I know Wagner et al. 2009 who quote 10% accuracy which is substantially higher than the quoted 4%, however the quoted reference is also more recent (2013), maybe double check?

Re: We have double checked it. The variation in different temperature ranges correspond to different error. The temperature of O₄ absorption cross section we selected is 293K, and its corresponding error is around 4%. The 10% error you mention corresponds to the O₄ absorption cross section at 203K.

l.191: Please check the sentence, something goes wrong here

Re: "Given that the measurement period (from January 1 to March 31, 2021) is in the winter-spring season, we roughly estimated the maximum temperature difference to be 45 K."

l.192: What do you mean by "temperature gap"?

Re: To avoid misunderstanding, we have changed the description, "temperature gap" -> "maximum temperature difference"

l.194: I very strongly recommend, as already mentioned in the general comments, to do such investigations. Given the importance of the trace gas profile shape in this study, it is important to have a really good understanding of the effect of imperfect aerosol retrieval on the trace gas retrieval.

Re: Thank you for this comment. We have done many investigations and added some descriptions about the effect of imperfect aerosol retrieval in Section 2.4.

"The trace gas retrieval errors, arising from the uncertainty in aerosol retrieval, were estimated as the total error budgets of the aerosols. Based on a linear propagation of the aerosol errors, the errors of trace gases were roughly estimated at 27% for VCDs and 14% for near-surface concentrations for the two trace gases. The perturbations of trace gas concentrations at each altitude caused by aerosol profile retrieval uncertainty resulted in a slight change in the profile shape. According to Friedrich et al. (2019), trace gas concentrations at 1.5-3.5 km respond most sharply to perturbations in the AEC profile, especially oscillations in the AEC below 0.5 km. The trace gas profile below 1.5 km shows a low sensitivity to AEC variation. Therefore, in this study, we focus mainly on the concentration variation below 1.5 km."

Eq.2: Or just sqrt(0.5)(va + ua)

Re: These two equations are different. The sqrt(0.5)(va + ua) reflects the wind speed in the real direction, while the $WS = v \times \cos\frac{\pi}{4} + u \times \sin\frac{\pi}{4}$ represents the wind speed projection in the southwest-northeast pathway.

Impacted by the semi-basin topography, southwesterly or southerly winds play an important role in the JJJ region's pollutant transport. In this study, we mainly pay attention to the southwest-northeast transport. Therefore, we adopted the latter equation. The $F_i$ and $F_c$ both point to the transport flux projection in the southwest-northeast pathway.

[Figure]

To avoid misunderstanding, we have changed the demonstrations at the start of Section 2.5.

"Owing to the semi-basin topography, southwesterly or southerly winds play a dominant role in pollutant transport in the JJJ region. In this study, we mainly focused on pollutant transport in the southwest-northeast direction, and thus selected four different stations along this pathway, namely, Shijiazhuang (SJZ), Wangdu (WD), Nancheng (NC), and Chinese Academy of Meteorological Sciences (CAMS) (Fig. 1). We calculated the hourly transport fluxes of each layer ($F_i$) and column transport fluxes ($F_c$) at each station to illustrate the dynamic transport process of pollutants along the southwest-northeast pathway. The detailed calculation methods are described below.

First, the wind speed projection (WS) in the southwest-northeast direction was calculated as follows: … "

Eq 3: The dimension of flux should be "quantity over area over time". If you use, as indicated by this equation, in fact the concentration (in e.g. "molec/cm3"), then the dimension is correct, since you get: molec/cm3 * m/s --> 100 molec/cm2/s. However, this seems not to be what you actually did, considering Fig. 3. Given that the same concentration in terms of molec/cm3 corresponds to a very different mixing ratio in terms of ppb at different heights, I would think that the former would be the better quantity to use (also, it is more in line with the classic definition of a flux as quantity per area per time).

Re: Thank you for this comment. Honestly speaking, we had considered using concentrations before. However, converting equations are related to temperature (T) and pressure (P) as follows.

$$X(ppb) = \frac{C(\mu g/m^3) \cdot M_g (g/mol)}{V_m (L/mol)}$$

$$V_m = \frac{R \cdot T}{P}$$

There are large differences at temperatures and pressures of different altitudes. Unfortunately, we don't have any profiles of T or P at these four stations. If we adopt unified standard atmospheric pressure or temperature, it will introduce larger uncertainties, since the differences of T and P at different layers cannot be neglected. But we think that using mixing ratios can represent the density of pollutants at corresponding heights as well. In addition, we just use the highest transport flux to roughly determine the MTL, and using other units has little impacts on the main conclusions. Therefore, we didn't use concentrations here.

Eq.4: In order not to introduce "per unit width" which is somewhat confusing, I would recommend to divide by sum/H_i, that way you keep the correct dimension of flux.
Re: Thank you for this advice. However, summing the $F_i$ multiplied by the height of each layer might match with the definition of column transport flux ($F_c$) better, since $F_c$ means the sum of each layer's transport flux and its shape is like a column. If diving by $\sum H_i$ to get the average, $F_c$ might deviate from its original meaning. To make this section more understandable, we have added a paragraph describing data processing methods (e.g., treating the different height grids from the retrieval and the model wind) in Supplementary materials (Supplementary Sect. 4).
l.221: high --> highest
Re: Thank you for this advice. We have corrected this one.
l.222: Eq. 4 is simply the definition, it does not demonstrate anything
Re: "Equation 3 demonstrated that MTL was determined by the concentration and wind speed in the corresponding layer." -> "According to the definition and Eq. 3, we know that the MTL is determined by the concentration and wind speed in the corresponding layer."
Sect. 2.5: You need to include a discussion here about the model error on velocity and the effect on the total flux error. You do not put any error on your flux, c.f. comment in general comments.
Re: Thank you for your comment. We added these contents in Supplementary Sect. S4 as follows:

"Remarkably, there is an error of wind speed ($\delta_{w_i}$) caused by model uncertainty and

an error of pollutant concentration ($\delta_{c_i}$) at each layer. These two kinds of errors

propagate into the final transport flux results (i.e., $F_i$ and $F_c$), which can be quantified as follows:

$$\delta_{F_i} = \frac{\partial F_i}{\partial W_i} + \frac{\partial F_i}{\partial C_i} = C_i \times \delta_{W_i} + W_i \times \delta_{C_i}$$

(s12)

$$\delta_{F_c} = \sum \left( \delta_{F_i} \times H_i \right) = \sum \left( H_i \times C_i \times \delta_{W_i} + H_i \times W_i \times \delta_{C_i} \right)$$

(s13)

However, an accurate evaluation of wind speed simulation error is an enormous project and involves many factors, such as input parameters, topography and resolution (García-Bustamante et al., 2008; Carvalho et al., 2012; Orrell et al., 2001). By comparing simulation and observation results, Shimada et al. (2011) gives a relationship between relative biases and altitudes. Accordingly, we roughly estimated wind speed relative errors at different heights (0-400 m: 50 %, 400-800 m: 40 %, 800-1200 m: 20 %, 1200-1500 m: 10 %, 1500-3000 m: 3 %, > 3000m: 1 %). For pollutant concentration errors, we mainly considered retrieved errors (i.e., the sum of smoothing and noise errors), which play a dominant role in the total error budgets. The errors of $F_i$ and $F_c$ are displayed in Fig. S6 and Fig. S7, respectively."

[Figure]

**Fig. S6.** The vertical distribution of $F_i$ errors at SJZ, WD, NC, and CAMS stations for AECs, NO2, and HCHO.

[Figure]

**Fig. S7.** The $F_c$ errors at SJZ, WD, NC, and CAMS stations for (a) AECs, (b) NO2, and (c) HCHO.

Sect. 2.6: Please make clear how and for what you use this ancillary data. Would it not be better to use reanalysis data for the wind? Why did you perform your own

simulations? What is the time resolution of the data from CNEMC?

Re: Thank you for this advice. We have added some information about usage and time resolution in this Section 2.6 as follows.

"We obtained the surface $NO_2$, $PM_{2.5}$, CO, and $O_3$ concentrations from the CNEMCs with a sampling resolution of 1 h (https://quotsoft.net/air/). We validated the MAX-DOAS measurements by comparing the lowest layer results from the MAX-DOAS observations with the CNEMC data. Using the CO and $O_3$ concentrations, we performed source apportionment of ambient HCHO to identify the contribution ratios of primary and secondary HCHO. Moreover, we depicted a spatiotemporal distribution of $PM_{2.5}$, reflecting surface $PM_{2.5}$, and concentration variations during the transboundary transport process.

The Aerosol Robotic Network (AERONET) is a ground-based aerosol remote sensing observation network jointly established by NASA and LOA-PHOTONS (B.N.Holben et al., 1998). During the measurement period, there were two AERONET sites, Beijing-CAMS (39.933 °N,116.317 °E) and XiangHe (39.754 °N,116.962 °E), adjacent to our MAX-DOAS stations, namely, CAMS (39.95 °N, 116.32 °E) and XH (39.76 °N, 116.98 °E). In this study, we used the Level 1.5 AOD results from these two AERONET sites to validate AODs measured by MAX-DOAS.

We obtained the spatial distributions of $NO_2$ and HCHO from TROPOMI at a spatial resolution of 3.5 × 7.0 km (Veefkind et al., 2012), and the spatial distributions of AOD and dust from Himawari-8 with a 0.5 ×2.0 km spatial resolution and a 10 min temporal resolution (Bessho et al., 2016). Satellite observations helped identify the pollutant transport phenomena because transport tends to cause large-scale continuous distribution of pollutants that can be detected by satellite measurements.

We simulated the wind speed and direction using the Weather Research and Forecasting Model, version 4.0 (WRF 4.0). See Supplementary Sect. S5 which details the model and parameter settings. In terms of wind speeds and pollutant mixing ratios in different layers, we calculated transport fluxes at different heights to reflect the dynamic transport processes of various pollutants. In addition, we used wind-field information to reveal the transport direction at different altitudes.

We calculated the 24-h backward trajectories of the air masses using the Hybrid Single-Particle Lagrangian Integrated Trajectory (HYSPLIT) model (Supplementary Sect. S6). In our study, the 24-h backward trajectories were calculated to investigate the dust origins and pathways that reached the NCP on March 15, 2021."

About reanalysis data, there is no denying that it is widely used in many studies. However, for our study, it has a fatal disadvantage; it doesn't have enough vertical resolutions below 1 km. Taking ERA5 for example, its vertical layers below 1 km include: 1000hPa (0 m), 975hPa (~250 m), 950hPa (~700 m), 925hPa (~800 m), 900hPa (~ 930 m). The MAX-DOAS retrieval is more sensitive to the pollutant below 1 km, which means the vertical distribution of pollutants is more convincing and should be paid more attention to. Comparatively, WRF has 14 layers below 1 km. If we use reanalysis results instead, the number of height grids in Section 3.1 will decrease a lot, which is unfavorable for us to locate the MTL of three pollutants. Moreover, the spatial

resolution of WRF (10×10 km) is better than that of reanalysis data (> 50×50 km).

Using WRF simulation, we could get denser data points.

l.228: add "See" before "Supplement"

l.229: add "of" between "details" and "the model"

Re: Thank you for these suggestions. We have corrected them.

"Supplement Sect. S1 details the model and parameter settings." - > "See Supplementary Sect. S5 which details the model and parameter settings."

l.228ff: This refers to horizontal?

Re: Yes, it refers to horizontal, but "a spatial resolution" is usually used to describe this characteristic of the satellite.

l.231: What is Himawari-8?

Re: This is another type of satellite. Because TROPOMI doesn't contain the spatial distribution information of aerosol optical density and dust that we need, we had to use the data from Himawari-8 here. The detailed introduction to this satellite could be referred to Bessho et al. (2016). We cited this reference at the end of this sentence.

l.237: A correlation of ~0.6 -- 0.7 is not really good. Can you relate the differences you find to the errors you quote in Sect. 2.4?

Re: For lessening the impacts of "abnormal value" on the correlation, we adopt a filtering method here to improve the correlation.

Supplementary Sect. 8: "For lessening the impacts of "abnormal value" caused by occasional extreme conditions, we needed to adopt a method to seek out the abnormal values and filter them out. In a series of data, we firstly found the first quartile (Q1), median (Q2), and the third quartile (Q3), which are the 25th, 50th and 75th percentile of all values from small to large, respectively. The difference between Q1 and Q3 is called interquartile range (IQR) (i.e., IQR = Q3-Q1). The upper limit ($L_{upper}$) and lower limit $L_{lower}$ were defined as Q3 plus IQR, and Q1 minus IQR, respectively (i.e., $L_{upper}$=Q3+IQR, $L_{lower}$=Q1-IQR). The values larger than $L_{upper}$ or lower than $L_{lower}$ were defined as abnormal values, and discarded. After filtering the data, the correlation had increased from 0.615 to 0.752, and 0.671 to 0.74, for aerosol and $NO_2$, respectively."

In addition, we have added error bars according to the surface errors in Sect. 2.4. Some relatively high values tend to have higher errors, which might explain some large deviations.

[Figure]

l.238: You do not just exclude "some" stations, you exclude exactly half of them. While I agree on your criterion, saying "some" is not ok if it's actually 50%. While 10 km seems like "arbitrary", your group of 8 stations has actually a clear division in terms of "closeness" to a monitoring station: 4 of them are closer than 5 km, the other 4 are further away than 15 km. Maybe there is a way to make it more clear that 10 km is actually a good choice, and choosing any number between 5 and 15 would not have changed anything.

Re: Thank you for this suggestion. We calculated the $O_4$ effective optical path to determine the distance threshold, which is around 5 km here. We listed the exact calculation procedure in Supplementary Sect. S7.

Main text: "We calculated the $O_4$ effective optical path as the distance threshold (~ 5 km) to exclude some MAX-DOAS stations from the correlation analysis (Supplementary Sect. S7)."

Supplementary Sect. S7: "In order to determine which stations could be contained in the correlation analysis, we needed to calculate the $O_4$ effective optical path as the distance threshold. The $O_4$ effective optical path can be calculated as follows (Wagner et al., 2004):

$$L_{O_4} = \frac{DSCD_{O_4}}{n_{O_4}}$$

(s14)

where $n_{O_4}$ represents the number density of $O_4$. The $O_4$ concentration equals the quadratic $O_2$ density (Greenblatt et al., 1990), and the $O_2$ concentration nearly keeps constant, which is proportional to atmospheric density $C_{air}$. The calculation formula of the $O_4$ number density is as follows:

$$n_{O_4} = \left(n_{O_2}\right)^2 = (0.20942 \cdot C_{air})^2$$

(s15)

Atmospheric density $C_{air}$ can be directly calculated using the following formula:

$$C_{air} = \frac{N}{V} = \frac{P \cdot N_A}{T \cdot R}$$

(s16)

where N denotes the number of air molecules; V is the volume of air; $N_A$ is Avogadro constant ($6.02 \times 10^{23}$ mol$^{-1}$); R is molar gas constant, with a value of 8.314 J·mol$^{-1}$·K$^{-1}$;

P and T represent the atmospheric pressure and temperature, respectively. Here, we used standard atmospheric pressure and temperature, which are $1.01 \times 10^5$ pa and 273.15 K, respectively. After bringing all the values into the formula, we obtained the value of $C_{air}$, which was $2.69 \times 10^{25}$ molec·m$^{-3}$. Subsequently, we further calculated the $n_{O_4}$ as $3.17 \times 10^{49}$ molec$^2$·m$^{-6}$. In this measurement, the average $O_4$ DCSD was around $1.52 \times 10^{43}$ molec$^2$·cm$^{-5}$. Accordingly, we could calculate the average $O_4$ effective optical path in Eq. s14, which was around 4.79 km. Therefore, in this study, we used 5 km as the distance threshold to exclude some stations from the correlation analysis."

l.242: influenced --> influences

R: Thank you for this advice. We have corrected it.

Figure 2: Can you comment on the vast difference in valid data (factor 3)? Please make those correlation plots use equal aspect ratio. Consider using a different estimate such as Theil Sen, by "eye", your linear fit looks like a bad fit. Have you considered including also AOD comparison to aeronet stations? Do you get vastly different values considering the 4 stations separately? Regarding the in-situ data: I assume that those are available on a very fine time resolution (it is not described in sect. 2.6). It is not clear to me whether you use the closest in time or some time average. Please state. If you use the closest in time: Is this a good choice? How does this actually compare with the time resolution of a scan? Similarly: Do you consider a single station or do you maybe have several stations located in the line of sight from your corresponding instruments? How does the distance relate to the area you probe with your instrument?

Re:

We considered to use aeronet stations for comparison before, but there are only two stations around our selected MAX-DOAS stations (CAMS, XH). Here, we displayed our comparison results in Fig. S3. Red spots represent CAMS, and green ones denote XH. The correlations of 0.816 and 0.941 can verify the reliability of our measurements.

[Figure]

**Fig. S3.** Correlation analysis of 15-min averaged AOD from AERONET and MAX-DOAS at (a) CAMS, and (b) XH stations from January to March, 2021.

Considering the stations separately, there is an acceptable difference of correlation among different stations.

[Figure]

We revise the correlation plots using equal aspect ratio as follows.

[Figure]

We add the time resolution of CNEMC in Sect. 2.6 as follows.

"We obtained the surface $NO_2$, $PM_{2.5}$, CO, and $O_3$ concentrations from the CNEMCs with a sampling resolution of 1 h (https://quotsoft.net/air/)."

Theil Sen estimate is insensitive to abnormal values, and is a good tool to predict the varying tendency in the future. However, we use linear fit to validate the MAX-DOAS measurements instead of predicting the varying tendency of pollutants here. Therefore, using other estimates seems a little unreasonable here. Up to know, the linear least-squares fit has been used in many studies to verify the MAX-DOAS results, comparing them with in situ or other observations (Wang et al., 2017; Lampel et al., 2015; Kumar et al., 2020; Chan et al., 2019; Chan et al., 2018). Few of studies use other estimates.

But we used another way to filter the abnormal values and described the method in Supplementary Sect. 8.

Supplementary Sect. 8: "For lessening the impacts of "abnormal value" caused by occasional extreme conditions, we needed to adopt a method to seek out the abnormal values and filter them out. In a series of data, we firstly found the first quartile (Q1), median (Q2), and the third quartile (Q3), which are the 25th, 50th and 75th percentile of all values from small to large, respectively. The difference between Q1 and Q3 is called interquartile range (IQR) (i.e., IQR = Q3-Q1). The upper limit ($L_{upper}$) and lower limit $L_{lower}$ were defined as Q3 plus IQR, and Q1 minus IQR, respectively (i.e., $L_{upper}$=Q3+IQR, $L_{lower}$=Q1-IQR). The values larger than $L_{upper}$ or lower than $L_{lower}$ were defined as abnormal values, and discarded. After filtering the data, the correlation had increased from 0.615 to 0.752, and 0.671 to 0.74, for aerosol and $NO_2$, respectively."

Given that the sampling resolution of MAX-DOAS is 10-15min and $NO_2$ is easy to get photolyzed, we used the averaged data of 10 min (-5 min-+5 min) as $NO_2$ result at each hour. For aerosol, we thought its lifecycle is quite long, and thus averaged 1 h AEC (-30 min-+30 min) as aerosol results at each hour.

We calculated the average $O_4$ effective optical path (~5 km) as our selection criteria as mentioned above. We chose CNEMC stations if the distance between CNEMC and MAX-DOAS station was lower than 5 km. After checking, we found that there is only

one CNEMC station within the effective optical path around each instrument.

l.249 ff: I wonder if this paragraph should not better go to the introduction.

Re: Thank you for this advice. However, in the introduction, we just gave a brief introduction that transport had a large impact on the environment of megacities and it was not specific enough. In Section 3.1, we mainly focused on the regional transport occurring within the JJJ region, and thus the works listed in this paragraph were all about regional transport in the JJJ region. It is unsuitable to move this paragraph to the introduction because this paragraph is less associated with Section 3.2 and 3.3, since we talked about different types of transport in different sections.

l.252: "simulation" --> "simulations"

Re: Thank you for this advice. We have corrected it.

l.257: "According to the TROPOMI results" is a weird formulation. Please reformulate. Also, you could rever your statements (the ones connected with "whereas") and it would still be true. I do not see the difference here which you clearly want to highlight using "whereas".

Re: "According to the TROPOMI results, we found that that $NO_2$ was continuously distributed between SJZ and WD, whereas a HCHO distribution belt connected NC with CAMS on February 5, 2021 (Fig. S2)." -> "The TROPOMI results indicated that $NO_2$ was homogeneously distributed between SJZ and WD and that, on February 5, 2021, the HCHO distribution belt connected NC with CAMS (Fig. S4)."

Figure S2: This is double, and Figure S1 is missing in the supplement.

Re: Thank you for this advice. We have corrected it.

l.260: For me it looks much more W --> E than SW --> NE. That is the impression I get from Fig. S3.

Re: Admittedly, there are more wind directions of W→E than SW→NE, as displayed in Fig. S3. However, in Section 3.1, we only focused on the transport phenomena along the pathway among the four stations. Along this pathway, we could find many westerly winds gradually converted into southwesterly wind. For example, on the left of SJZ, most winds blew from west to east. When you looked at the area between WD and NC, many winds gradually turned into southwesterly winds. This might be affected by the topography of the JJJ region. There are many high mountains on the left of these four stations (Fig. 1). It favors the formation of southwest-northeast transport within the JJJ region.

Most of westerly winds were located at high-altitude areas (colored by yellow and brown in Fig. 1). They contributed less to this southwest-to-northeast transport phenomena occurring at the plain, and thus we didn't discuss them here.

Fig. S3: Please enhance the arrow thickness. This plot is not yet too bad. I can see the direction of arrows, but I cannot see the orientation. I do get an impression of the wind speed here, for later wind-field plots this is not the case. This one is still ok in this respect.

Re: Thank you for this comment. We have revised Fig. S3. We have enhanced the arrow thickness and changed the arrows' colors according to wind speeds, making orientation and direction more obvious.

[Figure]

**Fig. S5.** The wind field in the (a) 0–20, (b) 200-400, (c) 400-600, and (d) 600–800 m layers simulated by WRF at 13:30 on February 5, 2021. The arrows represent the wind direction, and their lengths and colours stand for the wind speed.

l.266: "also": Where did you additionally identify this?

Re: "We also identified this southwest–northeast regional transport process in the temporal variations in the vertical distributions of aerosol, NO2, and HCHO (Fig. 3)"

-> "Figure 3 presents the temporal variations in the vertical distributions of aerosols, NO$_2$, and HCHO during this regional transport."

l.267: You just refer to Fig. 3, and I do not see the previous statement in Fig. 3. It is maybe intended to say that you will further explain this in the following paragraph, but this is not clear.

R: To make the meanings clearer, we firstly followed Fig.3 and introduced the spatiotemporal distribution of dominant pollutants at each station as follows.

"Figure 3 presents the temporal variations in the vertical distributions of aerosols, NO$_2$, and HCHO during this regional transport. At the CAMS and NC stations, the aerosol,

NO$_2$, and HCHO concentrations were consistently high near the surface, primarily because of the heavy traffic flow and dense factory emissions in Beijing (Zhang et al., 2016; Li et al., 2017). Previous studies have suggested that urban air pollution in Beijing is dominated by a combination of coal burning and vehicle emissions, which results in severe particulate pollution (Wang and Hao, 2012; Wu et al., 2011). At SJZ, the NO$_2$ concentration was high (~ 12 ppb) in the morning and late afternoon, whereas the concentration was lowest (~ 6 ppb) near noon, which is explained by the morning and evening rush hour. Comparatively, the overall AEC and NO$_2$ levels were relatively low at the WD station, whereas a continuous high-value HCHO distribution (> 2 ppb) occurred at 0-1500 m between 11:00 and 16:00. This occurred because the WD station is located in a farm field with less traffic flow and high vegetation coverage; therefore, large amounts of HCHO are directly emitted by biogenic sources and secondarily produced by natural and anthropogenic volatile organic compound (VOC) photolysis (Wang et al., 2016; Wu et al., 2017a)."

l.268: How do you deal with the different layer distribution between retrieval grid and model layers?

Re: We have added some descriptions about transport flux calculation details in Supplementary materials.

Supplementary Sect. S3: "Due to the different height grids from the retrieval and the model wind, we needed to design a unified height grid. Thus, we divided 3.1 km into 13 layers: 0–100, 100–200, 200–300, 300–400, 400–600, 600–800, 800–1000, 1000–1200, 1200–1400, 1400–1600, 1600–2000, 2000–3000, 3000–3100 m. We averaged the wind speeds and pollutant concentrations at each layer to represent $W_i$ and $C_i$ in layer i, respectively. In addition, the time resolution of wind simulation is 1 hour, whereas that of the vertical profile is 15 minutes. In order to unify the time resolution, we averaged vertical profiles of an hour to calculate transport flux. Given that the results of last 15 minutes in each hour fit the situation of next hour better, we averaged the results from -15 min to +45 min as the hourly vertical profiles. For instance, we calculated the average from 9:45 to 10:45 to represent the vertical profile of 10:00."

l.266 -- 303: This description jumps all the time between Figures 3 and Figures 4. It is impossible to follow and I cannot see the statements made in this paragraph (it is also far too long) confirmed in the data, or at least, in the way the data is presented. As already suggested in my pre-review: try explaining your points to a not-involved co-worker and take notes of additional information/ plots that were needed to present in order to convince your co-worker that the statements you make are supported by your data.

R: Thank you for this comment. Our descriptions jump between Fig. 3 and Fig. 4, which easily cause misunderstanding. Therefore, we have reorganized the paragraph. We firstly followed Fig.3 and introduced the spatiotemporal distribution of dominant pollutants at each station. Then, we mainly followed Fig.4 and discussed the MTL of aerosol, NO2, and HCHO. Meanwhile, we deleted many speculative conclusions (e.g., secondary aerosol generation, aerosol transport among stations) to make paragraphs more understandable and emphasize the main points. The revised paragraphs are listed

as follows.

[revised manuscript text omitted]

l.271: The wind field you provide is from 13:30 and it shows W --> E at SJZ not SW --> NE.

Re: According to the calculation in Section 2.5, the $F_i$ and $F_c$ both point to the transport flux projection in the southwest-northeast pathway. To avoid misunderstanding, we have changed the sentence here.

"As shown in Fig. 4, the positive $Flux_i$ indicated that $NO_2$, HCHO, and aerosols were all transported from southwest to northeast at the four stations." -> "As shown in Fig. 4, a positive $F_i$ indicates that all transport flux projections in the southwest-northeast pathway are from southwest to northeast at the four stations."

l.272: high AEC over 300 m at WD around 11:00: I do not see this in the plot. Before 11, it seems higher at the surface. What is the distance here? Does it at all fit together

with the wind speed and the time distance here?

l.273: I can maybe see an increase at 11, but at 12? Also again at 13 h but then to > 0.3 / km. Please check.

l.274: "at 200--800 m at CAMS at 12:15", for me it looks more as this is a raising aerosol layer than a transported one. How do you exclude this?

Response to l.272-274: We had thought to discuss aerosol transport among these four stations in Line 270-274. However, it may lack direct evidence as you said. Only with existing data, it is not convincing to discuss whether the distance fits together with the wind speed and the time distance. This is because the distance among SJZ, WD, and NC is too long (> 40 km), and wind speeds would experience many uncertain changes during this process. Maybe these arguments should be considered as hypotheses based on the observations instead of conclusions evidently inferred from the measurements. Given that the main points of Section 3.1 are to discuss the MTL of various pollutants, we decide to delete such descriptions.

l.275: "After 16:00", is this really 4pm? For me it looks as 5pm? (for CAMS, but it isn't even clear with respect to which station you make this comment).

R: We have changed the demonstration.

"After 16:00, the high-extinction air mass shifted MTL from to 300–1000 m toward the surface at SJZ, with the AEC gradually exceeding 0.5 km$^{-1}$ (Fig. 3)" -> "After 16:00, the shift in the high-AEC air mass caused the transport fluxes in the lower layers (100–200 m) to increase to > 1.1 and ~ 2 km$^{-1}$·m·s$^{-1}$ for the CAMS and SJZ stations, respectively."

l.276 "from to 300 --1000 m"?

R: "After 16:00, the high-extinction air mass shifted MTL from to 300–1000 m toward the surface at SJZ, with the AEC gradually exceeding 0.5 km$^{-1}$ (Fig. 3)" -> "In the late afternoon, aerosols gradually accumulated towards the surface, and triggered a variation in the distribution of $F_i$. After 16:00, the shift in the high-AEC air mass caused the transport fluxes in the lower layers (100–200 m) to increase to > 1.1 and ~ 2 km$^{-1}$·m·s$^{-1}$ for the CAMS and SJZ stations, respectively."

l.277: "similar": similar in which way?

R: The description like "similar" is vague. We have adopted more detailed demonstrations here as follows.

"In the late afternoon, aerosols gradually accumulated towards the surface, and triggered a variation in the distribution of $F_i$. After 16:00, the shift in the high-AEC air mass caused the transport fluxes in the lower layers (100–200 m) to increase to > 1.1 and ~ 2 km$^{-1}$·m·s$^{-1}$ for the CAMS and SJZ stations, respectively. Surface aerosol accumulation is closely linked to the collapse of the mixing layer and formation of a stable nocturnal boundary layer (Ding et al., 2008; Ran et al., 2016). Remarkably, high-altitude aerosol air masses began to mix with near-surface aerosols after 14:00 at the NC station (Fig. 3), triggering a variation in the MTL (Fig. 4). This might be explained by enhanced vertical mixing due to the heating of the surface during the course of the day (Castellanos et al., 2011; Wang et al., 2019). Generally, the MTL of aerosols was situated at 400–800 m during the daytime, where variations in the boundary layer and

increased vertical mixing can influence the MTL."

l.280: Surely the fact that the emissions are from traffic cannot be seen in Fig.4?
l.281: "because of the heavy traffic flow in Beijing" how do you know this from Figure 3?
l.283: What is meant by "variation in high-concentration"?
l.283/284: How does it agree "well with the shift in the corresponding MTLSs (Fig.4)"? I don't even know what I should be searching for in Fig.4.
Responses to l.280-283: Given that the description structure here is confusing, we have reorganized this part about the MTL of $NO_2$ as follows.
"In contrast to aerosols, we found that a high-value $NO_2$ $F_i$ frequently occurred in the 0–400 m layer. Except that $F_i$ reached the highest level of ~ 50 ppb·m·s$^{-1}$ in the 400–600 m layer at 16:00 at the SJZ station, the other highest $F_i$ all occurred below 400 m at any station and at any time. This indicated that the MTL of $NO_2$ was 0–400 m. Near-surface $NO_2$ emission sources (e.g., vehicle and factory emissions) might be the main reason for this phenomenon."

l.284: When and where do you show this, you haven't yet talked about HCHO or have you?
R: This means that we will further explain this in the following paragraph, but it is not clear and easily make readers confusing. Thus, we use another sentence to start the discussion about the MTL of HCHO.
"Compared with aerosols and $NO_2$, we found that high-value HCHO $F_i$ extended to higher altitudes. Taking CAMS as an example, …"

l.285: Can you actually exclude that this is a retrieval artifact?
l.285: But it is actually intensifying, should it not be diluting if it is transported?
l.287: It is unclear to me how the satellite can capture a transport and not simply a presence.
Responses to l.285-287: In Line 285-287, we had thought to discuss HCHO short-distance transport between NC and CAMS. However, it may lack more direct evidence. Besides, it is not convincing to discuss whether the distance fits together with the wind speed and the time distance, since wind speeds would experience many uncertain changes during this process. Maybe the statement should be considered as hypotheses based on the observations instead of conclusions evidently inferred from the measurements. Given that the main point of this part is to discuss the MTL of HCHO, we decide to delete Line 285-287.

l.288 "much more likely" sounds like you made some statistical analysis here. Where is it? What is this statement based on?
R: To avoid misunderstanding, we have changed the demonstration here.
"The MTL of HCHO ranged from 400 to 1400 m, and was much more likely to extend to higher altitudes than that of aerosols or $NO_2$. -> "Compared with aerosols and $NO_2$, we found that high-value HCHO $F_i$ extended to higher altitudes."

l.289: What is "strong"? By which measures is it "strong"?

R: "Taking CAMS as an example, we found a strong HCHO $Flux_i$ at 600–1400 m, which lasted 4 h (10:00–14:00) (Fig. 4)" -> "Taking CAMS as an example, we found the strongest HCHO $F_i$ constantly emerging at 1000–1200 m from 8:00 to 13:00, and averaging 9.18 ppb·m·s$^{-1}$. During the same period, surface HCHO $F_i$ only averaged 6.44 ppb·m·s$^{-1}$. However, at the CAMS station, the surface HCHO concentration was much higher than that of the 1000–1200 m layer between 8:00 and 13:00 (Fig. 3), proving that high-altitude transport contributed more to overall HCHO transport."

l.290: "Likewise" likewise what? "increased to" increased from what or from which value?

R: "Likewise, the MTL of HCHO increased to over 800 m between 11:00 and 16:00 at the WD station." -> "After 10:00, we found that the highest HCHO $F_i$ gradually increased from ~ 8 to ~ 20 ppb·m·s$^{-1}$ at WD, with the MTL of HCHO ranging from 400 to 1000 m."

l.291ff: I would start the whole paragraph with this sentence here: "High HCHO concentrations tend to appear at higher altitudes than those of NO$_2$ and aerosols. ...". This is one of the few things in the description which is clearly visible and which has a whole set of sound explanations following the statement.

R: Thank you for your praise. We would revise the whole passage according to the structure of this part.

l.297: Can you exclude a retrieval artifact here?

l.298: If it is at the same time, how can it "follow"?

l.299: If I am not mistaken, then you argue on line 274 that the high AEC at CAMS is due to transport? Now you attribute it to formation due to NO$_2$?

l.300: Or you can use the argument from line 272 and argue it is transported from SJZ. How do you decide when you attribute it to transport and when you attribute it to secondary aerosols? This is not clear.

Responses to l.297-303: We had thought to discuss that secondary aerosol generation accompanied this regional transport process in Line 297-303. However, it lacks confirmation. We don't have any model calculations or other measurements here. It could only be considered as one of speculations in our observations. Considering that it is less associated with the main points of Section 3.1 (the MTL of various pollutants), we have deleted these demonstrations.

Figure 4: This relates to my comment to Eq.2: I think you should really use concentrations and not mixing ratios. Apart from really relating to the physical quantity (molecules instead of molecules fraction) it has the dimensions of a classical flux. I am not sure how I should treat a flux with dimensions m/s. It is not clear to me here how you treated missing data. Compared to Fig.3, there seem to be more data points "filled". E.g.: NO$_2$ at SJZ shows many gaps inn Fig. 3 but not in Fig.4.

Re: Thank you for this comment. Honestly speaking, we had considered using concentrations before. However, converting equations are related to temperature (T) and pressure (P) as follows.

$$X(ppb) = \frac{C(\mu g / m^3) \bullet M_g(g/mol)}{V_m(L/mol)}$$

$$V_m = \frac{R \bullet T}{P}$$

There are large differences at temperatures and pressures of different altitudes. Unfortunately, we don't have any profiles of T or P at these four stations. If we adopt unified standard atmospheric pressure or temperature, it will introduce larger uncertainties, since the differences of T and P at different layers cannot be neglected. But we think that using mixing ratios can represent the density of pollutants at corresponding heights as well. In addition, we just use the highest transport flux to roughly determine the MTL, and using other units has little impacts on the main conclusions. Therefore, we didn't use concentrations here.

It seems more data points in Fig. 4, but actually it is not. This is because Fig.3 and Fig. 4 have different time resolutions. The Fig.3 displays one piece of vertical distribution of pollutant per 15 minutes. For example, in the HCHO results of SJZ, we could see four different HCHO vertical distributions from 16:00 to 17:00. Comparatively, the Fig.4 only has one result per an hour, since wind simulation's time resolution is 1 hour. To get enough data points, we got the vertical profile at each hour by averaging the data in a certain period. For example, if we want to get the vertical distribution at 10:00, we average the results from 9:45 to 10:45, since we considered the data at 9:45-10:00 matched better with the situation at 10:00 and results from 10:45 to 11:00 fit the situation at 11:00 better.

l.313ff: Could this also be a bias introduced by having data at different times of the day? E.g.: CAMS has no $NO_2$ data from 8 --1, a time where values for e.g. WD are especially low.

Re: Thank you for this comment. But we find that the impact caused by missing data can be neglected and don't affect the main conclusions. For example, during 12:00-16:00, there is no missing $NO_2$ column transport flux at any station. We can find that the $NO_2$ result of CAMS (purple color) is higher than that of WD (yellow color) at each hour. For verification, we also do a comparison among averaged column transport fluxes at the selected time period, during which there is no missing data, as follows.

[Figure]

11:00-16:00 (aerosol): SJZ ($3.30 \times 10^3$ km$^{-1}$·m$^2$·s$^{-1}$) > NC ($2.85 \times 10^3$ km$^{-1}$·m$^2$·s$^{-1}$) > CAMS ($2.45 \times 10^3$ km$^{-1}$·m$^2$·s$^{-1}$) > WD ($1.82 \times 10^3$ km$^{-1}$·m$^2$·s$^{-1}$)

12:00-16:00 (NO$_2$): SJZ ($6.06 \times 10^4$ ppb·m$^2$·s$^{-1}$), NC ($4.65 \times 10^4$ ppb·m$^2$·s$^{-1}$), CAMS ($6.10 \times 10^4$ ppb·m$^2$·s$^{-1}$), WD ($2.47 \times 10^4$ ppb·m$^2$·s$^{-1}$).

(11:00-12:00)+(14:00-16:00) (HCHO): SJZ ($2.09 \times 10^4$ ppb·m$^2$·s$^{-1}$), NC ($2.10 \times 10^4$ ppb·m$^2$·s$^{-1}$), CAMS ($1.99 \times 10^4$ ppb·m$^2$·s$^{-1}$), and WD ($3.77 \times 10^4$ ppb·m$^2$·s$^{-1}$). The HCHO column transport flux at WD is the highest, and substantially higher than the other three stations.

The order doesn't change, and thus we know that the impact caused by missing data can be neglected.

The main purpose of comparing different column transport fluxes among four stations here is to state that "In terms of the relative locations of stations (Fig. 1) and the $F_c$ results, we considered that SJZ was an important source of transported aerosol and NO$_2$, and WD was one of the main HCHO sources during this regional transport, which largely affected the air quality of cities along the southwest-northeast transport pathway."

l.318ff: This should be moved to conclusions. However, since it is unclear to me how you discriminate btw transport, local primary production and rising, and secondary production, I do not see your statement supported.

Re: Thank you for this comment. We have deleted this sentence (line 318ff). There are many speculative arguments making Section 3.1 confusing and less convincing. Therefore, we have deleted many speculative conclusions (e.g., secondary aerosol generation, aerosol transport among stations) to make paragraphs more understandable and emphasize the main points (the MTL of aerosol, NO$_2$, and HCHO).

l.323: which satellite? what are the satellite results that reveal things?

Re: "The satellite" referred to the Himawari-8, which can monitor the spatial distribution of aerosols and dust.

The satellite result was displayed in Fig. S5. In the Figure, we could find the NCP was severely affected by the dust storm (represented by pink).

To make it clearer, we have changed some words.

"The satellite results…" -> "The Himawari-8 observations…"

"**Fig. S5.** A severe dust storm invaded northern China at (a) 8:00 and (b) 14:00 on March 15, 2021." -> "**Fig. S8.** The Himawari-8 observations: a severe dust storm invaded northern China at (a) 8:00 and (b) 14:00 on March 15, 2021. The dashed black contour line indicates the NCP region." (Supplementary materials)

Fig. S5: I am very lost here. Can you please indicate the regions introduced in Fig.1? The legend is not readable. What is the source here? A description is missing. Also: why is the source of the dust important? In which way is this relevant for the rest of the analysis? What was the main point of doing this back trajectory analysis? This is not clear to me.

Re: To make it clearer, we have revised the Fig. S5.

To make the legend more readable, we enlarged the legend and fonts. Actually, in the legend, we just needed to focus on the pink ones (represent dust storm), other colors stood for different types of clouds. In addition, we have added the dashed black contour line to indicate the NCP region.

[Figure]

**Fig. S8.** The Himawari-8 observations: a severe dust storm invaded northern China at (a) 8:00 and (b) 14:00 on March 15, 2021. The dashed black contour line indicates the NCP region.

The main point of back trajectory analysis was to find the source of this dust storm. In Fig. S8 (satellite results), we could only see the affected area, while the source was not obvious. In Fig. S9 (back trajectory analysis), we could clearly know that the dust storm

was sourced from Mongolia. What's more, this result further confirmed that the four stations we selected were all influenced by this dust storm. Based on this condition, we could only discuss the effects of dust transport on four stations' local environment.

This source information is not so important to the following analysis, so we put Fig. S8 and Fig. S9 in the Supplementary materials. But this dust storm is a very rare extraordinarily dust storm, we think it is necessary to clarify its source. Different from other pollutants (e.g., $NO_2$, aerosol), tracing dust storm is not so complicated.

Figure 5: Why such an inconsistent choice of time? (minimize missing data I presume. But since you have many more months of data, don't you have better data than this?) Maybe use a different color in the label for the dusty day? Maybe use a separate x-axis scale for the different days (although I also do of course see the point of using the same and if you had used the same, I would maybe say use the same... always difficult. It is just that you comment on some details of aerosol profiles on 2021.03.06 that are hardly visible in the plot. Maybe you could add an insert?

Re: Thank you for this comment. As you said, we had to choose inconsistent time periods to minimize missing data and make comparison more reasonable.

Different from frequent other pollutants (e.g., $NO_2$, aerosol) transport, the dust storm events are quite rare in China now. Such a severe dust storm, like this, dates back to 2015, and we don't have data during that period. Both the World Meteorological Organization (https://public.wmo.int/ en/media/news/severe-sand-and-dust-storm-hits-asia, last access: 6 November 2021) and CNN (https://edition.cnn.com/2021/03/15/asia/beijing-sandstorm-decade-intl-hnk,last access: 6 November 2021) described the 3.15 sand and dust storm event as the biggest one in almost a decade. What's more, this dust storm event only lasted one day, so we had no choice but to use the data on March 15, 2021.

We had considered to use a different color in the label for the dusty day before. However, there had been four colors to represent four different stations. If we use a different color for the dusty day, the total number of colors would increase to 8, which might cause readers' confusion. Compared with that, we thought adding exact dates at the left of Fig. 5 to distinguish the results might be better and more understandable.

Using the same x-axis scale could make the concentration variation on dusty day more obvious.

We have revised Fig.5 as follows. The detailed concentration distribution of Fig.5 is listed in Table. S3-S5.

[Figure]

**Figure 6.** The daily averaged vertical profiles of AECs (left), NO₂ (middle), and HCHO (right) at DY, NC, SJZ, and XH stations during two clean days (March 6 and 22, 2021) and one dusty day (March 15, 2021). The upper annotation in each subplot represents their corresponding time periods, during which each station generated the greatest amount of data as possible.

l.334: I agree on the AEC part, but for HCHO, data is mainly missing except for DY?

Re: Yes. On one hand, outages of the instrument result in some periods of failure, and make some stations lack SCDs during the certain period. On the other hand, some results cannot meet the selection criteria (DOF > 1.0, relative error < 50%), and are filtered out. If we don't filter the data, there will be more profiles, but many results are obviously unreasonable.

l.334: "while" I do not see any discrepancy here with the statement before, so why "while"?

Re: "On the dust storm day, the AEC and HCHO concentrations substantially increased, especially near the surface (Fig. S7 and S9), while NO₂ concentrations also increased a lot at SJZ and DY (Fig. S8)." -> "On the dust storm day, the AEC and HCHO concentrations substantially increased, especially near the surface (Fig. S10 and S12). Meanwhile, NO₂ concentrations also increased significantly in SJZ and Dongying (DY) (Fig. S11)."

l.335/336: Fig. 5 does not show a classification, it just shows the profiles but no classification.

Re: "As shown in Fig. 5, ..." -> "As shown in Fig. 6, the vertical profiles at all four stations displayed peaks in the high layers on clean days."

As you said, some little differences, like the difference between Gaussian-shaped vertical profile with high surface concentration and two-peak shape, could be caused by other factors (e.g., an imperfect aerosol retrieval) instead of pollutant vertical distribution itself. Instead, we find a better method, which can summarize the varying features of vertical profiles on the dusty day better—the movement of high-altitude peaks.

l.337: It's hard to see this because the curves are on top of each other: see comment Figure 5. Maybe make an insert?

Re: Thank you for this advice. We have inserted an enlarged partial picture here.

l.339: Do you really consider these profiles very different? If you consider the AKs, aren't they rather almost the same profile? In fact, we cannot know, you don't show a single AK.

Re: Thank you for your comment. We feel that we paid much attention to the differences between the vertical profile shapes among different stations before. For example, we thought the vertical profile shapes could be classified into multiple-peak and Gaussian shapes on clean days. However, as you said, some little differences, like the difference between Gaussian-shaped vertical profile with high surface concentration and two-peak shape, could be caused by other factors instead of pollutant vertical distribution itself. In addition, the different vertical profile shapes in different stations are not our main points here. Thus, such descriptions might not be reasonable. The main point here is to discuss the profile variation caused by the dust. In other words, we should discuss the difference of vertical profile on the dusty day and clean days. Therefore, we have given up emphasizing the profile shape differences among different stations. Instead, we find a better method, which can summarize the varying features of vertical profiles on the dusty day better—the movement of high-altitude peaks.

For clean days, we demonstrate like "As shown in Fig. 6, the vertical profiles at all four stations displayed peaks in the high layers on clean days."

When it is on the dusty day, we describe like "During dusty periods, aerosol, $NO_2$, and HCHO concentrations notably increased, particularly near the surface, at most stations (Fig. 6). The high-layer peaks dropped to lower altitudes and even disappeared (Tables S3-S5). For example, on the dusty day, we found that high-altitude peaks disappeared and the only peak emerged at the surface for aerosol, $NO_2$, and HCHO vertical profiles at the NC, XH, and SJZ stations (Tables S3-S5). Meanwhile, the $NO_2$ and HCHO concentration peaks both dropped to the 100 m layer at the DY station (Tables S4, S5). These changes might trigger variations in the vertical profile shapes and convert many vertical profile shapes (e.g., AEC vertical profiles at all stations) into an exponential shape (Fig. 6)."

In summary, we mainly focus on the variation of high-altitude peaks between the dusty day and clean days, not paying much attention to different vertical profile shapes at different stations any more.

l.342: This was not shown. But maybe you mean that you are going to show this in this paragraph? Not clear...

Re: In this paragraph, we wrote in the order of phenomenon-explanation. In line 336-340, we demonstrated high-altitude peaks at four stations on clean days. We have

changed descriptions as follows.

"As discussed in Section 3.1, the prevalent regional transport strongly influenced the air quality in the JJJ region, corresponding to the occurrence of high-altitude peaks." -> "In the JJJ region, we selected three stations (i.e., NC, SJZ, and XH) and found that high-altitude peaks occurred at 300–500 m for aerosols, $NO_2$, and HCHO (Table S3-S5). For example, the AEC vertical distribution at SJZ displayed a high peak of 0.75 $km^{-1}$ at 500 m on March 6 (Table S3). At the NC station, the AEC vertical distribution exhibited the only peak (0.70 $km^{-1}$) at 300 m. This may be explained by the prevalent regional transport, which strongly influences the air quality in the JJJ region (Ge et al., 2018; Wu et al., 2017; Xiang et al., 2021). As discussed in Section 3.1, high-altitude transport phenomena trigger high-values of pollutant distribution in the high layers. The surface peaks on clean days were possibly caused by dense traffic and factory emissions in the JJJ region (Qi et al., 2017; Zhu et al., 2018; Yang et al., 2018; Han et al., 2020)." In the sentence above, we attributed the high-altitude peaks at NC, XH, and SJZ on clean days to regional transport (as discussed in Section 3.1). Then, we started to explain the high-altitude peaks at DY, in "In contrast to the JJJ region, the DY station is…"

l.345: "On dusty day", maybe on "the" dusty day?
l.350: what's the dof? How does it compare to the a priori?
Re: We don't mention the DOF here. We have supplemented some information about DOF in Section 2.3 as follows. It has no relationship with a priori.
"The sum of the diagonal elements in the averaging kernel matrix is the degrees of freedom (DOF), which denotes the number of independent pieces of information that can be measured."

l.352: really? Seems rather flat if not surface peaked? It seems as if the surface concentration doubled?
Re: Our meaning is to emphasize that the vertical profile shape still kept a Gaussian shape here, even if concentration increased a lot. But we have given up discussing different varying features among different stations here, because the variation of high-altitude peak can summarize the changes at vertical profile better and apply to all the stations.
"During dusty periods, aerosol, $NO_2$, and HCHO concentrations notably increased, particularly near the surface, at most stations (Fig. 6). The high-layer peaks dropped to lower altitudes and even disappeared (Tables S3-S5). For example, on the dusty day, we found that high-altitude peaks disappeared and the only peak emerged at the surface for aerosol, $NO_2$, and HCHO vertical profiles at the NC, XH, and SJZ stations (Tables S3-S5). Meanwhile, the $NO_2$ and HCHO concentration peaks both dropped to the 100 m layer at the DY station (Tables S4, S5)."
l.355f: I don't follow your argumentation.
Re: This statement is mainly about why $NO_2$ and HCHO peaks only dropped to 100 m instead of the surface at the DY station. We have deleted this argumentation here.

l.358: I think this should refer to Fig. S8. at NC, there is hardly any valid data. Why not directly refer to Fig. 5= $NO_2$ (the few valid data that is there) at the same times look rather similar to the same times at the 22nd, so the big difference is not between the dusty day and the clean days, but between the clean day and the dusty day together with the clean day after.

l.358: This is not true any longer if you take March 22 as the benchmark. In a later Sect. you make a more detailed comparison, comparing separately to both "benchmark" days. I'd rather keep the more detailed one in the later section. And completely skip this description here.

l.360: XH seems to actually show higher SC in the evening so I'm not sure I agree with your statement here.

Responses to l358-360: Given that we mainly discuss the dust can inhibit dissipation, aggravate pollution accumulation, and trigger a variation in high-altitude peaks, we have followed your suggestions and deleted the statements here. The $NO_2$ variation caused by dust will be discussed in detail in the following.

l.361: what is meant by "optical variation"?

Figure 6: I would show this together with the histogram from the supplements. In the box plots, the division line you chose seems fairly arbitrary, especially because you would also need to explain why you consider the mean and not the median. I find the distinction more clear directly from the histograms.

l.368: It is not clear what the "optical signal intensity" is. The total integrated spectrum? Please specify.

Fig. S10: It is not clear to me at which days this is. All days?

l.372ff: I do not follow this definition (later it makes total sense, but the description is weird)

Responses to l.361-372:

According to the suggestions of Referee 3, we decided to remove the discussion based on the classification of optical intensity. On one hand, MAX-DOAS instruments are usually not radiometrically calibrated. The "optical signal intensity" points to the averaged optical signal intensity received by spectrometers. Even if the spectrometers are of the same type, the signals from different instruments cannot be directly compared to each other since they depend on many parameters, such as the gain of the amplifier, as well as on the adjustment of the telescope optics, the length of the fibre bundle, etc. On the other hand, this classification is not an indispensable part of the following discussions. As you said, "the text is already very acronym heavy" and this will make readers confused easily. Therefore, we give up classifying the four stations here.

We have changed the descriptions of L368-374 as follows, and removed Fig. 6 and Fig. S10.

"In addition to aggravating pollutant accumulation, transported dust can affect the environment and pollutant concentrations in other ways. To quantitatively demonstrate the impacts of dust on various pollutants, we introduced growth rate in the comparative analysis (Supplementary Sect. S9). For convenience, we defined the comparison of the results of March 6 and 15, 2021, as precomparison (PRE), and we defined the

comparison between March 15 and 22, 2021, as postcomparison (POST)."

Instead, we depict a relative difference in optical intensity between dusty day and clean days as Fig. 13. This Figure is used to describe the impacts of dust storm on light intensity, due to the light attenuation of dust and enhanced aerosols.

[Figure]

**Fig. S13.** The difference of optical signal intensities received by MAX-DOAS between dusty day and clean days. (a) PRE: Intensity (March, 15) – Intensity (March, 6); (b) POST: Intensity (March, 15) – Intensity (March, 22).

l.377: Where are the "proofs" of this statement?

Re: This is just one of our speculations here. We have adopted a more speculative method to describe our assumptions as follows.

"As described above, the advent of dust results in unfavorable meteorological conditions (e.g., decreased PBL height and more stable PBL) and enhances local pollutant accumulation, which boosts aerosol increase in the lower layer. Moreover, such meteorological conditions are always accompanied by higher levels of RH in the lower PBL (Huang et al., 2020), creating good conditions for enhanced secondary production of aerosols through aqueous-phase and heterogeneous chemical reactions (Ravishankara, 1997; Mcmurry and Wilson, 1983). These two factors could be the main reasons for the enhanced aerosol concentrations on the dusty day."

l.386: "with" --> "to"

Re: Thank you for this advice. We have corrected this one.

Fig. S11: Are these means? medians? How do you take into account that the distribution of time during day (in terms of valid measurements) is different? Put labels: BG and DG (also, I think the text is already very acronym heavy, I would just use BRIGHT and DARK).

Re: The Fig. S11 presents averaged AEC vertical distribution at a certain period. In the comparing process, we also considered the different distribution of valid measurements at different stations and on different dates. In order to make sure each station had as much data as possible in each comparison, we selected different time periods for different stations (i.e., NC, SJZ, XH, and DY), pollutants (i.e., aerosol, $NO_2$, and HCHO), and comparison groups (i.e., PRE and POST).

For example, we selected 10:30-16:15 to compare AEC vertical profiles on March, 15 with that on March, 6. And we chose 10:30-17:00 to compare AEC vertical profiles on March, 15 with that on March, 22. If we display different time periods in the Figure, this may confuse readers. Therefore, we don't present selected time periods here.

We have given up classifying the four stations into two groups based on light intensity, and thus labels are not needed any more. To some extent, this can help reduce the number of acronyms. We have incorporated Fig. S11 and Fig. 7 into one Figure (Fig. 7) to present the impacts of dust on various pollutants and environment.

l.386: I thought it was just argued that there was a considerable difference also for AEC?

Re: The AEC increased substantially at all the stations, while $NO_2$ increased at SJZ and DY, and decreased at NC and XH. We have changed the descriptions, which can help avoid misunderstanding.

"As shown in Fig. 7A, the AEC noticeably increases at all stations on the dusty day, especially below 0.5 km."

"In contrast to aerosols, we observed large differences in $NO_2$ growth rates (Fig. 7B). At SJZ and DY, $NO_2$ concentrations exhibited a substantially increasing trend. The surface growth rates at the SJZ and DY stations were 6.97 and 17.50 in PRE, and 2.06 and 6.50 in POST, respectively (Fig. 7B (c-1, c-2, d-1, d-2)). In contrast, we observed decreased $NO_2$ concentrations at almost every height at stations NC and XH. The near-surface growth rates at NC and XH were -0.81 and -0.76 in the PRE, and -0.30 and -0.59 in the POST, respectively (Fig. 7B (a-1, a-2, b-1, b-2)). This indicated that dust and aerosols have different effects on the trace gas concentration."

l.386ff/ Fig.7: I do not see the added value of Fig.7 over Fig.S8. Figure S8 also clearly shows that there is a lack of data to compare at NC and that, using the same times than XH should show actually higher SC. (my previous argument: can you really compare the different stations if you compare at vastly different times of day?)

Re: In the comparing process, we have considered the different distribution of valid measurements at different stations and on different dates. In order to make sure each station had as much data as possible in each comparison, we selected different time periods for different stations (i.e., NC, SJZ, XH, and DY), pollutants (i.e., aerosol, $NO_2$, and HCHO), and comparison groups (i.e., PRE and POST).

For example, we selected 10:30-16:15 to compare AEC vertical profiles on March, 15 with that on March, 6. And we chose 10:30-17:00 to compare AEC vertical profiles on March, 15 with that on March, 22. If we display different time periods in the Figure, each comparison will contain a certain time period and this may confuse readers. Therefore, we don't present selected time periods here.

l.386ff: I like this paragraph (in fact both paragraphs) in general and I find it easier to follow than all the rest of the manuscript (with the exception of line 291ff which is also very sound). Given that large parts of HCHO are not at the surface, how good is it to use tracers from surface measurements? Of course I am aware that you have to use what you have. But maybe you can discuss a bit more the effect on e.g. the correlation, what part of the "unexplained" HCHO could be attributed to a location mismatch of tracers and HCHO? How does the fit translate to errors in the division between primary and secondary? I would maybe also build up the paragraph differently: Dust and aerosols

can have different effects on the trace gas concentration: On the one hand, it limits the received radiation and hence prevents $NO_2$ destruction. In the same direction (concentration increase) acts the effect of reducing turbulence and hence the diminishing of mixing. However, the aerosols and dust particles act as surface for heterogeneous $NO_2$ destruction processes and this leads to a diminishing of $NO_2$. The received total light intensity anti-correlates with the $NO_2$ concentration and hence.

Re: Thank you for this comment.

Apportioning the measured HCHO using a multiple linear regression model has been widely utilized in previous many studies (Xue et al., 2022; Su et al., 2019; Garcia et al., 2006; Hong et al., 2018; Friedfeld et al., 2002). This is a statistics-based method to separate sources of ambient HCHO. I think your suggestions are quite valuable to us, and worth discussing. However, this is difficult to be answered from statistics perspective. In the future, we plan to discuss this by model simulations.

This is a good suggestion. We have built up the paragraphs about $NO_2$ concentration variation according to your suggestions as follows.

"In contrast to aerosols, we observed large differences in $NO_2$ growth rates (Fig. 7B). At SJZ and DY, $NO_2$ concentrations exhibited a substantially increasing trend. The surface growth rates at the SJZ and DY stations were 6.97 and 17.50 in PRE, and 2.06 and 6.50 in POST, respectively (Fig. 7B (c-1, c-2, d-1, d-2)). In contrast, we observed decreased $NO_2$ concentrations at almost every height at stations NC and XH. The near-surface growth rates at NC and XH were -0.81 and -0.76 in the PRE, and -0.30 and -0.59 in the POST, respectively (Fig. 7B (a-1, a-2, b-1, b-2)). This indicated that dust and aerosols have different effects on the trace gas concentration. On the one hand, they limited the received radiation (Fig. S13), thereby preventing $NO_2$ photolysis, which prolongs its lifetime and favors its accumulation (Chang and Allen, 2006). Moreover, they also had the effect of reducing turbulence and inhibiting dissipation, eventually intensifying surface $NO_2$ accumulation. On the other hand, aerosols and dust particles can act as surfaces for heterogeneous $NO_2$ destruction processes, leading to a decrease in $NO_2$. On the dusty day, large amounts of dust and aerosols provide surface areas for heterogeneous reactions and deposition of different trace gases. Heterogeneous reactions on dust and aerosol surfaces can result in a general decrease in the atmospheric concentrations of trace gases, such as $O_3$, nitrogen oxides, and hydrogen oxides (Kumar et al., 2014; Bauer, 2004; Dentener et al., 1996). Among these reactions, the conversion of $NO_x$ ($NO + NO_2$) to HONO plays an important role in $NO_2$ removal (Stemmler et al., 2006; Ndour et al., 2008; George et al., 2005). Under high AEC and RH conditions, the conversion of $NO_2$ to HONO is further promoted (Xing et al., 2021b). With March 6 and 22 as the comparison benchmarks, we found that the surface growth rates in HONO concentration were 2.70 and 3.52 at the NC station, respectively, which validates our hypothesis (Fig. S14). "

Supplement Sect. 4: This refers to Fig. 10 a which is the histogram? This should probably refer to Figure S13a? Please explain the color coding in the Figure caption of Figure S13a.

Re: Thanks for this correction. We made a mistake about Figure Number here, since we had moved some Figures from manuscript to the Supplementary materials. The color

coding in the Fig. S13a represented the density of spots. The specific meaning is the spots number within area of 1 x-scale unit multiplied by 1 y-scale unit. Here, for example, it stands for the number of spots within 1 $ppbv^2$, since the units of x-scale and y-scale are both ppbv. As the color bar units always changed with the units of x-scale and y-scale, it is unsuitable to add a unit label here.

l.402: Why is this owning to the worsening meteorological conditions?

Re: As described above (line 347-350), on dusty day, elevated dust concentrations weaken turbulence and decrease PBL heights. Unfavorable meteorological conditions not only impede pollutant dissipation and transport, but also favor the accumulation of locally produced pollutants (including direct emissions and secondary production).

To avoid misunderstanding, we have changed the descriptions as follows'

"Owing to amplified pollutant accumulation, increased HCHO concentrations were recorded at both stations SJZ and DY (Fig. 7C)."

Here, we firstly discussed that the meteorological conditions triggered near-surface growth of HCHO, and then elaborated the impact of reduced solar radiation on the HCHO concentration.

l.405: How did you "note" it?

Re: To avoid triggering confusion, we have changed this sentence.

"In addition, we noted the impact of reduced solar radiation on the HCHO concentration." -> "In addition, we believed that reduced solar radiation also influenced the HCHO concentration."

l. 407: Maybe quickly say that you used CO as tracer for primary HCHO and Ox [I am not a chemist, but I have never seen this notation as Ox=O3+NO2, I have only seen NOx.] for photo chemical production (So you use O3?).

Re: Thanks for this suggestion. The $O_x$ ($O_x=O_3+NO_2$) refers to odd-oxygen, while $NO_x$ ($NO_x = NO + NO_2$) are nitrogen oxides. They are different things. The notation as $O_x$ has appeared in many previous studies (Wood et al., 2010; Sun et al., 2021; Su et al., 2019; Hong et al., 2018). The $O_3$ concentrations were collected from CNEMC.

l.424: "in" --> "of"

Re: Thanks for this suggestion. We have corrected it.

l.424f: You mean this is what you are going to show in the following paragraph?

Re: To avoid misunderstanding, we have simplified this paragraph.

l.425: "overpassing" --> "exceeding"

Re: Thanks for this suggestion. We have corrected it.

l.425ff: according to map 1, only parts of NCP. I cannot see the southward moving very well. The east part of YRD is much more covered in the satellite images already at the start? For me it looks more as if the AE moves west, also it looks more "growing".

Re: Let's focus on XH station. At the start, only a few parts of the NCP region were influenced by high-AOD air masses, with XH station out of the affected area. On January 19, 2021, the affected areas got further enlarged so that XH station was completely covered by high-AOD air masses. On January 20, 2021, the polluted areas kept expanding and nearly the whole NCP region had been covered by this high-AOD air masses. When it came to January 21, 2021, it could be found that XH station had been located at the upper boundary of this air masses. On January 22, 2021, XH station

had been completely separated from this air masses. According to the relative position between XH station and high-AOD air masses, we could find this southward moving tendency. However, the satellite results didn't include wind direction information, so we just took them as Supplementary materials. The more convincing results were wind simulations and the southward moving was more obvious in the results of January 22, 2021 in Fig. 8.

You said the AE moved west, and this could also be explained by the wind simulations. As we said, the northward transport (YRD-to-NCP) firstly took shape in upper layers (800-1500 m) on January 19, 2021, while wind moved west in lower layers (0-500 m) (Fig. S18). This would definitively cause some parts of pollutants to move west. However, this was not the main discussion point in Section 3.3. We only focused on the transport between the NCP and YRD here.

l.427: How does this south-to-north transport fit together with the southward mentioned in line 425?

Re:

Satellite observations helped identify the pollutant transport phenomena because transport tends to cause large-scale continuous distribution of pollutants that can be detected by satellite measurements. Actually, Section 3.1, 3.2 and 3.3 were all organized in this logic: satellite observation (pollutant distribution indicated that there might be transport phenomena) -> wind field information or backward trajectories (confirm the occurrence of transport phenomena and reveal transport direction) -> MAX-DOAS observations (analyze the transport process)

In this paragraph, the variation of AOD distribution revealed by the Himawari-8 satellite observations indicated that there might be back-and-forth transport between the NCP and YRD. Then, we used wind simulations to further validate this process. To make it clearer, we have changed this paragraph as follows.

"The Himawari-8 satellite observations revealed a substantial increase in aerosol concentrations within the NCP from January 18 to 20, 2021, with an overall AOD overpassing 0.9 (Fig. S16). Subsequently, high-concentration aerosol air masses assumed a southward movement tendency, gradually leaving the NCP and covering the YRD on January 21–22, 2021. We attributed this phenomenon to the back-and-forth transport of aerosols between these two regions, which we validated using wind-field simulations. The wind field results indicated that the wind blew towards the East China Sea at every altitude on January 18, 2021 (Fig. 8). A south-to-north transport belt firstly formed in the upper layers (500–1500 m) on January 19 and lasted for nearly two days. Around 12:00 on January 21, the wind direction began to change, and the north-to-south transport trend strengthened in the 0–1000 m layer on January 22. The diurnal variation of wind fields in different layers on January 18–22, 2021, were provided in Figs. S17-S21. In terms of overall transport direction, we classified the MAX-DOAS monitoring results into four periods: West-to-East, YRD-to-NCP, Transformation, and NCP-to-YRD, to further explore their vertical characteristics during the transport process (Fig. 9)."

l.426: I do not see this in the data

Re: The high-AOD air masses movement could be seen in Fig. S16, and the wind

simulations could confirm its southward movement. To make this paragraph more understandable, we have simplified this paragraph, as mentioned in Response above.

l.429: I just cannot see this in the plots, I cannot see wind directions at all or wind speeds in this plot. Please work with colors for the speeds and reduce the number of arrows but make them way thicker and longer (i.e. the 8 ms arrow should also grow of course).

Re: Thank you for this comment. We have revised Fig. 8 and Fig. S17-S21. We have enhanced the arrow thickness and changed the arrows' colors according to wind speeds, making orientation and direction more obvious.

[Figure]

l.436: "wind gathered towards"??

Re: We changed this sentence as follows.

"On January 18, 2021, no direct long-range transport occurred between the NCP and YRD (Fig. 8 and Fig. S17)."

l.437: Do you base this comment for the whole region on just the four stations? Why don't you use the network of large number of GB stations? (In fact you, but later. I think I would present it earlier)

Re: Thank you for this question. We found that we had missed some important

information at the beginning of the Section 3.3 (line 416-423), which might be the reason for triggering confusions. Therefore, we have rewritten this paragraph, as follows:

"Back-and-forth transboundary long-range transport between the NCP and YRD is common, especially during winter (Huang et al., 2020; Petaja et al., 2016). During the transport process, the aerosol–PBL interaction can amplify the overall haze pollution and deteriorate the air quality of these two regions (Petaja et al., 2016; Ding et al., 2016; Huang et al., 2014; Huang et al., 2018b). Based on model simulations, Huang et al. (2020) elaborated on this transport process and the haze-amplifying mechanism in three stages. First, air pollutants from the YRD are transported to the upper PBL over the NCP and substantially affect PBL dynamics. Subsequently, under the influence of aerosol-PBL interaction, local pollutant accumulation and secondary production of aerosols are enhanced, causing severe pollution in the NCP. Finally, strong weather patterns (e.g., cold fronts), can dissipate low-PBL pollutants in the NCP and transport them over long distances back to the YRD. Many model simulations have suggested that the mechanism of aerosol–PBL interaction amplifies the overall haze pollution during the transport process (Petaja et al., 2016; Ding et al., 2016; Huang et al., 2014; Huang et al., 2018b). Using MAX-DOAS measurements, we investigated the spatiotemporal variation in aerosols along the transport pathway, and validated the haze-amplifying mechanism of this transboundary transport."

This back-and-forth transboundary transport process between the NCP and YRD and its corresponding haze-amplifying mechanism had been confirmed by many previous studies (Huang et al., 2020). Huang et al. (2020) mainly based their conclusions on the model simulations, and they also used GB stations as validation. However, we could find that some parts of transport process, such as "air pollutants from the YRD are transported to the upper PBL over the NCP", could not be validated by GB stations, since GB stations could only collect surface pollutant concentrations. Therefore, we used vertical distribution of aerosols collected from MAX-DOAS to validate this back-and-forth transport process. This section could be seen as an extended work of previous related studies. If we only use GB stations, it will largely reduce the research values of this section.

l.438: I can clearly see a region exceeding 120? "average" means time average? spatial average? Not clear.

Re: There were a few $PM_{2.5}$ around or over 120 $\mu g/m^3$ in Dongying region, but most of stations had a relatively low $PM_{2.5}$ level, which make their overall average concentrations only around 43.71 $\mu g/m^3$. The "average" meant spatiotemporal average, i.e., the sum of concentrations divided by 994 (the number of stations in this map) and 12 (a total of 12 hours from 8:00 to 19:00).

"These two regions both had acceptable air quality, with the maximum AEC at the four stations being less than 0.88 $km^{-1}$ (Fig. 8) and the overall spatiotemporal average $PM_{2.5}$ concentration approximately 43.71 $\mu g/m^3$ (Fig. S22a)."

Fig. S20: add date headers here.

Re: Thanks for this suggestion. We have added them in Fig. S22.

[Figure]

l.439: How do you define and how do you identify a transport belt?

Re: If there was continuous high-valued pollutant distribution at a certain altitude accompanied with obvious transport direction, we defined this as a transport belt. To make this sentence more understandable, we have changed the "transport belts" into "continuous high-AEC aerosol distribution".

l.440: How do values of 6.1 fit to the color scale (which ends at 2)?

Re: Thank you for this comment. We have revised the Figure and don't set the exact upper limit for the color bar.

l.440: The value of 1.41 quoted seems to be rather the value in the lowest layer and not in the 100--700 m layer?

Re: Yes, but we just pointed to each station's maximum AEC here, not limited to a certain layer. To avoid misunderstanding, we planned to divide this sentence into two ones.

"We noted aerosol transport belts at XH in the 100–1400 m layer, at DY in the 200–1000 m layer, and at NB in the 100–700 m layer, with the peak AECs reaching 6.10, 1.41, and 0.96 km$^{-1}$ at XH, DY, and NB, respectively (Fig. 9)." -> "We noted that a continuous high-AEC distribution occurred at XH in the 100–1400 m layer, at DY in the 200–1000 m layer, and at NB in the 100–700 m layer (Fig. 9). The maximum AECs reached 6.10, 1.41, and 0.96 km$^{-1}$ for XH, DY, and NB, respectively."

l.441: Where do you show this?

Re: To avoid confusion, we have changed the describing order. Firstly, we talked about the YRD-to-NCP transport forming in the 500-1500 m on January 19 (Fig. 8 and Fig. S18). Therefore, there were high-AEC aerosols in upper layers at XH, DY and NB on that day (Fig. 9).

"The wind simulation indicated that south-to-north transport initially took shape in layers of 500–1500 m on January 19 (Fig. 8 and Fig. S18), which we defined as the start of the YRD-to-NCP transport period. During this period, the overall AECs at all stations began to increase in varying degrees. We noted that a continuous high-AEC distribution occurred at XH in the 100–1400 m layer, at DY in the 200–1000 m layer, and at NB in the 100–700 m layer (Fig. 9). The maximum AECs reached 6.10, 1.41, and 0.96 km$^{-1}$ for XH, DY, and NB, respectively."

l.442: I do not see this.

Re: We have reorganized this part of description in the order of wind variation, and added corresponding figure numbers to the end of each key sentence.

"On January 18, 2021, no direct long-range transport occurred between the NCP and YRD (Fig. 8 and Fig. S17). These two regions both had acceptable air quality, with the maximum AEC at the four stations being less than 0.88 km$^{-1}$ (Fig. 8) and the overall spatiotemporal average PM$_{2.5}$ concentration approximately 43.71 μg/m$^3$ (Fig. S22a). The wind simulation indicated that south-to-north transport initially took shape in layers of 500–1500 m on January 19 (Fig. 8 and Fig. S18), which we defined as the start of the YRD-to-NCP transport period. During this period, the overall AECs at all stations began to increase in varying degrees. We noted that a continuous high-AEC distribution occurred at XH in the 100–1400 m layer, at DY in the 200–1000 m layer, and at NB in the 100–700 m layer (Fig. 9). The maximum AECs reached 6.10, 1.41, and 0.96 km$^{-1}$ for XH, DY, and NB, respectively. According to the wind simulation for January 19–21, the YRD-to-NCP transport lasted until 12:00 on January 21 (Fig. 8 and Fig. S18-S20). During this period, large amounts of aerosol from the YRD were transported to the upper layers (500–1500 m) of the NCP. In addition, secondary particle formation intensified because the transport of warm and humid air masses favors aqueous and heterogeneous reactions (Huang et al., 2014). These factors jointly led to a sharp increase in AECs in the high layers at stations XH, DY, and NB (Fig. 9). In contrast, the increase in the near-surface AEC was slower than that in higher layers. On

January 19, for instance, the surface AECs were mostly less than 0.6 km$^{-1}$ from 10:00 to 16:00 in XH, while surface peak AECs in the morning and late afternoon could be explained by the diurnal variation in PBL height (Ding et al., 2008; Ran et al., 2016). At the DY station, the average surface AEC only increased from 0.61 km$^{-1}$ on January 18 to 0.62 km$^{-1}$ on January 19. The reason is that surface transport was driven mainly by the east wind on January 19 (Fig. 8), resulting in PM$_{2.5}$ concentrations at many western CNEMCs exceeding 80 μg/m$^3$ (Fig. S22b). From January 20 to 21, 2021, the surface wind converted into a south wind, but became so weak that near-surface transport contributed little to the NCP (Fig. 8 and Fig. S19-S20). However, we continued to observe a substantial increase in AEC at ground level on January 20–21, 2021 (Fig. 9). At station DY, for example, the average surface AEC increased from 0.61 km$^{-1}$ on January 18 to 1.03 km$^{-1}$ on January 20, which was a 68.9% growth rate. A possible reason for this was the strong dome effect caused by high-layer aerosols. As a result of the aerosol–PBL interaction, PBL height decreases while temperature and humidity increase in the lower PBL, which favors pollution accumulation and secondary aerosol production (Bharali et al., 2019; Huang et al., 2020; Petaja et al., 2016). Generally, this YRD-to-NCP transport intensifies local pollution in the NCP region, causing a substantial increase in aerosol concentrations on January 18–20 (Fig. S16)."

l.446: But the surface AEC seem to increase tremendously?

Re: Our description here was inaccurate, and we have changed it.

"This also explains why the increase in near-surface AEC was less than that in the higher layers." -> "In contrast, the increase in the near-surface AEC was slower than that in higher layers."

l.446 vs. line 451: This seems to be a contradiction?

Re: We have rewritten these two sentences to avoid misunderstanding.

"This also explains why the increase in near-surface AEC was less than that in the higher layers." -> "In contrast, the increase in the near-surface AEC was slower than that in higher layers."

"Despite the minimal contribution from surface transport, we observed a substantial increase in AEC at the ground level on January 20–21, 2021 (Fig. 9)." -> "However, we continued to observe a substantial increase in AEC at ground level on January 20–21, 2021 (Fig. 9)."

In this way, I believe there will be no contradiction. The slow increase in near-surface AEC was validated by MAX-DOAS results on January 19. A gradual and obvious increase could be found in MAX-DOAS observations on January 20-21.

l. 449: XH decreased after the 19th? (c.f. FIg. 9)

Re: Compared with the results on the 19$^{th}$, the high-altitude AEC decreased but was still higher than the AEC levels at corresponding heights on the 18$^{th}$, which suggested the high-altitude transport phenomena still existed after the 19$^{th}$. This didn't affect our final conclusion.

l.461: 4.42 in Fig. 9 with color scale ending at 2

l.466: The station in both regions should be HNU. But HNU drops earlier than DY, not later?

Re: As stated in the manuscript, the AECs in the 0–1 km layer at HNU assumed a continuously increasing tendency during the whole period, without dropping. We attributed this to a longer dome effect determined by their overlapping zone location.

Fig.9: As mentioned above, maybe color scale maximum of 2 is not a good choice if you have values of > 4 and > 6. If you fear losing detail at lower aerosol depths, maybe consider a logarithmic scale?

Responses to l.440, 461, Fig.9:

We have tried a logarithmic scale as follows, but the result doesn't seem very good. It cannot display the remarkable variation in aerosol distribution during the transport process.

[Figure]

After consideration, we still decide to adopt the normal scale, but we don't set the exact upper limit for the color bar. In this way, we think this figure can display the values over 2 km$^{-1}$.

[Figure]

Sect. 4: I do not see much of the conclusions really supported by the data. And exception are lines 493 --lines 502 which I do see supported. The paragraph following that, (lines 503 -- 510) I cannot say whether I agree or not, because I cannot see anything in the provided wind field plots. I could give you the benefit of doubt here, but I would prefer to see this more clearly in the plots. I cannot agree with your summary (line 511 to 515) because I don't think that the data was clearly supporting your statements.

Re: Thank you for this comment. According to your suggestions, we have found some key problems to cause misunderstanding and confuse readers.

For Section 3.1, we find there are many speculative arguments making Section 3.1 confusing and less convincing. We have separated the discussions around Fig.3 and Fig. 4 into two parts, reducing the frequency of jumping descriptions between two Figures to some extent. To make Section 3.1 more understandable and emphasize the main points, we have deleted many speculative conclusions (e.g., secondary aerosol generation, aerosol transport among stations). The main point of Section 3.1 is to discuss different varying characteristics of MTL for aerosol, $NO_2$, and HCHO. We have changed the structure of Section 3.1 and added some demonstrations to make the descriptions about the MTL more prominent in this section.

For Section 3.2, we feel that we paid much attention to the differences between the vertical profile shapes among different stations before. For example, we thought the vertical profile shapes could be classified into multiple-peak and Gaussian shapes on clean days. However, as you said, some little differences, like the difference between Gaussian-shaped vertical profile with high surface concentration and two-peak shape, could be caused by other factors (e.g., an imperfect aerosol retrieval) instead of pollutant vertical distribution itself. Instead, we find a better method, which can summarize the varying features of vertical profiles on the dusty day better—the movement of high-altitude peaks. The previous study indicated that AEC in layers of 0-500 m contributed most to trace gas profile variation, and trace gas concentration at 1.5-3.5 km responded most sharply to perturbations in AEC below 500 m (Friedrich et al., 2019). The trace gas profile below 1 km shows low sensitivity to AEC variation. Moreover, the increase at AEC tends to intrigue a decrease at trace gas concentration in 0-500 m layers. However, we find that the explosive growth at low-altitude trace gas concentration always accompany a sharp increase at surface AEC on the dusty day. Therefore, we think the peaks of trace gas below 0.5 km are mainly attributed to concentration increase itself instead of imperfect aerosol retrieval.

For Section 3.3, we have modified the wind field Figures to make them clearer, including enhancing the arrow thickness and changing the arrows' colors according to wind speeds.

l.481-482: "attributed" is repeated
l.486: remove "the"
Re: Thank you for these suggestions. We have corrected these sentences.

l.490ff: Is it maybe that clean and dust days are mixed here? You showed this for the dusty days, not for the clean days.

l.491: when is it maintaining a Gaussian shape?

Responses to 490-491:

Thank you for your comment. We mixed the clean and dusty days here. We have changed our description as follows.

"By comparing the results of the dusty day (i.e., March 15, 2021) and clean days (i.e., March 6 and 22, 2021), we found that high-altitude concentration peaks dropped to a lower layer and even disappeared on the dusty day. We attributed this result to dust being able to suppress dissipation, weaken pollutant transport, and intensify local pollution accumulation."

Referee#3

*Thank you for your careful review and constructive suggestions. These suggestions are quite valuable to us, and help improve our manuscript a lot.*

Point-to-point responses

*We appreciate the reviewers for their valuable and constructive comments, which are very helpful for the improvement of the manuscript. We have revised the manuscript carefully according to the reviewers' comments. We have addressed the reviewers' comments on a point-to-point basis as below for consideration, where the reviewers' comments are cited in **black**, and the responses are in **blue**.*

**General Comments:**

The manuscript entitled "Evaluation of Transport Processes over North China Plain and Yangtze River Delta using MAX-DOAS Observations" by Song et al investigates the transport patterns and vertical distributions of NO2, HCHO and aerosols using a number of instruments from a MAX-DOAS network in China. The temporal variation of the air mass composition has been investigated using modelled wind fields, which allow to identify air masses moving from the region of one instrument to the other. This is a very useful approach that allows to investigate the dynamical and chemical processing of individual air masses.

A main problem of the manuscript is that there are many occasions where no clear distinction has been made between (1) conclusions evidently inferred from the measurements, (2) findings from other studies that support the measurements and (3) hypotheses based on the observations. Many conclusions drawn from measurements are highly speculative, for example that the presence of NO2 and HCHO enhance the AEC, and that secondary aerosols are present (Section 3.1). Another example is the statement "We discovered that secondary aerosol generation always accompanied the regional transport process" (L302), for which there is neither direct evidence from the measurements nor any other study mentioned that would support this finding.

Re: Thank you for this comment. There are many speculative arguments making Section 3.1 confusing and less convincing. In addition, our descriptions jump between Fig. 3 and Fig. 4, which easily cause misunderstanding. To make Section 3.1 more understandable and emphasize the main points, we have deleted many speculative conclusions (e.g., secondary aerosol generation, aerosol transport among stations) and changed the structure of Section 3.1 as follows.

[revised manuscript text omitted]

I suggest to remove the discussion on dust properties inferred from the measured intensity. MAX-DOAS instruments are usually not radiometrically calibrated. Even if the spectrometers are of the same type, the signals from different instruments cannot be directly compared to each other since they depend on many parameters, such as the gain of the amplifier, as well as on the adjustment of the telescope optics, the length of the

fibre bundle, etc. I therefore suggest to remove the corresponding paragraphs (Fig. 6 and L368-374), which anyway do not provide much extra information compared to the retrieved extinction profiles.

Re: Thank you for this comment. We have given up classifying the four stations and basing the following discussions on the classification here. We have changed the descriptions of L368-374 as follows, and removed Fig. 6 and Fig. S10.

"In addition to aggravating pollutant accumulation, transported dust can affect the environment and pollutant concentrations in other ways. To quantitatively demonstrate the impacts of dust on various pollutants, we introduced growth rate in the comparative analysis (Supplementary Sect. S9). For convenience, we defined the comparison of the results of March 6 and 15, 2021, as precomparison (PRE), and we defined the comparison between March 15 and 22, 2021, as postcomparison (POST)."

Instead, we depict a relative difference in optical intensity between dusty day and clean days as Fig. 10. This Figure is used to describe the impacts of dust storm on light intensity, due to the light attenuation of dust and enhanced aerosols.

[Figure]

**Fig. S10.** The difference of optical signal intensities received by MAX-DOAS between dusty day and clean days. (a) PRE: Intensity (March, 15) – Intensity (March, 6); (b) POST: Intensity (March, 15) – Intensity (March, 22).

Finally, the manuscript appears to require substantial revision regarding of the usage of the English language. I have mentioned only a few in the technical corrections below.

Re: We let a language revision institution help us polish our language and correct mistakes.

[Figure]

**Certificate of Elsevier Language Editing Services**

The following article was edited by Elsevier Language Editing Services:
**"Evaluation of Transport Processes over North China Plain and Yangtze River Delta using MAX-DOAS Observations"**

Authored by:
**Yuhang Song**

**Specific Comments:**

L46: How does the transport of pollutants lead to the production and emission of pollutants? Please explain.

Re: Thank you for this comment. We have changed this sentence to avoid misunderstanding.

"transportation directly deteriorates the environment through the production and emission of a large number of pollutants" -> "transport carries large amounts of pollutants, directly deteriorating air quality."

L73: This sentence is not only too general, but also incorrect. Satellite remote sensing data is certainly extremely useful to monitor variations in the atmospheric composition (although with no or only limited vertical resolution in the troposphere).

Re: Thank you for this comment. We have changed our descriptions.

"Furthermore, satellite remote sensing data cannot be used to monitor the vertical features of and variations in the atmospheric composition." -> "However, because of their limited temporal and spatial resolutions, satellite data cannot be used for the continuous monitoring of a specific area (Bessho et al., 2016; Veefkind et al., 2012). It is difficult to characterize the vertical distribution of atmospheric composition using only satellite remote sensing or CNEMC data."

L76: I guess the statement "Large uncertainties remain in pollutant distribution estimation" only refers to model simulations. Please clarify.

Re: Yes, it only refers to model simulations. To avoid misunderstanding, we have changed this sentence.

"Large uncertainties remain in pollutant distribution estimation, primarily owing to the effects of emission inventories, meteorological fields, and hypothetical conditions (Huang et al., 2016; Xu et al., 2016; Zhang et al., 2017)." -> "However, considerable uncertainties remain in estimating pollutant distribution using model simulations, primarily owing to the effects of emission inventories, meteorological fields, and of assumptions made (Grell et al., 2005; Huang et al., 2016; Xu et al., 2016; Zhang et al.,

2017)."

L77: What do you mean with "hypothetical conditions"?

Re: Maybe using "assumptions" is better. As we all know, there are some basic model assumptions in simulation models (e.g., WRF/Chem, MM5/Chem), and the uncertainties caused by these assumptions cannot be ignored.

For example, Grell et al. (2005) indicated that the leaf temperature assignment assumption influenced $O_3$ statistics. In addition, other model components, such as surface layer parameterizations and boundary condition assumptions, contributed to model uncertainty.

L86: Describing DOAS as a "a cutting-edge and promising method" seems inappropriate. DOAS is a well established and well validated technique that has been applied for the measurement of atmospheric trace gases since decades.

Re: Thank you for this comment.

"The differential optical absorption spectroscopy (DOAS) technique (Platt and Stutz, 2008) is a cutting-edge and promising method for the quantitative analysis of many crucial atmospheric gases." -> "The differential optical absorption spectroscopy (DOAS) technique (Platt and Stutz, 2008) is a well-established and reliable method for the quantitative analysis of many crucial atmospheric gases."

L93: I think this statement is not correct, since LIDAR has a much better vertical resolution than MAX-DOAS (at least for aerosols).

Re: Thank you for this comment. We have amended this sentence.

"Compared with the above techniques, MAX-DOAS is high resolution and low cost, and its operation is automatic." -> "Compared with the above techniques, MAX-DOAS does not require radiometric calibration and has many other advantages such as simple design, low power demand, possible automation, low cost, and minimal maintenance."

L96: MAX-DOAS is a not a hyperspectral method since spectral information is only obtained from a single viewing direction at one time. It is also not clear why MAX-DOAS should be a stereoscopic technique.

Re: We have deleted this description to make the sentence more reasonable.

"a mature ground-based hyperspectral stereoscopic remote sensing network" -> "a mature ground-based remote sensing network"

Section 2.3: Please discuss the fit errors and ==detection limits== for the retrieved species. The optical density of HCHO and HONO shown in figure S2 are very weak. Can these trace gases be detected reliably, and is the signal-to-noise ratio sufficient for a useful retrieval of the vertical distribution of HONO and HCHO?

Re: We have supplemented the discussion about the fit errors and detection limits in Section. 2.3 as follows.

"We assumed two times the fitting error RMS as the DSCD detection limits (Wang et al., 2017; Lampel et al., 2015), which were $7 \times 10^{41}$ (molec$^2 \cdot$cm$^{-5}$), $1.6 \times 10^{15}$, $3.6 \times 10^{15}$, and $5.8 \times 10^{14}$ molec·cm$^{-2}$ for $O_4$, $NO_2$, HCHO, and HONO, respectively."

The results below can meet the RMS ($< 1 \times 10^{-3}$) and detection limit filtering criteria.

[Figure]

But we plan to use another HCHO and HONO result, and change the colors usage in this Figure as follows.

[Figure]

L140: Here it is not clear what you mean with "we calculated the ring spectrum as the measured spectrum, considering the contribution of the stratosphere to the DSCDs".

Re: "Furthermore, we calculated the ring spectrum as the measured spectrum, considering the contribution of the stratosphere to the DSCDs" -> "The Ring spectrum was added to the fitting settings to remove the influence of the stratosphere on the DSCDs."

Section 2.4: Given that the signal-to-noise ratio apparent in Figure S2 seems to be very low, I am surprised that the smoothing and noise error components of the HCHO profiles are similar to those of the NO2 profile. It is not clear what you mean with "algorighmic error". Is this error due to inaccurate model parameters b or due to general incapabilities of the forward model to realistically represent the underlying physics?

Re: Actually, the HCHO smoothing and noise error is larger than the NO$_2$ result, especially for near-surface value (HCHO:42%, NO$_2$:11%), which agrees with the SCD retrieved results.

The descriptions about "algorithmic error" might be a little confusing. We have changed the demonstrations in the main text as follows and further elaborate it in Supplementary Sect. S3.

Main text: "Algorithmic error (i.e., the difference between the measured and modeled DSCDs) arises from an imperfect minimum of the cost function. This error is a function of the viewing angle. However, it is difficult to assign discrepancies between the measured and modeled DSCDs at each profile altitude. Therefore, the algorithm error on the near-surface values and column densities cannot be realistically estimated. Given that measurements at 1° and 30° elevation angles are sensitive to the lower and upper air layers, respectively, the average relative differences between the measured and modeled DSCDs for a 1 and 30° elevation angles can be used to estimate the algorithm errors on the near-surface values and column densities, respectively (Wagner et al., 2004). Considering its trivial role in the total error budget, we estimated these errors on the near-surface values and the column densities at 4 and 8 % for aerosols, 3 and 11 % for NO$_2$, and 4 and 11 % for HCHO, respectively, according to Wang et al. (2017)."

Supplementary Sect. S3: "The algorithm error is the discrepancy between the measured ($\mathbf{y}$) and modelled DSCDs ($F(\mathbf{x}, \mathbf{b})$), which is mainly caused by an imperfect minimum of the cost function ($\chi^2$) in Eq. s3. This error is a function of the viewing angle. Due to the difficulty of assigning this error to each altitude of profile, the algorithm errors on the near-surface values and column densities are usually estimated by calculating the average relative differences between the measured and modeled DSCDs at the minimum and maximum elevation angle (except 90°), respectively (Wagner et al., 2004).

$$\chi^2 = (\mathbf{y} - F(\mathbf{x},\mathbf{b}))^T \mathbf{S}_\varepsilon^{-1}(\mathbf{y} - F(\mathbf{x},\mathbf{b})) + (\mathbf{x} - \mathbf{x}_a)^T \mathbf{S}_a^{-1}(\mathbf{x} - \mathbf{x}_a) \tag{s3}$$

where $F(\mathbf{x},\mathbf{b})$ is the forward model; $\mathbf{b}$ represents the meteorological parameters; $\mathbf{y}$ is the measured DSCDs; $\mathbf{x}_a$ is the a priori vector that serves as an additional constraint; $\mathbf{x}$ is the state vector."

L208: Explain why the wind-speed in north-easterly direction (and not in any other direction) is of relevance here.

Re: Thank you for this comment. We have supplemented explanations at the start of Section 2.5.

"Given the major role of pollutant transports in the JJJ region, hourly transport fluxes of each layer ($Flux_i$) and column transport fluxes ($Flux_c$) at each station were

calculated to illustrate the dynamic transport process of pollutants along southwest-northeast pathway." -> "Owing to the semi-basin topography, southwesterly or southerly winds play a dominant role in pollutant transport in the JJJ region. In this study, we mainly focused on pollutant transport in the southwest-northeast direction, and thus selected four different stations along this pathway, namely, Shijiazhuang (SJZ), Wangdu (WD), Nancheng (NC), and Chinese Academy of Meteorological Sciences (CAMS) (Fig. 1). We calculated the hourly transport fluxes of each layer ($F_i$) and column transport fluxes ($F_c$) at each station to illustrate the dynamic transport process of pollutants along the southwest-northeast pathway."

[Figure]

Figure 2: Converting the in situ NO2 from μ g/m3 to ppb would allow for a much better quantitative assessment of the agreement between both datasets.

Re: We have converted unit of NO2 from μ g/m3 to ppb using the following formula.

$$C(ppb) = \frac{X(\mu g/m^3) \cdot V_m(L/mol)}{M_g(g/mol)}$$

$$V_m(L/mol) = \frac{R(J/(mol \cdot K)) \cdot T(K)}{P(kpa)}$$

In addition, to lessen the effects of occasional extreme conditions, we filtered the "abnormal values" of MAX-DOAS and in situ measurements before comparison. This filtering helped improve the correlations from 0.615 to 0.752, and 0.671 to 0.74, for

aerosol and NO₂, respectively.

The filtering procedure is elaborated in Supplementary Sect. S8 as follows.

**"Section S8. The abnormal values definition and filtering**

For lessening the impacts of "abnormal value" caused by occasional extreme conditions, we needed to adopt a method to seek out the abnormal values and filter them out. In a series of data, we firstly found the first quartile (Q1), median (Q2), and the third quartile (Q3), which are the 25th, 50th and 75th percentile of all values from small to large, respectively. The difference between Q1 and Q3 is called interquartile range (IQR) (i.e., IQR = Q3-Q1). The upper limit ($L_{upper}$) and lower limit $L_{lower}$ were defined as Q3 plus IQR, and Q1 minus IQR, respectively (i.e., $L_{upper}$=Q3+IQR, $L_{lower}$=Q1-IQR). The values larger than $L_{upper}$ or lower than $L_{lower}$ were defined as abnormal values, and discarded. After filtering the data, the correlation had increased from 0.615 to 0.752, and 0.671 to 0.74, for aerosol and NO₂, respectively.
"

[Figure]

Figure 3: Why are there so many missing profiles? Is this due to outages of the instrument or has the profile retrieval failed in these cases?

Re: On one hand, outages of the instrument result in some periods of failure, and make some stations lack SCDs during the certain period. For example, the vertical profiles are lacked after 16:00 at the WD station, which is mainly attributed to measurement failure. On the other hand, some results cannot meet the selection criteria (DOF > 1.0, relative error < 50%), and are filtered out. If we don't filter the data, there will be more profiles, but many results are obviously unreasonable (as follows). For instance, there are >0.5 km⁻¹ AECs at each altitude during 11:00-12:00 at the SJZ station.

[Figure]

L270ff: I feel that the description of the temporal and vertical distribution of aerosols at the different stations is not representing the overall picture appropriately. For example, it is stated that there is a "subtle increase" in aerosols above NC around 12:00, but it is not mentioned that this is just the onset of the presence of a strong aerosol layer throughout the afternoon. The finding that a persistent and elevated aerosol layer is first present at SJZ, and later at NC and CAMS, is not explicitly discussed. Are the times at which the aerosol layer reaches the different locations in agreement with the transport times from station to station as estimated from the wind speed? This would give further evidence that long-range transport has indeed occurred. What could be the reason for the much lower AECs at WD than at the other stations?

Re: Thank you for this comment. As you said, we didn't make clear distinctions between (1) conclusions evidently inferred from the measurements, (2) findings from other studies that support the measurements and (3) hypotheses based on the observations. Only with existing data, it is not convincing to discuss whether the distance fits together with the wind speed and the time distance. This is because the distance among SJZ, WD, and NC is too long (> 40 km), and wind speeds would experience many uncertain changes during this process. Maybe these arguments should be considered as hypotheses based on the observations instead of conclusions evidently inferred from the measurements. Given that the main points of Section 3.1 are to discuss the MTL of various pollutants, we decide to delete such descriptions.

To emphasize the main points, we have changed structure of Section 3.1. For Fig.3, we mainly discuss spatiotemporal distributions of aerosol, $NO_2$, and HCHO at different stations during this regional transport as follows. And the reason for the much lower AECs at WD than at the other stations is also contained in the following descriptions.

"Figure 3 presents the temporal variations in the vertical distributions of aerosols, $NO_2$, and HCHO during this regional transport. At the CAMS and NC stations, the aerosol, $NO_2$, and HCHO concentrations were consistently high near the surface, primarily because of the heavy traffic flow and dense factory emissions in Beijing (Zhang et al.,

2016; Li et al., 2017). Previous studies have suggested that urban air pollution in Beijing is dominated by a combination of coal burning and vehicle emissions, which results in severe particulate pollution (Wang and Hao, 2012; Wu et al., 2011). At SJZ, the $NO_2$ concentration was high (~ 12 ppb) in the morning and late afternoon, whereas the concentration was lowest (~ 6 ppb) near noon, which is explained by the morning and evening rush hour. Comparatively, the overall AEC and $NO_2$ levels were relatively low at the WD station, whereas a continuous high-value HCHO distribution (> 2 ppb) occurred at 0-1500 m between 11:00 and 16:00. This occurred because the WD station is located in a farm field with less traffic flow and high vegetation coverage; therefore, large amounts of HCHO are directly emitted by biogenic sources and secondarily produced by natural and anthropogenic volatile organic compound (VOC) photolysis (Wang et al., 2016; Wu et al., 2017a)."

L297: How exactly do NO2 and HCHO enhance the AEC?

Re: Thank you for this comment. We speculated that $NO_2$ and HCHO enhance the AEC through secondary aerosol generations. As you mentioned in General Comments, we lacked direct evidence from the measurements that support this finding. Considering that this is not a conclusion evidently inferred from the measurements, we have deleted these descriptions.

L300ff: How do you know that secondary aerosols were generated? To my knowledge, this cannot be inferred from MAX-DOAS measurements. I cannot find any evidence for your statement that secondary aerosol generation is always accompanied the regional transport process. Does that come from model calculations or other measurements? If so, please explain in detail. Is NO2 really the main precursor for organic aerosols? Please cite relevant publications that support this statement.

Re: Thank you for this comment. Secondary aerosols generation is just one of our speculations. We didn't conduct any model calculations or other measurements here. Considering that it is less associated with the main points of Section 3.1, we have deleted these demonstrations.

L277: It appears from Fig. 3 that the decrease in MTL of aerosols at NC already occured at 14:00, not 16:00. I do not think that the decrease in the aerosol layer height is related to the formation of a nocturnal boundary layer, which is formed much later right before sunset, and is initially very shallow. Aerosols present aloft would reside in the residual layer above the nocturnal surface layer. It seems much more likely that the increase in aerosol layer width is instead caused by increased vertical mixing due to a heat up of the surface in the course of the day.

Re: Thank you for this suggestion. We have changed our demonstrations here.

"We observed a similar phenomenon at NC around 16:00 and at CAMS after 17:00. Surface aerosol accumulation is closely linked with the collapse of the mixing layer and the formation of a stable nocturnal boundary layer (Ding et al., 2008; Ran et al., 2016), triggering a descending tendency in the MTL." -> "In the late afternoon, aerosols gradually accumulated towards the surface, and triggered a variation in the distribution of $F_i$. After 16:00, the shift in the high-AEC air mass caused the transport fluxes in the lower layers (100–200 m) to increase to > 1.1 and ~ 2 $km^{-1} \cdot m \cdot s^{-1}$ for the CAMS and SJZ stations, respectively. Surface aerosol accumulation is closely linked to the collapse

of the mixing layer and formation of a stable nocturnal boundary layer (Ding et al., 2008; Ran et al., 2016). Remarkably, high-altitude aerosol air masses began to mix with near-surface aerosols after 14:00 at the NC station (Fig. 3), triggering a variation in the MTL (Fig. 4). This might be explained by enhanced vertical mixing due to the heating of the surface during the course of the day (Castellanos et al., 2011; Wang et al., 2019). Generally, the MTL of aerosols was situated at 400–800 m during the daytime, where variations in the boundary layer and increased vertical mixing can influence the MTL."
L285ff: According to Eq. 3, shouldn't the unit for trace gas flux be ppb·m·s$^{-1}$, and for aerosol extinction km$^{-1}$·m·s$^{-1}$?
Re: Yes. It's our fault to make descriptions jump all the time between Fig. 3 and Fig. 4, without inserting a Figure number here. L285ff is the description about Fig. 3. To avoid misunderstanding and make the paragraphs more understandable, we have changed the structure of Section 3.1. Now, we describe Fig. 3 firstly, and then mainly demonstrate Fig. 4 as follows.
"Figure 3 presents the temporal variations in the vertical distributions of aerosols, NO$_2$, and HCHO during this regional transport. At the CAMS and NC stations, the aerosol, NO$_2$, and HCHO concentrations were consistently high near the surface, primarily because of the heavy traffic flow and dense factory emissions in Beijing (Zhang et al., 2016; Li et al., 2017). Previous studies have suggested that urban air pollution in Beijing is dominated by a combination of coal burning and vehicle emissions, which results in severe particulate pollution (Wang and Hao, 2012; Wu et al., 2011). At SJZ, the NO$_2$ concentration was high (~ 12 ppb) in the morning and late afternoon, whereas the concentration was lowest (~ 6 ppb) near noon, which is explained by the morning and evening rush hour. Comparatively, the overall AEC and NO$_2$ levels were relatively low at the WD station, whereas a continuous high-value HCHO distribution (> 2 ppb) occurred at 0-1500 m between 11:00 and 16:00. This occurred because the WD station is located in a farm field with less traffic flow and high vegetation coverage; therefore, large amounts of HCHO are directly emitted by biogenic sources and secondarily produced by natural and anthropogenic volatile organic compound (VOC) photolysis (Wang et al., 2016; Wu et al., 2017a).
Nevertheless, Fig. 3 cannot reveal the exact layers in which the main transport phenomena occur. For instance, at the CAMS station, the AEC at the surface and upper layers both reached ~ 0.5 km$^{-1}$ around noon, making it difficult to determine the layer in which transport was more obvious. To further demonstrate the dynamic transport process of different pollutants, we calculated the hourly $F_i$ and $F_c$, and defined the MTL. As shown in Fig. 4, a positive $F_i$ indicates that all transport flux projections in the southwest-northeast pathway are from southwest to northeast at the four stations. The MTLs of aerosols, HCHO, and NO$_2$ exhibited different spatiotemporal characteristics. Although surface and high-altitude (400–800 m) AECs both remained at a relatively high level (> 0.3 km$^{-1}$) at CAMS during 12:00-17:00 (Fig. 3), there was a large discrepancy between their corresponding $F_i$ values (Fig. 4). The aerosol near-surface $F_i$ was ~ 1 km$^{-1}$·m·s$^{-1}$ after 12:00, while $F_i$ in layers of 400–800 m all exceeded 1.2 km$^{-1}$·m·s$^{-1}$, and even reached ~ 2 km$^{-1}$·m·s$^{-1}$ around 12:00. At the SJZ station, the AECs at surface and 300–1000 m layer mostly ranged from 0.3 to 0.4 km$^{-1}$,

especially after 10:00 (Fig. 3). However, the MTLs of aerosols were mostly at 400–800 m throughout the day, with many transport fluxes in those layers even reaching ~ 2 km$^{-1}$·m·s$^{-1}$ (Fig. 4). At the WD station, the highest $F_i$ also tended to occur at high layers (400–800 m), with maximum $F_i$ exceeding 1.7 km$^{-1}$·m·s$^{-1}$ at 400–500 m at 15:00. This suggested that aerosol transport occurred mainly in the upper layers. In the late afternoon, aerosols gradually accumulated towards the surface, and triggered a variation in the distribution of $F_i$. After 16:00, the shift in the high-AEC air mass caused the transport fluxes in the lower layers (100–200 m) to increase to > 1.1 and ~ 2 km$^{-1}$·m·s$^{-1}$ for the CAMS and SJZ stations, respectively. Surface aerosol accumulation is closely linked to the collapse of the mixing layer and formation of a stable nocturnal boundary layer (Ding et al., 2008; Ran et al., 2016). Remarkably, high-altitude aerosol air masses began to mix with near-surface aerosols after 14:00 at the NC station (Fig. 3), triggering a variation in the MTL (Fig. 4). This might be explained by enhanced vertical mixing due to the heating of the surface during the course of the day (Castellanos et al., 2011; Wang et al., 2019). Generally, the MTL of aerosols was situated at 400–800 m during the daytime, where variations in the boundary layer and increased vertical mixing can influence the MTL. In contrast to aerosols, we found that a high-value NO$_2$ $F_i$ frequently occurred in the 0–400 m layer. Except that $F_i$ reached the highest level of ~ 50 ppb·m·s$^{-1}$ in the 400–600 m layer at 16:00 at the SJZ station, the other highest $F_i$ all occurred below 400 m at any station and at any time. This indicated that the MTL of NO$_2$ was 0–400 m. Near-surface NO$_2$ emission sources (e.g., vehicle and factory emissions) might be the main reason for this phenomenon. Compared with aerosols and NO$_2$, we found that high-value HCHO $F_i$ extended to higher altitudes. Taking CAMS as an example, we found the strongest HCHO $F_i$ constantly emerging at 1000–1200 m from 8:00 to 13:00, and averaging 9.18 ppb·m·s$^{-1}$. During the same period, surface HCHO $F_i$ only averaged 6.44 ppb·m·s$^{-1}$. However, at the CAMS station, the surface HCHO concentration was much higher than that of the 1000–1200 m layer between 8:00 and 13:00 (Fig. 3), proving that high-altitude transport contributed more to overall HCHO transport. After 10:00, we found that the highest HCHO $F_i$ gradually increased from ~ 8 to ~ 20 ppb·m·s$^{-1}$ at WD, with the MTL of HCHO ranging from 400 to 1000 m. At station SJZ, the strongest HCHO $F_i$ increased from ~ 10 to ~ 16 ppb·m·s$^{-1}$ during 11:00–17:00, with the highest transport fluxes occurring mostly at 400–800 m. These findings indicated that the MTL of HCHO was mainly 400–1200 m. The sharp variation in the MTL at the NC station might be caused by atmospheric vertical mixing (Castellanos et al., 2011; Wang et al., 2019). As shown in Fig. 3, high HCHO concentrations tend to appear at higher altitudes than those of aerosols and NO$_2$. A possible explanation is that the precursor compounds of HCHO are transported to higher layers and converted into HCHO through photochemical reactions, resulting in elevated HCHO concentrations at higher altitudes (Kumar et al., 2020). Furthermore, strong high-altitude winds were more conducive to HCHO transport (Fig. S5), which further increased the corresponding transport flux. Notably, HCHO $F_i$ was enhanced around noon because the increased solar radiation promotes the secondary generation of HCHO. Long-term observations have revealed that secondary HCHO formation through VOCs photolysis plays a significant role in

Beijing (Liu et al., 2020; Zhu et al., 2018).”

L323: What kind of satellite results are your referring to? This should be explained in the main text, but is not even clear from the caption of Fig. S5.

Re: “The satellite” referred to the Himawari-8, which can monitor the spatial distribution of aerosols and dust.

To make it clearer, we have changed some words.

Main text: “The satellite results…” -> “The Himawari-8 observations…”

Supplementary materials: “**Fig. S5.** A severe dust storm invaded northern China at (a) 8:00 and (b) 14:00 on March 15, 2021.” -> “**Fig. S8.** The Himawari-8 observations: a severe dust storm invaded northern China at (a) 8:00 and (b) 14:00 on March 15, 2021. The dashed black contour line indicates the NCP region.”

Fig. S6: It is not clear what is shown here. Are these trajectories at different times or at different heights? The trajectories should be colour-coded for different heights/times.

Re: These trajectories are at different times. We have color-coded the trajectories for different times and added description information as follows.

[Figure]

**“Fig. S6.** The 24-h backward trajectory results of (a) SJZ, (b) DY, (c) NC, and (d) XH on March 15, 2021 by means of the HYSPLIT model.” -> “**Fig. S9.** The 24-h backward

trajectory results of (a) SJZ, (b) DY, (c) NC, and (d) XH from 00:00 to 23:00 on March 15, 2021 by means of the HYSPLIT model. The altitude of the receptor site was set to the 100 m above ground level.
”

**Technical Corrections**

L50: "driven by the southwest wind" -> "driven by south-westerly winds"

L66: "pollutant concentrations monitoring" -> "pollutant concentrations monitored"

L67: "Characterize" -> "Characterizing"

L74: "The chemical transport model" -> "Chemical transport models"

L75: "pollutant distribution" -> "pollutant distributions"; "is" -> "are"

Section 2: The title "Method and methodology" is a tautology. Use either "Methods" or "Methodology"

Re: Thank you for these corrections. We have followed the suggestions above and corrected all of them.

L124: "We operated a commercial MAX-DOAS instrument" -> "We operated seven (?) commercial MAX-DOAS instruments"

Re: "We operated a commercial MAX-DOAS instrument (Airyx, Heidelberg, Germany)" -> "We operated eight commercial MAX-DOAS instruments (Airyx SkySpec-1D, Heidelberg, Germany)"

L140: Please explain what SCDs are (integrated concentrations along the light path).

Re: We have added a sentence before to explain SCDs.

"Spectral analysis derives the slant column densities (SCDs), i.e., the integrated concentration along the light path."

Section 2.6: According to the ACP guidelines, all variables should be named according to the IUPAC conventions, with all variables being named using only a single lower-case letter. For example, in Equation (2) the expression va would be by convention interpreted as v·a, which is not what you mean here. I would suggest to use u and v  for the meridional and zonal wind, respectively, and to replace Fluxc with Fc, and WSi with wi.

Re: Thank you for this suggestion. We have corrected these problems.

L221: Do you mean "layer with highest transport"?

Re: "the layer with high transport flux" -> "the layer with the highest transport flux"

L223: "discrepancy" -> "differences"

L257: Two times "that".

Re: Thank you for these corrections. We have followed the suggestions above and corrected all of them.

L258: "continuously" -> "homogeneously"

Re: "According to the TROPOMI results, we found that that $NO_2$ was continuously distributed between SJZ and WD, whereas a HCHO distribution belt connected NC with CAMS on February 5, 2021 (Fig. S2)." -> "The TROPOMI results indicated that $NO_2$ was homogeneously distributed between SJZ and WD and that, on February 5, 2021, the HCHO distribution belt connected NC with CAMS (Fig. S4)."

**References**

Grell, G. A., Peckham, S. E., Schmitz, R., McKeen, S. A., Frost, G., Skamarock, W. C., and Eder, B.: Fully coupled "online" chemistry within the WRF model, Atmos Environ, 39, 6957-6975, 10.1016/j.atmosenv.2005.04.027, 2005.

---

## Author Response (AR2)

*Thank you for your careful review and constructive suggestions. These suggestions are quite valuable to us, and help improve our manuscript a lot.*

**Point-to-point responses**

*We appreciate the reviewers for their valuable and constructive comments, which are very helpful for the improvement of the manuscript. We have revised the manuscript carefully according to the reviewers' comments. We have addressed the reviewers' comments on a point-to-point basis as below for consideration, where the reviewers' comments are cited in **black**, and the responses are in **blue**.*

I'm pleased to accept your revised manuscript "Evaluation of Transport Processes over North China Plain and Yangtze River Delta using MAX-DOAS Observations" for publication in ACP subject to minor revisions as listed below.

As the reviewer pointed out, your explanation of the Ring effect is not correct.
Re: Thank you for this comment.
"The Ring spectrum was added to the fitting settings to remove the influence of the stratosphere on the DSCDs." -> "The Ring spectrum was added to the fitting settings to remove the influence of inelastic rotational Raman scattering on solar Fraunhofer lines (Chance and Spurr, 1997; Grainger and Ring, 1962)."

Please check again your error discussion - I agree with the reviewer, that it could be improved
Re: Thank you for this comment. We have changed our demonstrations in Section 2.4 and Supplementary Sect. S3 as follows.
Section 2.4: "Algorithm error (i.e., the difference between the measured and modeled DSCDs) mainly arises from an imperfect representation of the real radiation field in the RTM - spatial inhomogeneities of absorbers and aerosols, clouds, real aerosol phase functions etc."
Supplementary Sect. S3: "Algorithm error is the discrepancy between the measured ($\boldsymbol{y}$) and modelled DSCDs ($F(\boldsymbol{x}, \boldsymbol{b})$). As displayed in Eq. s3, the error sources that result in this discrepancy include forward model error from an imperfect approximation of forward function F, forward model parameter error from selection of parameters $\boldsymbol{b}$, and errors not related to the forward function parameters, like detector noise (Rodgers, 2004). Algorithm error is a function of the viewing angle. Due to the difficulty of assigning this error to each altitude of profile, the algorithm errors on the near-surface values and column densities are usually estimated by calculating the average relative differences between the measured and modeled DSCDs at the minimum and maximum elevation angle (except 90°), respectively (Wagner et al., 2004).

$$\sigma_{algorithm} = \mathbf{y} - F(\mathbf{x}, \mathbf{b}) \tag{s3}$$

where $F(\boldsymbol{x}, \boldsymbol{b})$ is the forward model; $\boldsymbol{b}$ represents the meteorological parameters; $\boldsymbol{y}$ is the measured DSCDs; $\boldsymbol{x}$ is the state vector."

In the abstract (line 25), please add the region (China or more specific) your measurements took place.
Re: Thank you for this comment.
"… and analyzed three typical transport phenomena." -> "… and analyzed three typical transport phenomena over the North China Plain (NCP) and Yangtze River Delta (YRD)."

Page 4, line 3 please remove "technological"
Re: "…, providing technological support for horizontal pollutant transport analysis." -> ", providing support for horizontal pollutant transport analysis."

Page 4, line 95: While it is correct that MAX-DOAS does not require absolute radiometric calibration, to my knowledge, the other techniques mentioned also don't require it. This is not the main advantage of MAX-DOAS.
Re: "Compared with the above techniques, MAX-DOAS does not require radiometric calibration and has many other advantages such as simple design, low power demand, possible automation, low cost, and minimal maintenance." -> "Compared with the above techniques, MAX-DOAS has many advantages such as simple design, low power demand, possible automation, low cost, and minimal maintenance."

Table 2: Please correct the units for O4

Re: "293 K, $I_0$ correction (SCD of $3 \times 10^{43}$ molecules cm$^{-2}$); (Thalman and Volkamer, 2013)" -> "293 K, $I_0$ correction (SCD of $3 \times 10^{43}$ molecules$^2$ cm$^{-5}$); (Thalman and Volkamer, 2013)"

Page 8, line 144: I do not think that this is what happens in your analysis. You cannot derive the SCDs and then take the difference - I assume that you derive the DSCD directly by using the zenith observation as the background I_0 in the DOAS equation.
Re: Thank you for this comment. We added this sentence to explain what SCDs are, but our demonstration here was inaccurate. We have corrected this mistake.
"Spectral analysis derives the slant column densities (SCDs), i.e., the integrated concentration along the light path. Subsequently, we calculated the differential slant column densities (DSCDs), which are defined as the difference between the off-zenith and zenith SCDs." -> "Slant column density (SCD) is defined as the integrated concentration along the light path. Firstly, we calculated the differential slant column densities (DSCDs), which are defined as the difference between the off-zenith and zenith SCDs. Subsequently, we analyzed the…"

Page 8, line 154: The explanation of the detection limit seems not correct

Re: Thank you for your comment. We have corrected this mistake and recalculated the detection limit.

"We assumed two times the fitting error RMS as the DSCD detection limits (Wang et al., 2017; Lampel et al., 2015), which were $7 \times 10^{41}$ (molec$^2 \cdot$cm$^{-5}$), $1.6 \times 10^{15}$, $3.6 \times 10^{15}$, and $5.8 \times 10^{14}$ molec·cm$^{-2}$ for O$_4$, NO$_2$, HCHO, and HONO, respectively."

-> "The DSCD detection limits were roughly estimated using two times of the mean RMS divided by the absorption cross-section (Nasse et al., 2019; Wang et al., 2017; Lampel et al., 2015), which were $2.4 \times 10^{42}$ (molec$^2 \cdot$cm$^{-5}$), $1.7 \times 10^{15}$, $8.9 \times 10^{15}$, and $2.5 \times 10^{15}$ molec·cm$^{-2}$ for O$_4$, NO$_2$, HCHO, and HONO, respectively."

Page 9, line 168: I assume that the RTM model covers the full atmosphere and only your inversion assumes that there is no aerosol / trace gas above 3 km? Is that a good assumption in case of transport events?
Re: Thank you for this comment. This suggestion reminds us that we have missed some retrieval information in our manuscript. We have added it in our manuscript as follows.
"The vertical distribution of trace gas above the retrieval height (3 km) was fixed to follow the U.S. Standard Atmosphere (Anderson et al., 1986)."
For some high concentration trace gases at high altitudes (e.g., O$_3$), their stratospheric profiles have been contained in RTM (Mayer and Kylling, 2005; Emde et al., 2016).

Page 9, line 170: In your description of the a priori, the surface concentration and the column are given, but not the assumed shape of the profile. Please add.
Re: Thank you for this comment. We have added the shape information of a priori profile in front of these sentences as follows.
"Exponentially decreasing functions with a scale height of 0.5, 0.5, 1.0, and 0.2 km were utilized as a priori profiles for aerosols, NO$_2$, HCHO, and HONO, respectively."

Page 9, line 190: I assume this difference is not from the "imperfect minimum of the cost function" but rather from the imperfect representation of the real radiation field in the RTM - spatial inhomogeneities of absorbers and aerosols, clouds, real aerosol phase functions etc.
Re: Thank you for this comment. We have changed our demonstration in Section 2.4 as follows.
"Algorithm error (i.e., the difference between the measured and modeled DSCDs) mainly arises from an imperfect representation of the real radiation field in the RTM - spatial inhomogeneities of absorbers and aerosols, clouds, real aerosol phase functions etc."

Overall flux discussion: In my opinion, the units you use are not a good choice. If you are interested in transport, then the number of molecules transported is relevant, not the

mixing ratio. The same mixing ratio at 3 km and at 0 km altitude lead to different numbers of transported molecules. Please convert to concentrations (molec / m3). This will also impact on your discussion of the most important transport levels.

Re: Thank you for this comment. We have supplemented the unit conversion process in Supplementary Sect. 4 as follows.

"To better demonstrate transport flux, we needed to convert trace gas mixing ratio (ppb) into molecular density (molec $\cdot$ m$^{-3}$) at first. The conversion formula involves temperature and pressure at different altitudes as follows.

$$C = \frac{X \bullet N_A}{V_m} \times 10^{-9} = \frac{X \bullet N_A \bullet P}{R \bullet T} \times 10^{-9}$$

(s12)

where $C$ denotes the trace gas molecular density (molec$\cdot$m$^{-3}$), and $X$ is trace gas mixing ratio (ppb); $N_A$ is Avogadro constant ($6.02 \times 10^{23}$ mol$^{-1}$); R is molar gas constant, with a value of 8.314 J$\cdot$mol$^{-1}\cdot$K$^{-1}$; P and T represent the atmospheric pressure and temperature at different altitudes, respectively. Berberan-Santos et al. (1997) described a relationship model which represents well the dependence of pressure and temperature on altitude for the whole troposphere (below 11 km) as follows.

$$T(z) = T_0 - \beta z$$

(s13)

$$P(z) = P(0)(1 - \frac{\beta z}{T_0})^{\frac{mg}{k\beta}}$$

(s14)

Here, T(z) and P(z) denote the temperature and atmospheric pressure at height z (km), respectively; $T_0$ and P(0) are the surface values; $k$ is Boltzmann constant ($1.38 \times 10^{-23}$ J$\cdot$K$^{-1}$); m is air molecular mass ($29 \times 10^{-3}$ kg$\cdot$mol$^{-1}$); g represents acceleration of gravity (9.8 m$\cdot$s$^{-2}$); $\beta$ equals 6.5 K$\cdot$km$^{-1}$."

Besides, we have re-depicted the transport flux variation figures (Fig. 4 and Fig. 5), and changed the transport flux descriptions in Section 3.1.

[Figure]

"Except that $F_i$ reached the highest level of ~ 1.8 × 10$^{18}$ molec·m$^{-2}$·s$^{-1}$ in the 400–600 m layer at 16:00 at the SJZ station, the other highest $F_i$ all occurred below 400 m at any station and at any time. This indicated that the MTL of NO$_2$ was 0–400 m. Near-surface NO$_2$ emission sources (e.g., vehicle and factory emissions) might be the main reason for this phenomenon. Compared with aerosols and NO$_2$, we found that high-value HCHO $F_i$ extended to higher altitudes. Taking CAMS as an example, we found the strongest HCHO $F_i$ constantly emerging at 1000–1200 m from 8:00 to 13:00, and averaging 2.51 × 10$^{17}$ molec·m$^{-2}$·s$^{-1}$. During the same period, surface HCHO $F_i$ only averaged 1.72 × 10$^{17}$ molec·m$^{-2}$·s$^{-1}$. However, at the CAMS station, the surface HCHO concentration was much higher than that of the 1000–1200 m layer between 8:00 and 13:00 (Fig. 3), proving that high-altitude transport contributed more to overall HCHO transport. After 10:00, we found that the highest HCHO $F_i$ gradually increased from ~ 3.5 × 10$^{17}$ to ~ 4.5 × 10$^{17}$ molec·m$^{-2}$·s$^{-1}$ at WD, with the MTL of HCHO ranging from 400 to 1000 m. At station SJZ, the strongest HCHO $F_i$ increased from ~ 2.6 × 10$^{17}$ to ~ 4.5 × 10$^{17}$ molec·m$^{-2}$·s$^{-1}$ during 11:00–17:00, with the highest transport fluxes occurring mostly at 400–800 m. These findings indicated that the MTL of HCHO was mainly 400–1200 m."

[Figure]

Flux discussion: In addition to using concentrations, you should convert your numbers to real fluxes by integrating over a unit area of 1 x 1 m2. This will then result in a flux in units of molec / s through a unit area. In the caption of Figure 4, you claim that this is what you show, but the units you give to not match that claim.

Re: Thank you for this comment. We didn't use the $flux_i$ in units of molec / s, because the $flux_c$ was calculated by multiplying $flux_i$ and $H_i$, which would result in the $flux_c$ in units of $molec \cdot m \cdot s^{-1}$. This would result in a confusing physical meaning. After comprehensively considering, we planned to use the $flux_i$ in units of $molec \cdot m^{-2} \cdot s^{-1}$ and integrated the $flux_c$ over a unit width(1 m), making the $flux_c$ unit $molec \cdot s^{-1}$.

[Figure]

"For NO₂ transport, the average $F_c$ values at SJZ ($1.56 \times 10^{21}$ molec·s⁻¹), NC ($1.10 \times 10^{21}$ molec·s⁻¹), and CAMS ($1.58 \times 10^{21}$ molec·s⁻¹) were substantially higher than those at WD ($5.57 \times 10^{20}$ molec·s⁻¹). Conversely, the average $F_c$ of HCHO was the highest in WD ($8.82 \times 10^{20}$ molec·s⁻¹), whereas the $F_c$ values in SJZ, NC, and CAMS were $4.81 \times 10^{20}$, $5.16 \times 10^{20}$, and $5.12 \times 10^{20}$ molec·s⁻¹, respectively."

"Figure 4. Transport flux per unit cross-sectional area at different altitudes ($Flux_i$) at CAMS, NC, WD, and SJZ stations on February 5, 2021." -> "Figure 4. Transport flux at different altitudes ($Flux_i$) at CAMS, NC, WD, and SJZ stations on February 5, 2021."

Section 3.2: In the presence of a dust storm, the vertical sensitivity of the MAX-DOAS measurements will change. Have you considered that? It may explain some of your observations (disappearance of elevated layers).

Re: Thank you for this comment. It is difficult to exclude this possibility, because it is not easy to design the experiment to measure the effects of dust on vertical sensitivity. If placing the two identical instruments in two nearby places, the influence range of sand storm is so large that can cover the whole places. However, if the distances between two instruments are too far, the different environment would be introduced as another uncertain factor. In a word, it is hard to control variables and measure the effects of dust on the vertical sensitivity by doing experiments.

A previous study (using LiDAR) on dust storm also indicated that dust layers would inhibit the dissipation of pollutants and enhance surface air pollution, by depressing the PBL and weaken the turbulent exchange (Wang et al., 2020). This conclusion is in agreement with our findings. Thus, we thought the profile shape changes were more attributed to the accumulation of surface pollutants, and the effects of vertical sensitivity variation was relatively little compared to the actual pollutant increases.

However, we cannot completely exclude the possibility of the effects of dust on vertical sensitivity. Therefore, we have added it as one of uncertainties and demonstrated it at the end of Section 3.2.

"The comparison result between the dusty day and two clean days makes it possible to better understand the impacts of dust storm on local environment. However, there remain some uncertainties in this discussion. Although we selected the closest clean

days to lessen the effects of some factors (e.g., climate and temperature) on comparison, the uncertainties caused by other meteorological parameters (e.g., wind speed and directions) were unknown to us, since we did not make sure these parameters were nearly the same on these three days. Therefore, this comparison analysis is based on the assumption that there is little difference between meteorological parameters on various days or the effects caused by different meteorological parameters are negligible. Besides, a dust storm would trigger changes at the vertical sensitivity of MAX-DOAS measurements, which might influence profile shape. These impact factors are difficult to control in observations, and modelling correction may be a good solution."

Section 3.2: The discussion is based on three days - two "clean" days and one day during the dust storm. While your discussion is plausible, you should acknowledge that it is based on the assumption, that everything else is the same on the three days. This assumption is probably not correct - these are different days of the week (one is a Saturday), wind speed and directions are different and also the accumulation history of pollution in the BL is different. Very large day to day variations are observed at many stations without dust storms. Please at least mention this possibility.

Re: Thank you for this comment. This is one of uncertainties in our discussion, and we have added this uncertainty at the end of Section 3.2 as follows.

"The comparison result between the dusty day and two clean days makes it possible to better understand the impacts of dust storm on local environment. However, there remain some uncertainties in this discussion. Although we selected the closest clean days to lessen the effects of some factors (e.g., climate and temperature) on comparison, the uncertainties caused by other meteorological parameters (e.g., wind speed and directions) were unknown to us, since we did not make sure these parameters were nearly the same on these three days. Therefore, this comparison analysis is based on the assumption that there is little difference between meteorological parameters on various days or the effects caused by different meteorological parameters are negligible. Besides, a dust storm would trigger changes at the vertical sensitivity of MAX-DOAS measurements, which might influence profile shape. These impact factors are difficult to control in observations, and modelling correction may be a good solution."

References

Anderson, G. P., Clough, S. A., Kneizys, F., Chetwynd, J. H., and Shettle, E. P.: AFGL atmospheric constituent profiles (0.120 km), Tech. rep., AIR FORCE GEOPHYSICS LAB HANSCOM AFB MA, 1986.

Chance, K. and Spurr, R. J. D.: Ring effect studies: Rayleigh scattering, including molecular parameters for rotational Raman scattering and the Fraunhofer spectrum, Applied Optics, 36, 5224–5230, 10.1364/AO.36.005224, 1997.

Emde, C., Buras-Schnell, R., Kylling, A., Mayer, B., Gasteiger, J., Hamann, U., Kylling, J., Richter, B., Pause, C., Dowling, T., and Bugliaro, L.: The libRadtran software package for radiative transfer calculations (version 2.0.1), Geoscientific Model Development, 9, 1647-1672, 10.5194/gmd-9-1647-2016, 2016.

Grainger, J. F. and Ring, J.: Anomalous Fraunhofer line profiles, Nature, 193, 762,

10.1038/193762a0, 1962.

Mayer, B. and Kylling, A.: Technical note: The libRadtran software package for radiative transfer calculations – description and examples of use, Atmospheric Chemistry and Physics, 5, 1855-1877, 10.5194/acp-5-1855-2005, 2005.

Rodgers, C. D., Taylor, F. W. (Ed.): Inverse methods for atmospheric sounding, theory and practice, World Scientific, 255 pp.2004.

Wang, Z., Liu, C., Xie, Z., Hu, Q., Andreae, M. O., Dong, Y., Zhao, C., Liu, T., Zhu, Y., Liu, H., Xing, C., Tan, W., Ji, X., Lin, J., and Liu, J.: Elevated dust layers inhibit dissipation of heavy anthropogenic surface air pollution, Atmospheric Chemistry and Physics, 20, 14917-14932, 10.5194/acp-20-14917-2020, 2020.

*Thank you for your careful review and constructive suggestions. These suggestions are quite valuable to us, and help improve our manuscript a lot.*

**Point-to-point responses**

*We appreciate the reviewers for their valuable and constructive comments, which are very helpful for the improvement of the manuscript. We have revised the manuscript carefully according to the reviewers' comments. We have addressed the reviewers' comments on a point-to-point basis as below for consideration, where the reviewers' comments are cited in **black**, and the responses are in **blue**.*

The authors refer to the Ring effect as a kind of stratospheric correction. This is not correct. Instead, the Ring effect accounts for the filling-in of Fraunhofer lines caused by inelastic rotational Raman scattering [Grainger and Ring, 1962; Chance and Spurr, 1997]. This needs to be corrected.

Re: Thank you for this comment.

"The Ring spectrum was added to the fitting settings to remove the influence of the stratosphere on the DSCDs." -> "The Ring spectrum was added to the fitting settings to remove the influence of inelastic rotational Raman scattering on solar Fraunhofer lines (Chance and Spurr, 1997; Grainger and Ring, 1962)."

I feel that the definition of an "algorithmic error" is still not appropriate and remains vague. It is now mentioned that the "Algorithmic error (i.e., the difference between the measured and modeled DSCDs) arises from an imperfect minimum of the cost function." It is not clear to me what that means. It is hard to imagine that the minimization algorithm (e.g, Gauss-Newton or Levenberg-Marquard-Algorithm) is not working correctly and does not yield a minimum of the cost function. Do you suspect that the algorithm ends up in a local minimum of the cost function that is different form the global cost function? This can happen, but you would need to show that this is the case which is difficult to do. Usually, one distinguishes between the following error components [Rodgers, 2000], which should be discussed accordingly:

1. Smoothing error caused by the limited vertical resolution due to the limited information content of the measurements. These are discussed in the manuscript based on the averaging kernels.

2. Forward model errors due to imperfect representation of the physics of the system. In case of profile retrieval, this could for example be caused by horizontal inhomogeneities of trace gases and aerosols.

3. Retrieval noise caused by the noise of the measurements

4. Forward model parameter errors, caused by errors or incomplete knowledge of model parameters, such as the aerosol phase function.

Re: Thank you for this comment. As you said, it might be inaccurate to describe like "…arises from an imperfect minimum of the cost function". Algorithmic error points to the difference between the measured ($y$) and modeled DSCDs ($F(x, b)$) as follows.

$$\sigma_{a\lg orithm} = \mathbf{y} - F(\mathbf{x}, \mathbf{b})$$

where $F(x, b)$ is the forward model; $b$ represents the meteorological parameters; $y$ is the measured DSCDs; $x$ is the state vector.

In an ideal situation, the modeled DSCDs should equal the measured ones. The error sources that result in this difference include forward model error from an imperfect approximation of forward function F, forward model parameter error from selection of parameter $b$, and errors not related to the forward function parameters (e.g., detector noise). In other words, algorithmic error contains forward model error and forward model parameter errors, which are the main contributors, but not identical to the sum of them. On the other hand, the forward model error is hard to be quantified due to the difficulty of acquiring an improved forward model.

We have changed our demonstrations in Section 2.4 and Supplementary Sect. S3 as follows.

Section 2.4: "Algorithm error (i.e., the difference between the measured and modeled DSCDs) mainly arises from an imperfect representation of the real radiation field in the RTM - spatial inhomogeneities of absorbers and aerosols, clouds, real aerosol phase functions etc."

Supplementary Sect. S3: "Algorithm error is the discrepancy between the measured ($y$) and modelled DSCDs ($F(x, b)$). As displayed in Eq. s3, the error sources that result in this discrepancy include forward model error from an imperfect approximation of forward function F, forward model parameter error from selection of parameters $b$, and errors not related to the forward function parameters, like detector noise (Rodgers, 2004). Algorithm error is a function of the viewing angle. Due to the difficulty of assigning this error to each altitude of profile, the algorithm errors on the near-surface values and column densities are usually estimated by calculating the average relative differences between the measured and modeled DSCDs at the minimum and maximum elevation angle (except 90°), respectively (Wagner et al., 2004).

$$\sigma_{a\lg orithm} = \mathbf{y} - F(\mathbf{x}, \mathbf{b}) \qquad (s3)$$

where $F(x, b)$ is the forward model; $b$ represents the meteorological parameters; $y$ is the measured DSCDs; $x$ is the state vector."

References

Chance, K. and Spurr, R. J. D.: Ring effect studies; Rayleigh scattering, including molecular parameters for rotational Raman scattering and the Fraunhofer spectrum, Appl. Opt., 36, 5224–5230, https://doi.org/10.1364/AO.36.005224, 1997.

Grainger, J. F. and Ring, J.: Anomalous Fraunhofer line profiles, Nature, 193, 762, https://doi.org/10.1038/193762a0, 1962.

Rodgers, C. D.: Inverse methods for atmospheric sounding, theory and practice, edited by: Taylor, F. W., World Scientific, 2000.

Rodgers, C. D., Taylor, F. W. (Ed.): Inverse methods for atmospheric sounding, theory and practice, World Scientific, 255 pp.2004.